# TIVelo: RNA velocity estimation leveraging cluster-level trajectory inference

Muyang Ge ⓘ, Jishuai Miao ⓘ, Ji Qi, Xiaocheng Zhou & Zhixiang Lin ⓘ ✉

RNA velocity inference is a valuable tool for understanding cell development, differentiation, and disease progression. However, existing RNA velocity inference methods typically rely on explicit assumptions of ordinary differential equations (ODE), which prohibits them from capturing complex transcriptomic expression patterns. In this study, we introduce TIVelo, a RNA velocity estimation approach that first determines the velocity direction at the cell cluster level based on trajectory inference, before estimating velocity for individual cells. TIVelo calculates an orientation score to infer the direction at the cluster level without an explicit ODE assumption, which effectively captures complex transcriptional patterns, avoiding potential inconsistencies in velocity estimation for genes that do not follow the simple ODE assumption. We validated the effectiveness of TIVelo by its application to 16 real datasets and the comparison with six benchmarking methods.

Single-cell RNA sequencing (scRNA-seq) has revolutionized our understanding of cellular diversity and complexity by enabling transcriptome analysis at the individual cell level[1]. Trajectory inference (TI) leverages scRNA-seq data to order individual cells/cell clusters along a trajectory based on gene expression, thereby providing insights into cellular differentiation and development[2,3]. TI methods can be classified into two main categories: The first category, such as Monocle[4], Slingshot[5], and Palantir[6], involves the direct construction of a cell graph, where each node represents a cell and each edge denotes a connection between cells. The second category, such as PAGA[7], involves an initial clustering of the dataset, followed by the construction of a cluster graph, where each node represents a cell cluster and each edge denotes a connection between these clusters. One method closely related to TI is pseudotime analysis, which aims to order individual cells along a continuous trajectory representing a biological process. The continuous trajectory is often based on the result of TI analysis. A significant challenge associated with pseudotime analysis is its requirement for prior information, where a starting cell or cluster needs to be chosen with pseudotime set as 0. Acquiring this information can be difficult in practice, presenting a substantial obstacle to the effective application of pseudotime analysis methods.

RNA velocity is a concept that provides a dynamic view of cellular behavior. Compared with pseudotime analysis, RNA velocity estimation typically does not require prior information such as the starting cell or cluster, and can provide developmental velocity direction and intensity for each cell, instead of the single pseudotime value. The basic premise of RNA velocity lies in leveraging the information contained within the two forms of RNA for each gene: unspliced (nascent) RNA and spliced (mature) RNA, denoted by $u$ and $s$, respectively. Typically, spliced RNA $s$ is produced from unspliced RNA $u$, and the rate of spliced RNA $s$ production $\frac{ds}{dt}$ is often referred to as RNA velocity. The sign (positive or negative) of the RNA velocity of a specific gene within a cell can indicate the future regulation (upregulation or downregulation) of that gene.

The concept of RNA velocity was first proposed by velocyto[8], which provides a standard pipeline from extracting unspliced and spliced RNA counts from sequencing data, to estimating RNA velocity by a steady-state model based on the ordinary differential equation (ODE) assumption. scVelo[9], a successor to velocyto, still relies on the ODE assumption but leverages the EM algorithm to iteratively update the ODE rate parameters of each gene and the pseudotime of each cell, thereby fitting $u$-$s$ expression of each gene through an ODE curve. Following this, UniTVelo[10] introduces a radial basis function-based curve fitting strategy, and supports a unified pseudotime of each cell across different genes. Contrarily, VeloAE[11] does not directly infer velocity in the high-dimensional gene expression space. Instead, it projects the $u$-$s$ expression into a low-dimensional embedding space through an autoencoder (AE), and further estimates the velocity based

Department of Statistics, The Chinese University of Hong Kong, Shatin, Hong Kong SAR, China. ✉e-mail: zhixianglin@cuhk.edu.hk

on the latent embedding. Analogous methods include VeloVAE[12], which leverages a variational autoencoder[13] (VAE). It infers latent pseudotime and ODE rate parameters by an encoder using $u$ and $s$ as the input, and then generates $u$ and $s$ expression by a decoder to minimize the reconstruction loss of $u$ and $s$. VeloVI[14] shares a similar approach but employs a Bayesian deep generative model, outputting posterior distributions of ODE rate parameters and thus velocities. Another category of RNA velocity inference methods applies Neural ODE[15], such as scTour[16] and LatentVelo[17]. These methods first embed $u$ and $s$ expression into a low-dimensional latent space, then use Neural ODE to fit the developmental process for latent embedding along the cell trajectory. Instead of building an ODE model to fit the $u$-$s$ expressions of all cells for each gene, and then infer RNA velocity for each cell according to the fitted velocities of all genes, DeepVelo[18] and cellDancer[19] infer RNA velocity for each cell directly, based on the $u$-$s$ expressions in that cell's nearest neighborhood. In addition, RNA velocity estimation methods that combine unspliced and spliced RNA with other biological information have been developed. For instance, MultiVelo[20] uses chromatin accessibility, protaccel[21] uses protein abundances, Dynamo[22] uses new/total labeled RNA-seq, PhyloVelo[23] uses phylogenetic trees, and TFvelo[24] uses transcription factors (TFs).

Current RNA velocity estimation methods come with several limitations. Firstly, most RNA velocity inference methods are based on the ODE assumption, which assumes that the transcription process follows a simple ODE model, with constant rate parameters of each gene. Although it is a rough approximation to the transcription process and can easily achieve analytical solutions in practice, the naive ODE model fails to deal with complicated transcription dynamics[25,26]. Some variants of the ODE model attempt to address this by introducing variable rate parameters, including DeepVelo and cellDancer; their limitations are discussed in the Supplementary Note 1. Secondly, the ODE model and its variants are usually fitted for the expression of different genes independently, and the inferred velocities for individual genes are aggregated together as the final estimation. In practice, this strategy may lead to inconsistent or even reversed velocity estimation to the expected direction[10,25–27].

In this study, we introduce TIVelo for RNA velocity estimation. Rather than fitting an ODE model to individual cells and genes, TIVelo initially infers the velocity direction among different cell clusters. This cluster-level velocity estimation provides insights into the broader velocity trends, thereby preventing the potential misdirection in developmental trajectories that can occur when velocities are aggregated from independently fitted genes. To infer the direction on the cell cluster level, TIVelo leverages a model-free strategy based on an intrinsic property of the unspliced-spliced RNA relationship: the unspliced RNA should always be expressed and repressed earlier than the spliced RNA. This strategy allows for comprehensive mining of signals embedded in the $u$-$s$ expression profiles, without complicated mathematical modeling for the RNA transcription process. In this way, it mitigates the issue that ODE rate parameters may vary over different cell stages, which is hard to deal with through an ODE model with constant parameters. We validated TIVelo's capability to accurately infer RNA velocity in 16 real datasets, in comparison with six benchmarking RNA velocity estimation methods.

## Results
### Overview of TIVelo workflow
The basic idea of TIVelo is to first infer the direction on a cluster-level graph and then use this direction to supervise the velocity estimation of each cell. There are three primary steps in TIVelo: main path selection, orientation inference, and RNA velocity estimation.

In the main path selection step, the cluster graph will first be constructed for the dataset. In this cluster graph, each cell cluster is a node and each edge represents the connectivity between a pair of nodes (Fig. 1a). Subsequently, we identify possible "terminal states" in this cluster graph, which refers to either root cluster or end cluster in the data. One terminal state most likely to be a root/end cluster will be selected as the "origin node" (Methods; Fig. 1b). Finally, we select a "main path," which refers to a path in the cluster graph beginning from the origin node, and involving as many cells as possible (Methods; Fig. 1c).

In the orientation inference step, each cell along the main path will be assigned a pseudotime (Fig. 1d). Cells along the main path are then ordered by this pseudotime, forming time series of $u$ and $s$ expression of each gene $g$, which are denoted by $\tilde{u}_g$ and $\tilde{s}_g$. Then a specially designed orientation score $S_g$ will be calculated for gene $g$ based on time series $\tilde{u}_g$ and $\tilde{s}_g$, using the intrinsic property of $u$-$s$ relationship, such that the unspliced RNA $u$ should always be expressed earlier than the spliced RNA $s$ and decrease earlier than $s$ (Methods). The scores from $G$ pre-filtered genes will be aggregated together to evaluate if the current direction on the main path (from origin node to the end of the main path) is correct or not. If $\frac{1}{G}\sum_g S_g < 0$, the direction is considered incorrect, the origin node will be reset (Fig. 1e, f).

In RNA velocity estimation step, we begin from the new origin node and assign a level to each node in the cluster graph. A smaller number of level signifies closer proximity to the root cluster, and the level of the origin node is set at 0. Based on this level, we draw a directed edge from each node to its child nodes, providing a directed trajectory inference (DTI) analysis (Fig. 1g). To infer RNA velocity for each cell, we construct directed nearest neighborhood (dNN) for each cell $n$, referring to the near future state of that cell (Methods; Fig. 1h). The velocity vector of each cell is compelled to point toward the mean expression in the dNN, leading to the RNA velocity estimation results (Fig. 1i).

### TIVelo's efficacy and the comparison with existing methods
The efficacy of TIVelo comes from two aspects. The first aspect is that TIVelo calculates an orientation score for each gene to infer the direction on the main path, which is a simpler task than directly fitting RNA velocity for individual genes. Directly fitting RNA velocity for individual genes is a commonly used strategy by ODE-based methods, but may fail when there exist genes with expression patterns not agreeing with the assumptions in the ODE equation. Moreover, TIVelo divides the entire developmental process into short pseudotime sections and aggregates local transcription patterns in different sections. This strategy can fully exploit transcription features from each gene's $u$-$s$ profiles, instead of constructing an ODE model. On the other hand, ODE-based methods like scVelo assume that all cells should follow an ODE process with constant coefficients, ignoring the fluctuation during the transcription.

To show this point, we compared the performance of scVelo and TIVelo on the mouse gastrulation (erythroid) dataset. In the mouse gastrulation (erythroid) dataset, the expected direction of development is from cell type Blood progenitors 1 to Erythroid 3 (Fig. 2a, b). In addition, there exist multiple rate kinetics genes (MURK genes) in this dataset, which refer to genes whose transcription rate $\alpha$ is non-constant and will suddenly increase during cell development[28]. For convenience, we use the term "non-MURK genes" to denote all other genes that do not exhibit these MURK characteristics.

As shown in Fig. 2a, the ODE curve fitted by scVelo (dynamical mode) for non-MURK genes with large likelihood, such as *Mllt3*, accurately represents the expected developmental lineage from cell type Blood progenitors 1 to Erythroid 3. However, MURK genes, including *Gclm* exhibit a non-constant transcription rate ($\alpha$) over pseudotime (Fig. 2b, top left), which leads to a fitted ODE curve that contradicts the expected developmental direction (Fig. 2b). This limitation of scVelo can be further demonstrated by the velocity stream plot based on non-MURK genes and MURK genes, respectively. While the velocity stream based on non-MURK genes with top likelihood in scVelo can depict the expected direction (Fig. 2c), the velocity stream

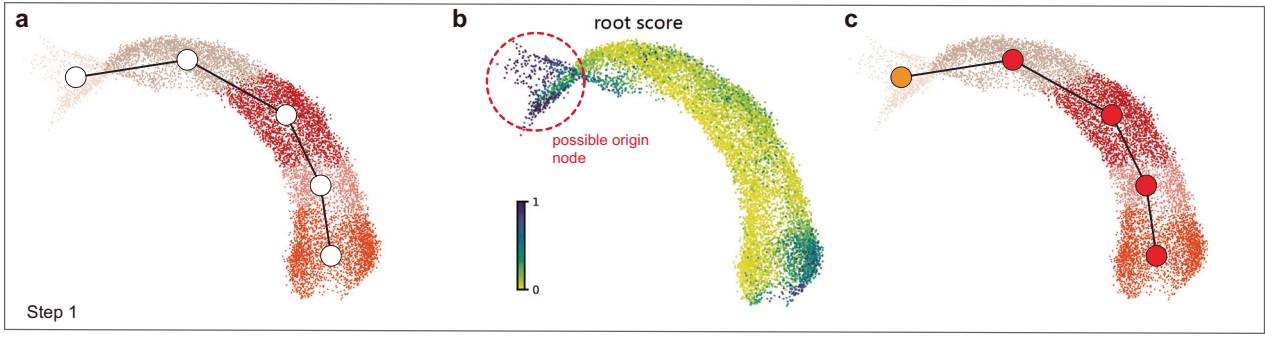

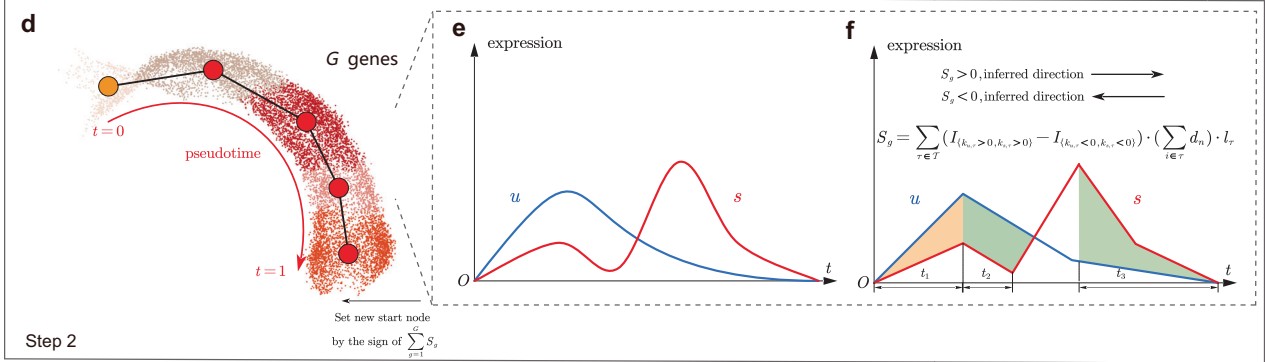

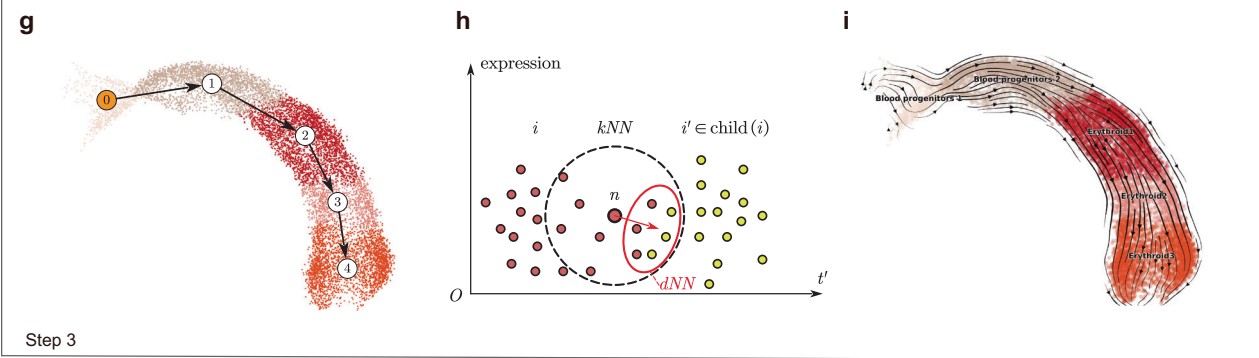

**Fig. 1 | Overview of the TIVelo pipeline. a** Constructing PAGA cluster graph for the dataset. **b** Finding possible cluster by root/end score as origin node. **c** Selecting main path beginning from the origin node. **d** Assigning diffusion pseudotime for cells along the main path. **e** Extracting time series $\bar{u}$ and $\bar{s}$ of each gene according to pseudotime $t$. Here $\bar{u}$ and $\bar{s}$ are rescaled to sum to 1 for each gene. **f** Calculating orientation score for $G$ genes within the dataset. Identifying expected direction of the main path and setting new origin node. **g** Assigning level and child node(s) for each node in the cluster graph. This facilitates directed trajectory inference (DTI) analysis. **h** Building directed nearest neighbor (dNN) for each cell. **i.** Inferring RNA velocity for each cell by dNN.

based on MURK genes presents a completely reversed and unexpected developmental direction (Fig. 2d).

Instead of fitting velocity for individual genes, TIVelo calculates an orientation score for each gene, which is used to infer the direction along the main path. To calculate the orientation score, TIVelo constructs a linear tree[29,30] model for each gene based on its unspliced and spliced RNA counts. Linear tree can help to split the entire cell development process into short pseudotime sections, and within each section TIVelo calculates the corresponding orientation score according to the intrinsic property that unspliced RNA $u$ should always be expressed earlier than spliced RNA $s$ and decrease earlier than $s$. This orientation score provides guidance for inferring direction at the cluster level, and the sign of the score indicates if the initial direction should be kept or reversed (Methods).

In the mouse gastrulation (erythroid) dataset, for non-MURK gene *Mllt3*, TIVelo captures the local transcription pattern from Blood progenitors 2 to Erythroid 1, yielding an orientation score of 18.03 (Fig. 2e, top). Similarly, for MURK gene *Gclm*, it accurately captures the transcription pattern from Erythroid 2 to Erythroid 3, and the orientation score is 13.93 (Fig. 2e, bottom). Consequently, TIVelo successfully obtains positive orientation scores for both genes, supporting the expected cell development trajectory from cluster Blood progenitors 1 to Erythroid 3.

The second aspect of TIVelo's effectiveness lies in its strategy to infer RNA velocity of each cell and each gene following the direction on the cluster graph. Specifically, TIVelo infers the direction on the cluster graph based on the orientation score (Fig. 2h); subsequently, for each cell TIVelo constructs a dNN according to the inferred direction on the cluster graph, which refers to the future state of that cell. For each gene of that cell, the inferred velocity should point toward the mean spliced expression in dNN (Method; Fig. 2i). The superiority of this strategy can be seen from the visualization of the velocity graph[9], which is the correlation between velocity and cell-to-cell transition for each pair of cells $(n, n')$. For mouse gastrulation (erythroid) dataset, the inferred velocity vector of cell $n$ can be written as $\boldsymbol{v}_n = (\boldsymbol{v}_{n,1}^\top, \boldsymbol{v}_{n,2}^\top)^\top$, where $\boldsymbol{v}_{n,1}$ and $\boldsymbol{v}_{n,2}$ are the velocities corresponding to MURK and non-MURK genes, respectively. For scVelo, since the velocities from MURK and non-MURK genes may have completely opposite directions (Fig. 2c, d),

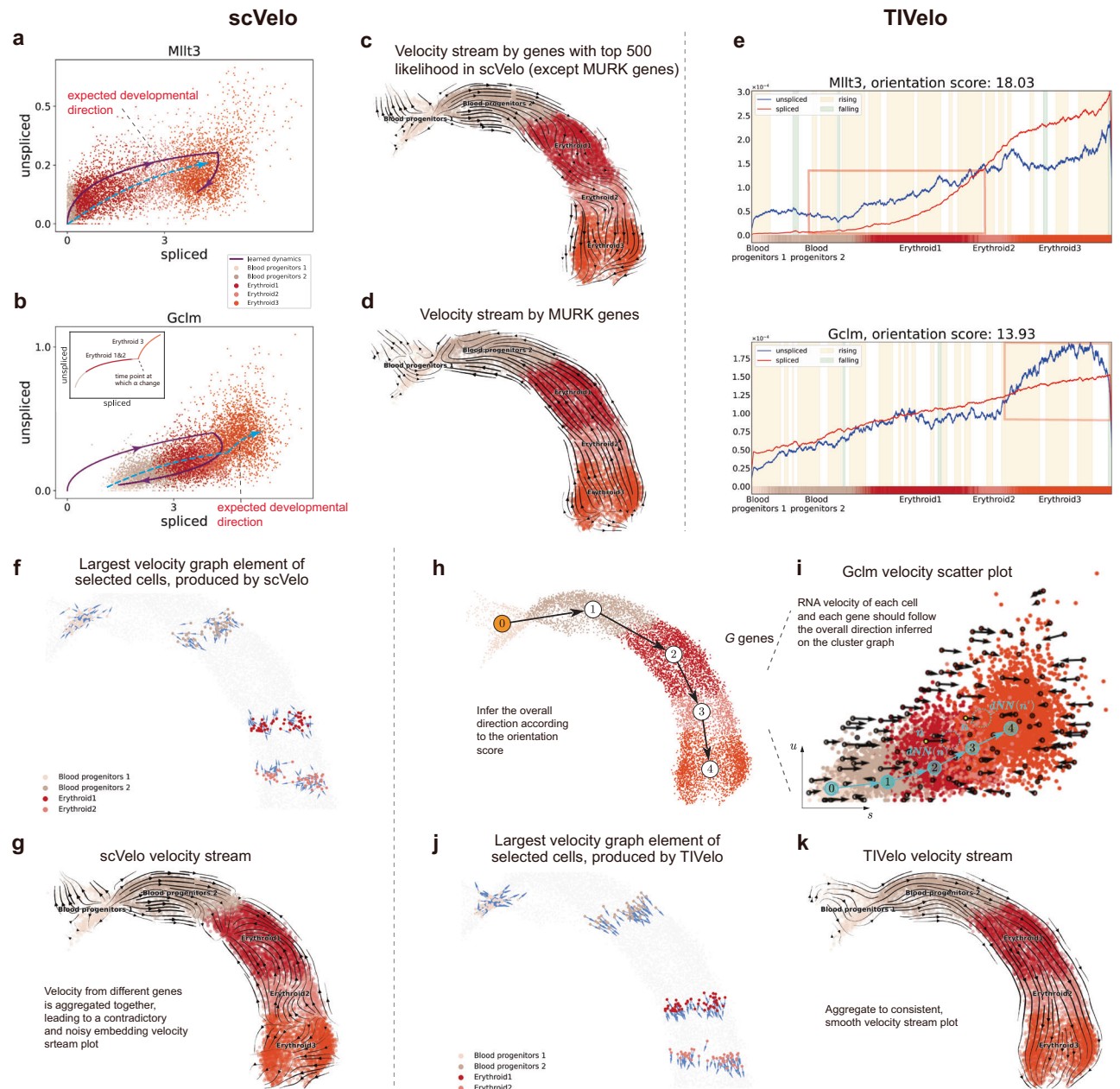

**Fig. 2 | TIVelo's efficacy and the comparison with ODE-based method. a** Gene *Mllt3* in mouse gastrulation (erythroid) fitted by ODE curves of scVelo. *Mllt3* is a gene with top likelihood fitted by scVelo. **b** Gene *Gclm* in mouse gastrulation (erythroid) fitted by ODE curves of scVelo. *Gclm* is a multiple rate kinetics gene (MURK gene), with rate parameter $\alpha$ suddenly increasing at some time point[25,28] (top left panel). **c** RNA velocity stream plot inferred from scVelo, based on genes with top 500 likelihood except MURK genes. **d** RNA velocity stream plot inferred from scVelo, based on MURK genes. **e** The *u-s* pattern along the main path for corresponding two genes. In each panel the colorbar below indicates the cell type, which is identical to the annotation in (**a**). **f** The velocity graph for selected cells created by scVelo. There is a unit length arrow showing the direction from selected cell $n$ to cell $n'$ on the UMAP embedding plot if $n' = \mathrm{argmax}_{n'} \pi_{n,n'}$. The line width of each arrow is proportional to its corresponding $\pi_{n,n'}$. **g** RNA velocity stream plot produced by scVelo. **h** TIVelo first infers the direction on the cluster level, according to the orientation scores. **i** In RNA velocity estimation, for each cell and each gene, the inferred velocity should follow the direction inferred in (**h**). **j** The velocity graph for selected cells created by TIVelo. **k** RNA velocity stream plot produced by TIVelo.

$\boldsymbol{v}_{n,1}$ and $\boldsymbol{v}_{n,2}$ can indicate totally different descendant cell populations for cell $n$. This leads to the lack of clear directionality toward expected future state in the velocity graph for the cells (Fig. 2f). In contrast, TIVelo's strategy will ensure both $\boldsymbol{v}_{n,1}$ and $\boldsymbol{v}_{n,2}$ pointing toward cells $n'$ within the dNN of cell $n$. As a result, only cells $n'$ within the dNN of cell $n$ will become the descendant cell populations of cell $n$, and these cells will have a large value of velocity graph. This leads to a velocity graph correctly reflecting the cell developmental direction (Fig. 2j).

Finally, by using the embedding position of each cell $n$, the embedding velocity stream plot can be drawn based on the velocity graph (Methods). The embedding velocity stream plot produced by scVelo presents a pattern contradictory to the expected developmental trajectory (Fig. 2g). TIVelo, in contrast, yields a velocity stream plot that aligns with the expected trajectory, with each cell's velocity more consistent with the velocity stream in its neighborhood (Fig. 2k).

## Cluster level direction facilitates velocity inference for individual cells

To evaluate the effectiveness of TIVelo, we investigated its application to the dentate gyrus dataset. In this dataset, the expected root cell

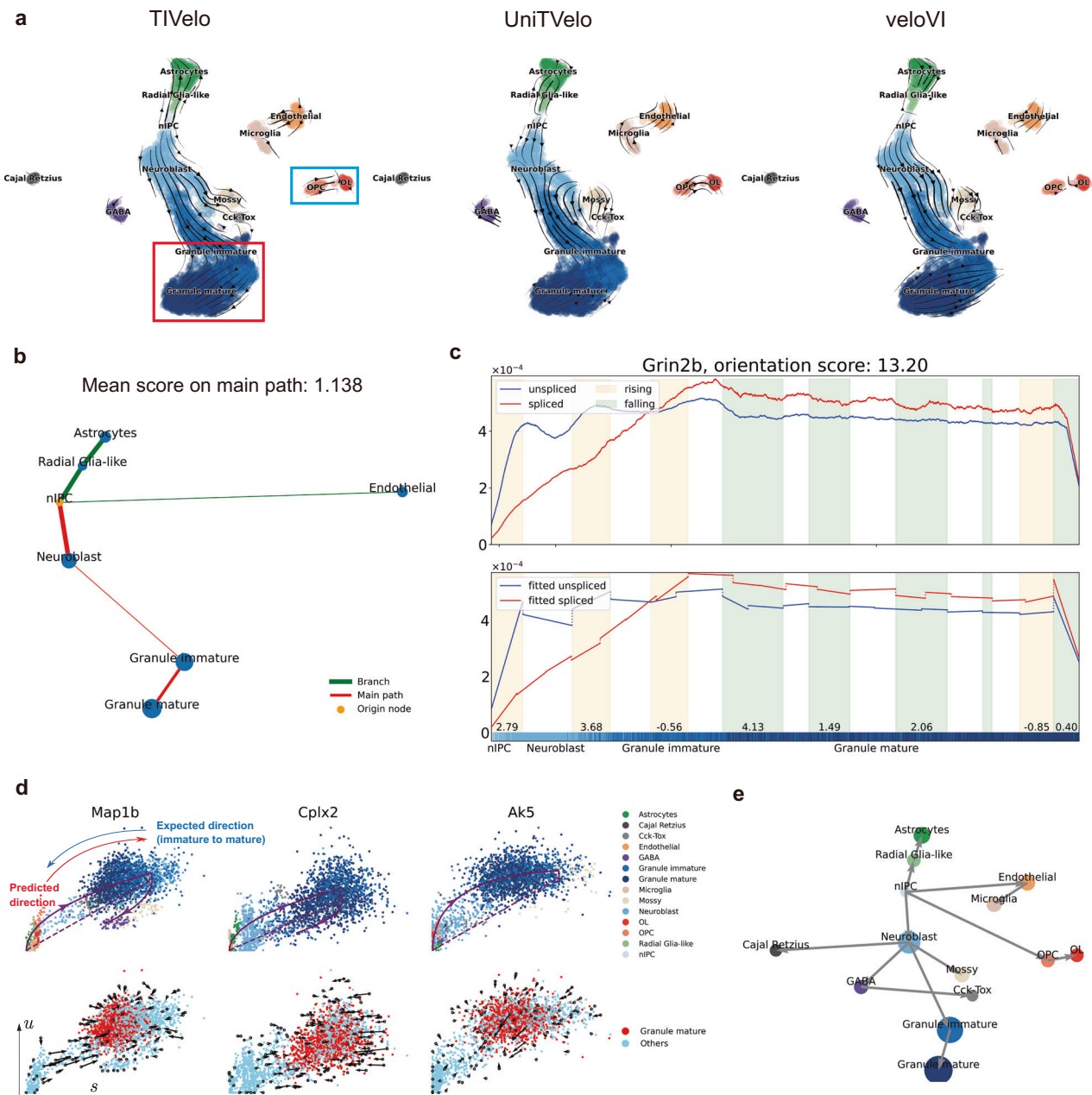

**Fig. 3 | Comparative analysis of TIVelo's identification of the developmental trajectory and velocity direction for dentate gyrus. a** Comparative RNA velocity stream plots of TIVelo, UniTVelo, and veloVI, highlighting TIVelo's inference of the velocity stream from immature to mature Granule (indicated by the red box) and from OPC to OL (blue box). **b** The cluster graph after graph pruning and main path selection. **c** The $\bar{u}$-$\bar{s}$ variation along the main path and the linear tree model constructed for gene *Grin2b*, with the orientation score of primary rising/falling section annotated below. The colorbar indicates the cell type, which is identical to the annotation in (**a**). **d** Velocity fitting for genes *Map1b*, *Cplx2* and *Ak5* from TIVelo. Top row: *u-s* scatter plots colored by cell type, with fitted ODE curve from scVelo (dynamical mode). Bottom row: fitted velocity vectors of sampled cells from TIVelo. The Granule mature cells are colored by red, and other cell types are colored by light blue. **e** Directed trajectory inference (DTI) analysis by TIVelo.

cluster is intermediate progenitor cells for neurons (nIPC). This dentate gyrus dataset contains two primary lineages: one is from nIPC to Granule mature, and the other is from nIPC to Astrocytes (Fig. 3a). There are several island-like cell clusters in the UMAP visualization of this data, such as oligodendrocyte precursor cells (OPC) and oligodendrocytes (OL) (Fig. 3a), which add an extra layer of complexity to RNA velocity estimation.

The comparison of RNA velocity stream plots of TIVelo, UniTVelo, and veloVI are shown in Fig. 3a. TIVelo, UniTvelo and veloVI provide generally accurate velocity estimation from nIPC to other lineages. However, the result of TIVelo highlights the developmental trajectory from Granule immature to Granule mature (Fig. 3a, red box). Conversely, the result of UniTVelo and veloVI displays an apparent reversed velocity flow from Granule mature to Granule immature. Another expected velocity flow from OPC to OL in the oligodendrocyte lineage is observed in the velocity stream plot of TIVelo (Fig. 3a, blue box), which is not depicted in the result of veloVI. The velocity stream plots produced by scVelo (stochastic mode), scVelo (dynamical mode), DeepVelo and cellDancer are shown in Supplementary Fig. 1c. The reversed velocity flow from Granule mature to Granule immature exists in results produced by both modes of scVelo, while the result from cellDancer cannot reflect the expected developmental trajectory.

The improvement of such details in TIVelo is primarily due to its strategy of first inferring the direction on the cluster graph. Figure 3b shows the cluster graph of the dentate gyrus after graph pruning and main path selection (Methods). The expected root cluster nIPC is selected as the origin node (orange node), from which a main path is selected (red path). For cells along this main path, we computed the orientation score for each gene, yielding an average score of 1.138. This positive average orientation score indicates that the actual developmental direction on the main path should be from nIPC to Granule mature. Consequently, the velocity vector of cells in the Granule immature cluster is compelled to point toward cells in the Granule mature cluster, thereby eliminating the reversed velocity flow mentioned earlier.

To show the details of orientation score calculation, the unspliced counts $\tilde{u}$ and spliced counts $\tilde{s}$ along the main path of a typical gene *Grin2b*, which has been reported to play a significant role in neurogenesis[31], together with the fitted linear trees, are shown in Fig. 3c. *Grin2b* demonstrates a clear induction pattern ($\tilde{u} > \tilde{s}$) during development from Neuroblast to Granule immature, and a repression pattern ($\tilde{u} < \tilde{s}$) during development from Granule immature to Granule mature (Methods). The orientation score for this gene is 13.20, which provides solid evidence for us to infer the correct main path direction.

TIVelo's strategy to infer the direction on the cluster graph not only produces velocity stream plots consistent with the expected cell development, but also achieves better RNA velocity estimation for individual genes. For some genes in dentate gyrus, cells in Granule immature tend to have larger $u$ and $s$ expression values than those in Granule mature, such as *Map1b*, *Cplx2*, and *Ak5* (Fig. 3d, top row). When fitted by ODE-based methods like scVelo, velocity vectors from Granule mature to Granule immature are observed, leading to reversed velocity stream compared to the expected developmental trajectory, which is shown in Fig. 3d (top row) and Supplementary Fig. 1c. Conversely, the velocity vectors in those three genes inferred from TIVelo correctly point to cells in Granule mature (Fig. 3d, bottom row). This further underscores the advantage of RNA velocity estimation of individual cells supervised by cluster-level direction.

Finally, TIVelo can provide DTI analysis at the cluster level. Specifically, a directed edge is drawn from each node in the cluster graph and pointing toward its child nodes (Methods). Figure 3e presents the DTI analysis result of the dentate gyrus, clearly illustrating the developmental relationship of cell clusters, including island-like clusters.

Similar results from several datasets where the orientation score on the main path is positive, are shown in supplementary information, including pancreatic endocrinogenesis (Supplementary Fig. 2), mouse gastrulation (erythroid) (Supplementary Fig. 3), mouse hindbrain (Oligo) (Supplementary Fig. 4), dentate gyrus development 2 (Supplementary Fig. 5) and mouse embryonic fibroblast reprogramming (Supplementary Fig. 6).

## TIVelo identifies the expected origin node of the cluster graph

To further assess the performance of TIVelo, particularly in scenarios where ODE models may fail, we applied it to the intestinal organoid dataset. This dataset has two developmental lineages, namely the secretory lineage (mouse intestinal stem cells (Stem cells) to Paneth cells) and the enterocyte lineage (Stem cells to Enterocytes)[32] (Fig. 4a).

A comparison of the velocity stream plots generated by TIVelo, UniTVelo, and veloVI is shown in Fig. 4a. Only the velocity stream inferred by TIVelo accurately reflects the bifurcation from Stem cells into the secretory lineage and the enterocyte lineage. As a comparison, velocity plots inferred from UniTVelo and veloVI display reversed velocity flow, originating from Paneth cells and Enterocytes and moving toward Stem cells. Of note, the result of UniTVelo is different from its original report, probably due to the use of (default) random initialization here, while the original study used a warm initialization. The velocity stream plots produced by scVelo (stochastic mode),

scVelo (dynamical mode), DeepVelo and cellDancer are shown in Supplementary Fig. 1d: while they can generally predict the development in the secretory lineage, the development in the enterocyte lineage is not accurately estimated except for DeepVelo.

The better performance of TIVelo comes from its inference of direction on the cluster graph of intestinal organoid, which is shown in Fig. 4b. The origin node initially selected by TIVelo is Enterocytes, an end cell cluster in the data. The main path is selected as the path from Enterocytes cells to Stem cells, which has the opposite direction for the enterocyte lineage. The average orientation score computed on the main path is −0.650, suggesting that the direction on the main path should be reversed, and cluster Stem cells should be set as the new origin node. This enables TIVelo to correctly infer the direction along the main path. *Muc13*, which has been reported to be involved in the maintenance of gastrointestinal epithelium[33], provides explicit evidence for us to infer the direction on the main path, as is illustrated in Fig. 4c. In the early stage of transition from Enterocytes to TA cells, the expression levels of $\tilde{u}$ and $\tilde{s}$ are nearly identical as both are increasing. However, in the later stage from TA cells to Stem cells, where both $\tilde{u}$ and $\tilde{s}$ are decreasing, $\tilde{s}$ is overtaken by $\tilde{u}$. Consequently, the orientation score of *Muc13* is computed as −13.74, which strongly supports the reversal of the direction on the main path.

Moving from the velocity stream estimation to the velocity inference for individual genes, TIVelo can accurately infer the velocity direction on the enterocyte lineage. We illustrate this by showing the $u$-$s$ scatter plot of several genes in this data, along with some sampled velocity vectors inferred from TIVelo (Fig. 4d). These genes, including *Ndrg1*, *Dhrs1*, *Gramd3*, *Cdr2*, and *Slc7a9*, exhibit differential $u$-$s$ expression across cells in the enterocyte lineage. As shown in Fig. 4d, for each gene, the velocity direction of each cell generally points to Enterocytes from Stem cells, thereby accurately indicating the developmental trajectory of the enterocyte lineage.

To demonstrate how TIVelo can provide insights for biological studies, we carried out several downstream analyses based on TIVelo's inferred velocities, including fate probability visualization, lineage-specific driver gene identification, identification of macrostates with functional insights, and kinetic rates inference. The details are discussed in Supplementary Note 2.

Similar results from several datasets where the orientation score on the main path is negative and the origin node is corrected, are shown in supplementary information, including mouse retina development (Supplementary Fig. 7), scNT-seq neuron KCl stimulation (Supplementary Fig. 8) and mouse hindbrain (GABA, Glial) (Supplementary Fig. 9).

## TIVelo achieves the highest sign accuracy on cell cycle datasets

We next investigated the performance of TIVelo on two cell-cycle datasets of fluorescent ubiquitination-based cell-cycle indicator (FUCCI) RPE1 and U2OS cells[32,34]. These datasets were employed by veloVI[14] to evaluate the RNA velocity estimation performance. One advantage of comparing different methods on these two datasets is that they provide additional validation of the inferred velocity/pseudotime through a cell-cycle score: the inferred pseudotime should be positively correlated to the cell-cycle score, and the inferred RNA velocity of cells with a lower cell-cycle score should ideally point toward cells with a higher cell-cycle score (Fig. 5a).

Since there is no cell type annotation provided with these two datasets, we first employed the clustering algorithm leiden[35] to obtain the cell clusters (Methods). Then, RNA velocity is estimated for the two datasets following the TIVelo pipeline. Figure 5b shows the comparison of RNA velocity stream plots of the RPE1-FUCCI dataset from TIVelo, veloVI and scVelo (stochastic mode). TIVelo and veloVI accurately predict the velocity direction following the increase of cell-cycle score, while scVelo (stochastic mode) incorrectly identifies an intermediate orange cluster as the end cluster, leading to reversed velocity flow from the blue cluster to the orange cluster. The cluster graph after

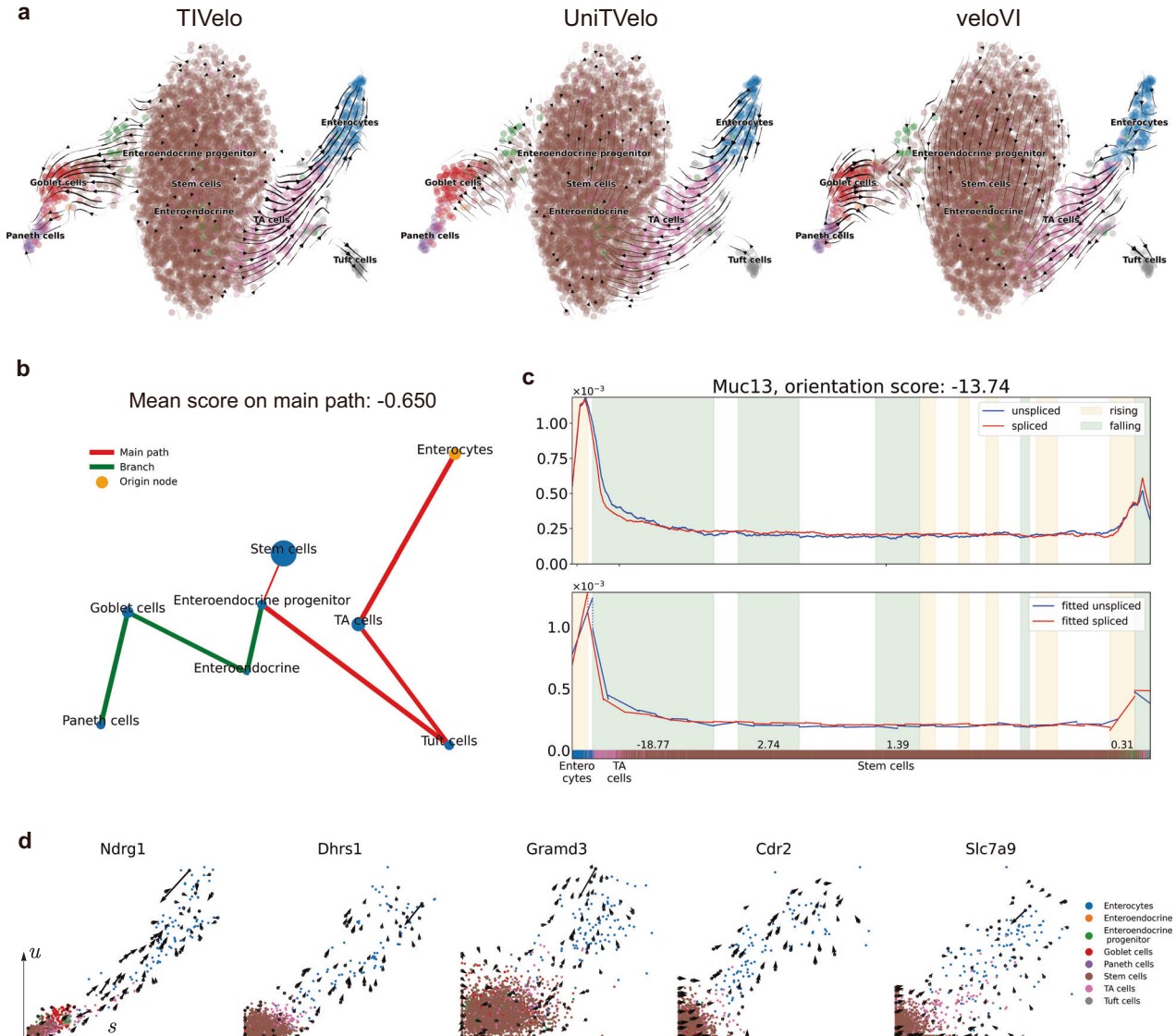

**Fig. 4 | TIVelo's delineation of two lineages in intestinal organoid.**
**a** Comparative velocity stream plots produced by TIVelo, UniTVelo, and veloVI.
**b** The cluster graph after graph pruning and main path selection. The end cluster Enterocytes is selected as the origin node. **c** The $\bar{u}$-$\bar{s}$ variation along the main path and the linear tree model constructed for gene *Muc13*, with the orientation score of primary rising/falling section annotated below. The colorbar indicates the cell type, which is identical to the annotation in (**a**). **d** Velocity fitting for five different genes from TIVelo. For each gene, the velocity direction of each cell generally points to Enterocytes from Stem cells.

graph pruning and main path selection, and the velocity stream plots from additional methods, are shown in Supplementary Fig. 10.

It is worth noting that in Fig. 5b, it seems that the inferred velocity streams of TIVelo in cluster 1 lack coherence in their orientation. However, this directional pattern is not indicative of an inconsistency in TIVelo's velocity estimation but rather reflects the underlying biological dynamics. The details are discussed in Supplementary Note 3.

To quantify the performance of different methods, we calculated the velocity sign accuracy proposed by veloVI for each cell-cycle position in both datasets (Methods). Figure 5c, d illustrate the comparison of sign accuracy calculated for both datasets using TIVelo and six other benchmarking methods. In both datasets, the mean sign accuracy of velocity inferred from TIVelo outperformed all six other methods.

**TIVelo infers RNA velocity for single-cell multi-omics data**
Single-cell multi-omics datasets with both the modalities of RNA and ATAC are available[36]. Although TIVelo only uses the modality of RNA, it can achieve comparable or better performance compared to methods that use both modalities. More specifically, we compared the performance of TIVelo and MultiVelo[20] on four single-cell multi-omics datasets that included both modalities of RNA and ATAC from four different tissues (embryonic mouse brain, SHARE-seq mouse skin, hematopoietic stem and progenitor cells (HSPCs) and developing human brain). While MultiVelo constructs an ODE of chromatin accessibility level $c$, unspliced RNA $u$ and spliced RNA $s$ for each gene, TIVelo only utilizes unspliced RNA $u$ and spliced RNA $s$ for RNA velocity estimation.

We first focused on the RNA velocity comparison on the HSPCs dataset, as shown in Fig. 6a. The RNA velocity stream plot produced by TIVelo accurately depicts the differentiation from hematopoietic stem cells (HSC) into three lineages: myeloid lineage, erythroid lineage and platelet lineage (Fig. 6c, top row). In contrast, the RNA velocity inferred by MultiVelo does not reflect the actual developmental trajectory of the erythroid lineage, and scVelo (dynamical mode) incorrectly identifies progenitors and megakaryocyte (Prog MK) as the root cluster.

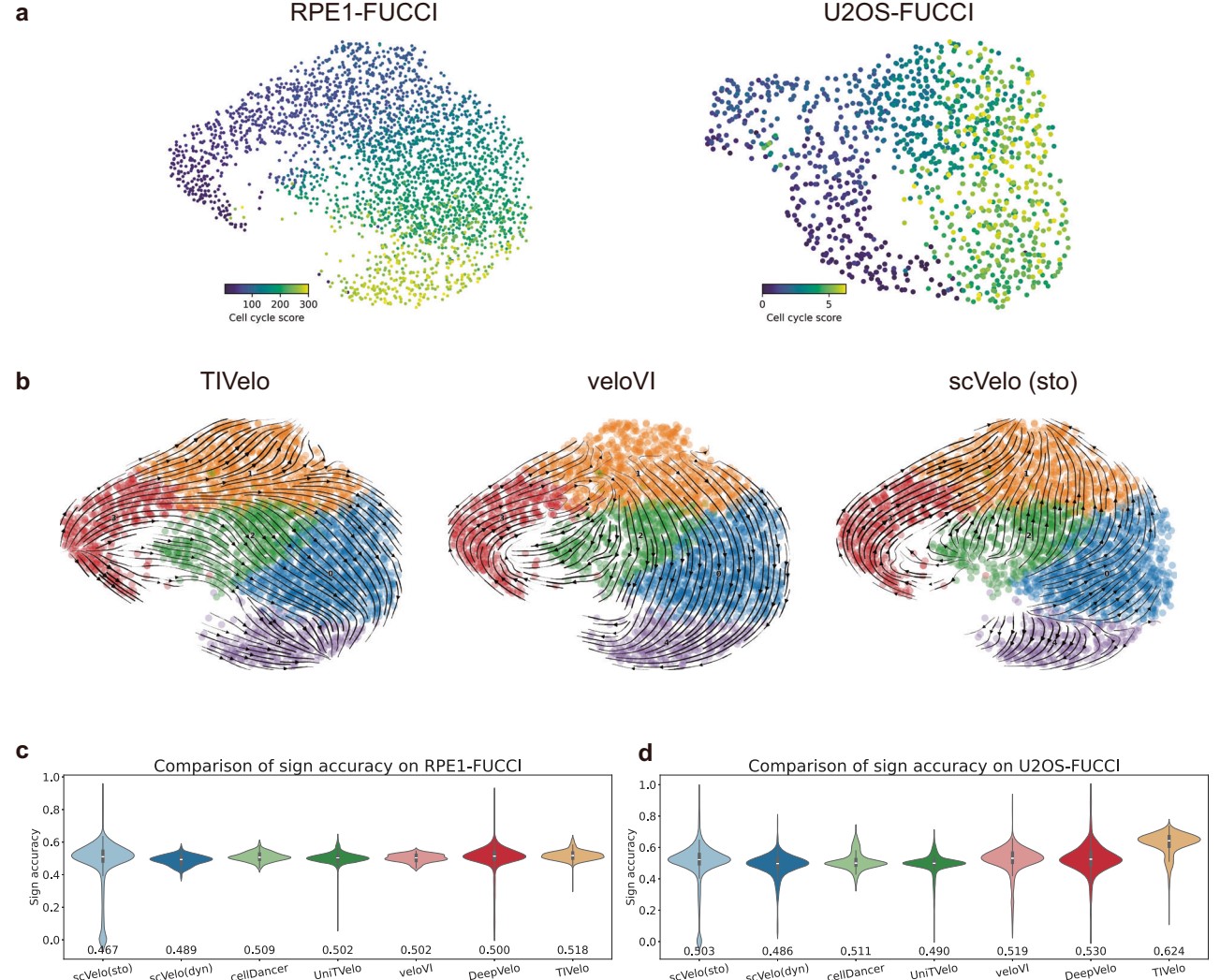

**Fig. 5 | Evaluation of TIVelo's performance on RPE1-FUCCI and U2OS-FUCCI with cell-cycle score. a** UMAP plots for RPE1-FUCCI and U2OS-FUCCI, colored by cell-cycle score of each cell. **b** Comparative velocity stream plots produced by TIVelo, veloVI and scVelo (stochastic mode) on RPE1-FUCCI. **c** Violin plots showing the comparison of sign accuracy across all positions ($n = 290$ positions) in RPE1-FUCCI, produced by TIVelo and six other benchmarking methods. The mean sign accuracy of each method is annotated below. In the box plot inside each violin, the lower bound, center and upper bound of the box plot stand for the first quartile ($Q_1$), median and the third quartile ($Q_3$) of the sign accuracy, respectively. The lower

whisker and the upper whisker stand for $Q_1 - 1.5 \times IQR$ and $Q_3 + 1.5 \times IQR$, where IQR = $Q_3 - Q_1$. **d** Violin plots showing the comparison of sign accuracy across all positions ($n = 996$ positions) in U2OS-FUCCI, produced by TIVelo and six other benchmarking methods. The mean sign accuracy of each method is annotated below. In the box plot inside each violin, the lower bound, center and upper bound of the box plot stand for the first quartile ($Q_1$), median and the third quartile ($Q_3$) of the sign accuracy, respectively. The lower whisker and the upper whisker stand for $Q_1 - 1.5 \times IQR$ and $Q_3 + 1.5 \times IQR$, where IQR = $Q_3 - Q_1$.

The enhanced performance of TIVelo can be better understood through the visualization of the HSPCs cluster graph (Fig. 6b). The origin node initially selected by TIVelo is the cell cluster Platelet, which is an end cluster in the HSPCs dataset. The main path selected by TIVelo is a path from Platelet to the expected root cluster HSC. The mean orientation score on the main path is −7.773, indicating that the direction should be reversed, and the origin node is reset as HSC. The direction inference at the cluster level facilitates the RNA velocity estimation for individual genes. Figure 6c shows the velocity vectors of three genes inferred from TIVelo in the bottom half, namely *AZU1*, *KEL*, and *VWF*, corresponding to myeloid lineage, erythroid lineage and platelet lineage, respectively, which are displayed in the upper half of the figure. For each gene, *u* and *s* are expressed more differentially across cells in its corresponding lineage than cells in other lineages. The velocity vectors of three genes from TIVelo accurately reflect the developmental trajectory in their corresponding lineage.

Finally, we compared TIVelo and MultiVelo across the four datasets employed by MultiVelo. We measure their performance by CBDir (UMAP space) and CBDir (Gene space), which are two metrics testing if the estimated velocity follows the expected developmental trajectory (Methods). TIVelo achieved much higher CBDir (UMAP space) scores for the mouse brain and HSPCs datasets, and comparable CBDir (UMAP space) scores for the mouse skin and human brain datasets (Fig. 6d). Furthermore, TIVelo consistently outperformed MultiVelo on all four datasets in terms of CBDir (Gene space) (Fig. 6e).

The cluster graphs used in TIVelo and the velocity stream plots comparisons between TIVelo and MultiVelo from another three single-cell multi-omics (RNA + ATAC) datasets, are shown in supplementary information, including embryonic mouse brain (Supplementary Fig. 11a), SHARE-seq mouse skin (Supplementary Fig. 11b) and developing human brain (Supplementary Fig. 11c).

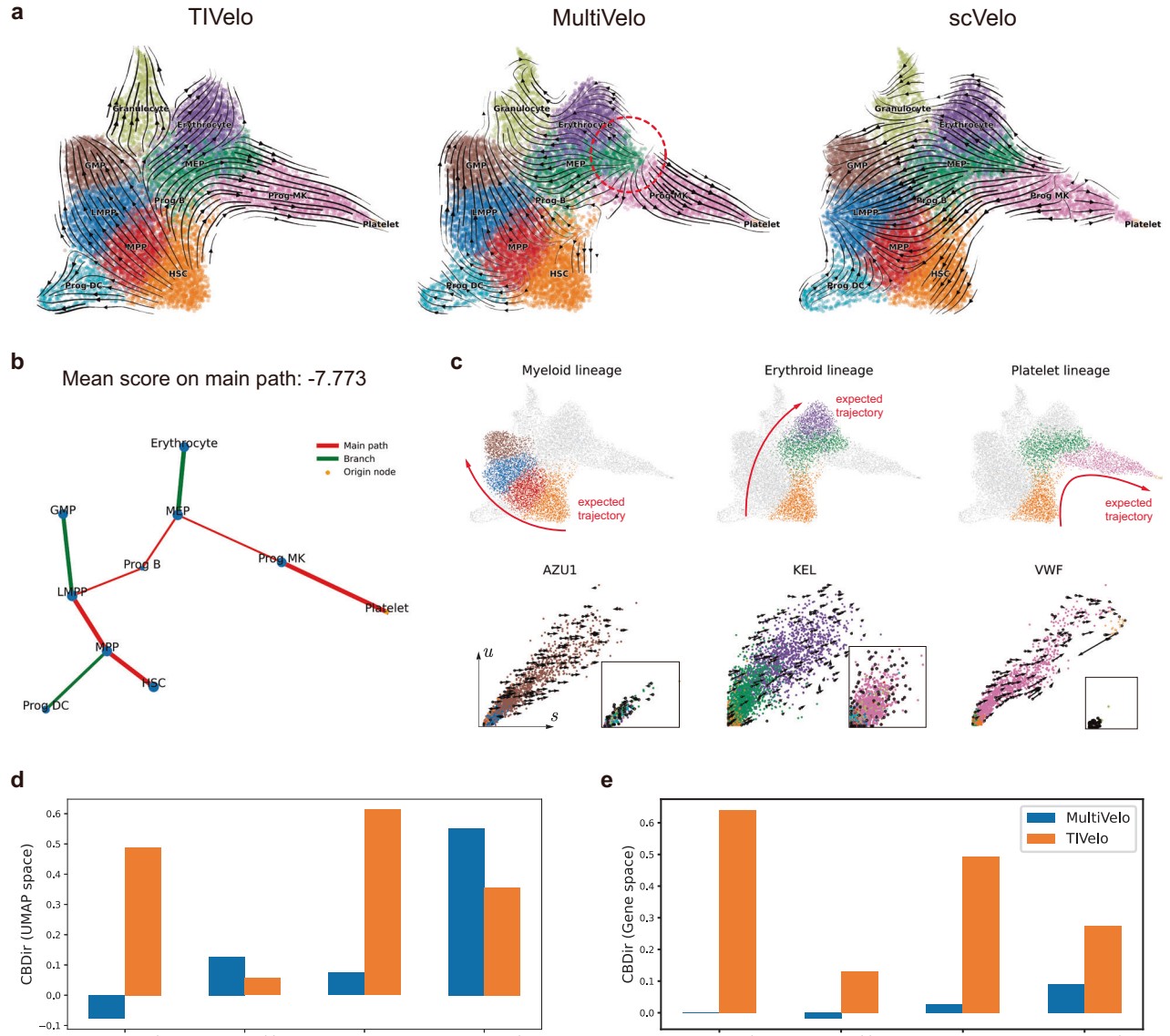

**Fig. 6 | Evaluation of TIVelo's performance for single-cell multi-omics data compared with MultiVelo. a** Comparative velocity stream plots of the HSPCs dataset generated by TIVelo, MultiVelo, and scVelo (dynamical mode). **b** The cluster graph after graph pruning and main path selection. The end cluster, Platelet, is selected as the origin node. The mean orientation score along the main path is −7.773. **c** Top: The depiction of three distinct lineages in the HSPCs dataset. Bottom: The *u-s* scatter plots and fitted velocity from TIVelo of three genes *AZU1*, *KEL,* and

*VWF*. For each gene, *u* and *s* are expressed more differentially across cells in its above lineage than cells in other lineages. For each gene, cells not in its corresponding lineage are shown in the panel at the bottom right. The velocity vectors of three genes from TIVelo accurately reflect the developmental trajectory in their corresponding lineage. **d** A comparison of the cross-boundary direction correctness (CBDir) (UMAP space) score across four datasets. **e** A comparison of the cross-boundary direction correctness (CBDir) (Gene space) score across four datasets.

## Quantitative comparison with benchmarking methods on ten real datasets

To thoroughly assess TIVelo's performance, we did a comparison on ten datasets commonly used by existing RNA velocity inference methods, as shown in Fig. 7. The comparison includes TIVelo and six other benchmarking methods, namely scVelo (stochastic mode), scVelo (dynamical mode), UniTVelo, cellDancer, veloVI and Deep-Velo. The metrics to measure their performance include cross-boundary direction correctness (CBDir) (Gene space), transition cosine similarities (TransCosine), and velocity coherence (VeloCoh)[14]. CBDir (Gene space) and TransCosine assess if the estimated velocity vectors follow the expected developmental trajectory, while VeloCoh tests if the inferred velocity vectors are consistent with the differences of spliced counts between the expected future state and current state (Methods). As is illustrated in

Fig. 7a, b the CBDir (Gene space) and TransCosine scores of TIVelo demonstrate superior or comparable performance to those of other methods across all ten datasets. The VeloCoh score of TIVelo is the highest in five datasets, while in the pancreas, dentate gyrus, mouse gastrulation (erythroid) and dentate gyrus 2, TIVelo achieves a score comparable to veloVI or DeepVelo (Fig. 7c). In summary, TIVelo outperforms or matches the benchmarking methods, underlining its superior performance in RNA velocity inference.

## Discussion

In this paper, we present TIVelo for RNA velocity estimation. TIVelo is based on TI at the cluster level, by inferring velocity direction on the cluster graph and supervising RNA velocity estimation for individual cells. TIVelo stands out from current RNA velocity inference methods due to several innovative features.

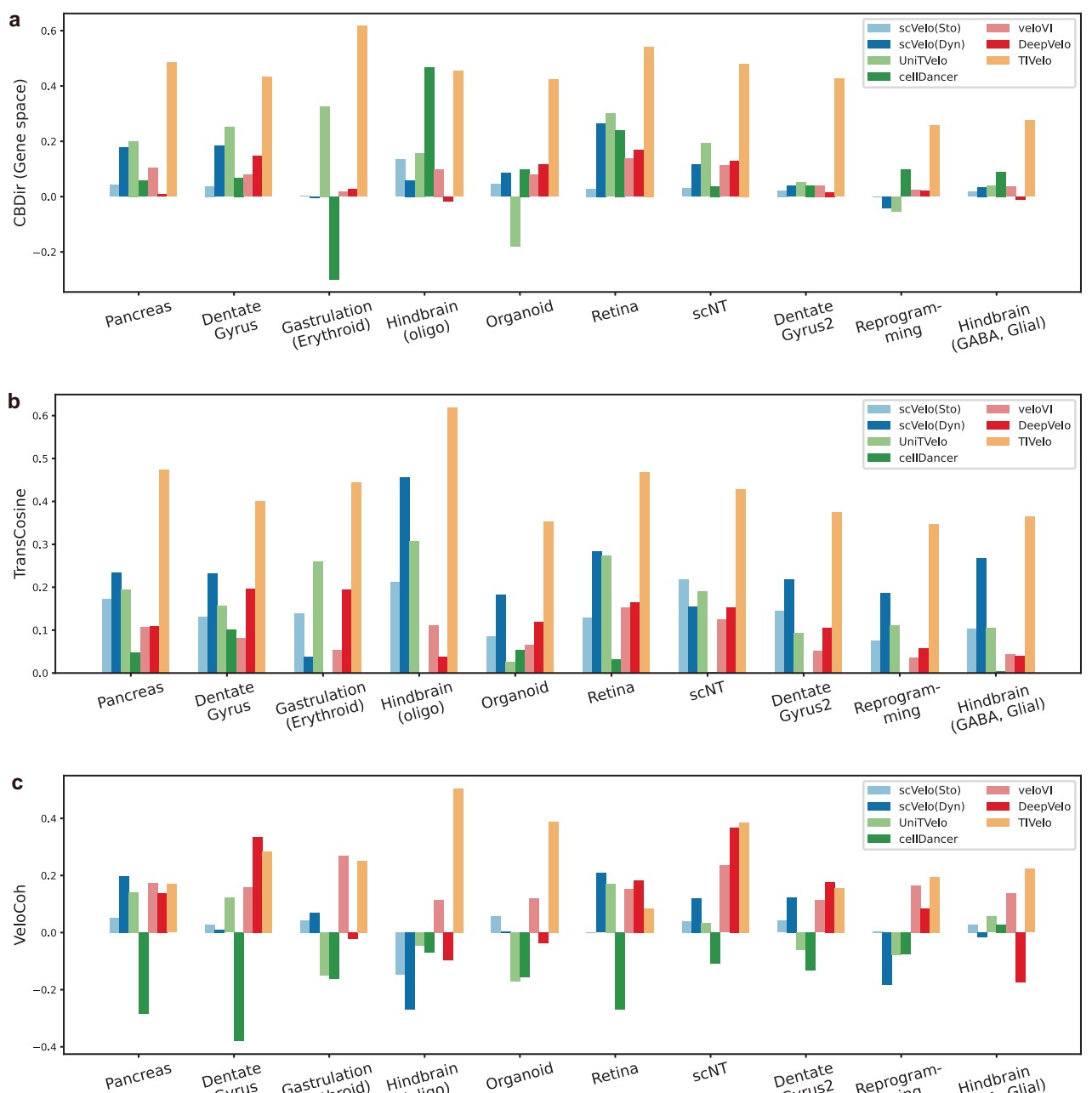

**Fig. 7 | Quantitative benchmarking across ten real datasets. a** Comparison of TIVelo and six benchmarking methods by cross-boundary direction correctness (CBDir) (Gene space) score across ten datasets. **b** Comparison of TIVelo and six benchmarking methods by transition cosine similarities (TransCosine) across ten datasets. **c** Comparison of TIVelo and six benchmarking methods by velocity coherence (VeloCoh) across ten datasets.

Firstly, TIVelo employs a linear tree model to determine the direction of the main path of the cluster graph. This approach allows for the comprehensive mining of signals embedded in gene expression profiles. By leveraging the intrinsic property of the unspliced-spliced relationship, TIVelo effectively mitigates the impact of genes with varying ODE rate parameters during development, enabling more reliable velocity estimation.

Secondly, TIVelo infers the RNA velocity vector of individual cells by referring to the direction on the cluster graph. This ensures consistency and avoids potential contradictions that may arise when aggregating velocity from independently fitted genes. By considering the direction of the cellular trajectory rather than relying solely on

individual gene expression patterns, TIVelo enhances the reliability of velocity estimation.

Thirdly, TIVelo only requires unspliced and spliced RNA for RNA velocity inference, eliminating the need for additional prior information or other biological modalities. This simplifies the workflow and reduces the burden of data acquisition and preprocessing, making TIVelo more accessible and user-friendly.

Despite its advantages, TIVelo does have some limitations that should be considered and further improved in the future. First, TIVelo may fail to accurately estimate velocity for large datasets with complex cluster graph topologies. As TIVelo's approach relies on inferring the direction on the cluster level graph, the presence of complicated

connections and branching patterns within the graph can pose challenges. In such cases, it may be beneficial to reapply the clustering algorithm using a lower resolution, resulting in larger cell clusters and a simplified structure of the corresponding cluster graph.

Second, it is worth noting that the velocity vectors produced by TIVelo may not exhibit the same level of smoothness as those generated by ODE-based methods. This can be observed through the velocity inferred from TIVelo for genes in dentate gyrus and organoid datasets (Figs. 3d and 4d). The strategy of constructing dNN for each cell and using the mean expression level in dNN as the future state can introduce noise into the velocity inference (Methods). By using the `fit` function of the RNA velocity inference step, which introduces a regularization term to enhance the velocity consistency, and increasing the hyperparameter $\lambda'$ controlling the weight of the regularization term, this issue can be alleviated (Methods).

## Methods

The dataset should undergo preprocessing prior to the utilization of TIVelo. TIVelo encompasses three main steps: main path selection, orientation inference, and RNA velocity estimation. During the main path selection, the cluster graph for the data is constructed, and a long, primary path within the graph is chosen. In the orientation inference step, we utilize the unspliced RNA $u$ and spliced RNA $s$ counts to infer the direction of the main path, based on the intrinsic characteristics of transcription: the unspliced RNA $u$ should always be expressed earlier than the spliced RNA $s$ and decrease earlier than $s$. This bypasses the need to explicitly specify an ODE model for $u$ and $s$. The assumptions in the ODE models may not be realistic and can be violated in real datasets. Our model-free approach provides robustness in inferring the direction along the main path. This is the pivotal step of TIVelo. In the step of RNA velocity estimation in TIVelo, the velocity vector of each cell will be inferred, adapting to the direction at the cluster level.

### Preprocessing

Datasets are preprocessed through scVelo[9] standard preprocessing pipeline. This pipeline includes filtering genes with at least 20 cells with non-zero unspliced and spliced counts, selecting the top 2000 highly variable genes, log-transforming, and data smoothing through averaging over the 30 nearest neighbors. The preprocessing can be implemented directly through `scvelo.pp.filter_and_normalize` and `scvelo.pp.moments` functions.

### Main path selection

In this step, we first construct a cluster graph for the dataset, followed by graph pruning to simplify the graph topology. Subsequently, we select an origin node in this cluster graph, from which a main path in the graph will be selected.

**Cluster graph construction and graph pruning.** The cluster graph, represented by cell clusters as nodes and connecting weights as edges, is initially constructed for the dataset using PAGA[7]. If cell annotation is included in the dataset, it is utilized as the group key for PAGA. If not, we employ Leiden for clustering with `resolution = 0.6`, and subsequently use the clustering label as the group key.

After the construction of the cluster graph, we perform graph pruning, where the purpose is to filter out small or isolated clusters (nodes) in the cluster graph.

The first step in graph pruning is to select the major nodes with a large number of cells in the graph and drop the other nodes. Suppose we have a dataset with $N$ cells and $C$ clusters, and cluster $i$ has $N_i$ cells where $\sum_i N_i = N$. Then (1) order clusters by cell numbers. Select clusters from large to small until the selected cell number $\sum_{i \in S} N_i > 0.5N$, where $S$ is the set of selected clusters. $S$ should include at least five clusters or all clusters if $C < 5$; (2) if $i'$ is connected to $i \in S$ with weight

$w_{i',i} > 0.6\max_{i \neq i'} w_{i',i}$, then $i'$ should be added to $S$. The graph is denoted as $G$ after this first step of graph pruning.

The second step in graph pruning is to reconnect disjoint sub-graphs in graph $G$. If there is more than one disjoint sub-graph in graph $G$, then for each pair of sub-graphs $(G_1, G_2)$, $G_1, G_2 \subseteq G$, we add back one cluster $i$ not included in graph $G$, which can connect $G_1$ and $G_2$, based on the following rule: let $w_{i1}$ and $w_{i2}$ denote the weights between cluster $i$ and its closest clusters in sub-graphs $G_1$ and $G_2$, respectively; we select cluster $i$ with the largest value in $w_{i1}w_{i2}$. If such a cluster $i$ does not exist, we recommend the user to run the subsequent steps of TIVelo for the subgraphs separately, because this indicates that there may be several independent lineages in the dataset.

**Initial selection of origin node.** Origin node refers to a node in the cluster graph, beginning from which we select the main path (see "Main path selection"). We initially select the origin node from terminal states, which refers to either root clusters or end clusters in the cluster graph. It is crucial to avoid selecting an intermediate cluster as an origin node, as this will affect the performance of inferring the direction on the main path (Supplementary Fig. 12). In the orientation inference step, the origin node can be reset based on the orientation score on the main path (see "Infer orientation").

We first implement the `scvelo.tl.terminal_states` function in the package scVelo to obtain the root and end score for each single cell. After obtaining the root and end score for each cell, the mean root and end scores, $R_i$ and $E_i$, are computed for each cluster $i$ ($1 \leq i \leq C$). Subsequently, we rank the mean root score $R_i$ and end score $E_i$ and get $k_1, \cdots, k_C$ and $k'_1, \cdots, k'_C$ such that $R_{k_1} > \cdots > R_{k_C}$ and $E_{k'_1} > \cdots > E_{k'_C}$, where $k_1, \cdots, k_C$ and $k'_1, \cdots, k'_C$ are two sets of indices for the clusters. The origin node is then selected based on the following procedure:

- For $i = 1$: $C$ select the first $k_i$ with $R_{k_i} > 0.1$ and $E_{k_i} < R_{k_i}$;
- If no $k_i$ is selected then for $i = 1$: $C$ select the first $k'_i$ with $E_{k'_i} > 0.1$ and $R_{k'_i} < E_{k'_i}$;
- If no $k'_i$ is selected then select $k_1$.

The detailed steps for selecting the origin node are presented in the Supplementary Note 4.

**Main path selection.** The main path in a cluster graph refers to a path that begins from the origin node and involves as many cells as possible. Let $o$ denote the origin node. We define down($i$) as the downstream nodes for node $i$, which represent nodes that are directly connected to $i$, excluding nodes in the path between node $o$ and $i$; and define successor($i$) as all nodes being successors to node $i$, which are nodes directly or indirectly connected to $i$, except for all nodes in the path between node $o$ and $i$ (Supplementary Fig. 13). The nodes in the main path are then selected sequentially as follows:

- Choose $i \in$ down($o$) to let $N_i + \sum_{i' \in \text{successor}(i)} N_{i'}$ attain its maximum. Then add $i$ to the main path;
- Choose $i' \in$ down($i$) to let $N_{i'} + \sum_{i'' \in \text{successor}(i')} N_{i''}$ attain its maximum. Then add $i'$ to the main path;
- Repeat this process until no node can be selected.

### Infer orientation

After the main path is selected, we proceed to infer the orientation/direction along the main path. We first assign a pseudotime to cells along the main path, beginning from the origin node. Consequently, for each gene, the corresponding unspliced RNA $u$ and spliced RNA $s$ can be ordered by this pseudotime to form two time series. Then we fit a linear tree model to capture the rising/falling time sections of each time series. Based on the linear tree model, an orientation score can be calculated to facilitate the inference of the direction along the main path. Finally, the orientation score from each gene will be aggregated, according to which we select a new origin node (root node) for the cluster graph.

**Linear tree construction for genes.** We set one root cell in the origin node as time 0, and infer a pseudotime for each cell along the main path, by using diffusion pseudotime[37]. The root cell with time 0 is chosen by the default way indicated by diffusion pseudotime: `adata.uns['iroot'] = np.flatnonzero(adata.obs['cell_types'] =='Origin')[0]`. Specifically, $t_1, \cdots, t_N$ are corresponding pseudotime for $N$ cells along the main path, with $t_1 < \cdots < t_N$. For each gene $g$, we reorder the $u$-$s$ expression vector $\boldsymbol{u}_g, \boldsymbol{s}_g \in \mathbb{R}^N$ according to the pseudotime of the cells, forming the time series of $u$ and $s$: $\tilde{\boldsymbol{u}}_g = (\tilde{u}_{1,g}, \cdots, \tilde{u}_{N,g})$ and $\tilde{\boldsymbol{s}}_g = (\tilde{s}_{1,g}, \cdots, \tilde{s}_{N,g})$. To alleviate the issue of noise, $\boldsymbol{u}_g$ and $\boldsymbol{s}_g$ will be smoothed by a 1-$d$ convolution with a kernel of $(\frac{1}{K}, \cdots, \frac{1}{K}) \in \mathbb{R}^K$, where $K$ is the kernel size and is set to be 100 by default.

Now we would like to see if the current cell pseudotime $t_1, \cdots, t_N$ is correct or should be reversed. To do this, we utilize an intrinsic property of $u$ and $s$, that is, $u$ should be expressed earlier than $s$ during transcription induction, and decrease earlier than $s$ during transcription repression, since $u$ is the precursor of $s$. To use this property, it is crucial to find the time section in which both $\tilde{\boldsymbol{u}}_g$ and $\tilde{\boldsymbol{s}}_g$ are increasing (induction phase) or decreasing (repression phase), which can be achieved by constructing the linear tree model for each time series. More precisely, for $\tilde{\boldsymbol{u}}_g$ and $\tilde{\boldsymbol{s}}_g$, the unspliced and spliced RNA for each gene $g$, we

- Fit a linear regression model for $\tilde{\boldsymbol{u}}_g$ and $\tilde{\boldsymbol{s}}_g$ against cell index $x \in [1, N]$ ordered by the pseudotime of the cells;
- Perform grid searching for a split point $x_s \in [1, N]$, so that when we fit a piece-wise linear regression model on both sides of the split point, i.e., on $[1, x_s]$ and $(x_s, N]$, the total mean squared error (MSE) is the lowest;
- Split the data into two parts $[1, x_s]$ and $(x_s, N]$, and for each part repeat the above steps until convergence. The convergence is defined by two conditions: either $MSE^{(r)} - MSE^{(r+1)} < 100$, where $MSE^{(r)}$ is the total MSE of iteration $r$, or the interval between any two split points becomes shorter than 10.

**Orientation score calculation.** Based on the linear tree constructed for each gene, an orientation score can be calculated, indicating if the current direction (determined by cell pseudotime) is correct or not. More specifically, for a gene $g$ the score is calculated by

$$S_g = \sum_{\tau \in \mathcal{T}_g} (I_{(k_{u,\tau}^{(g)} > 0, k_{s,\tau}^{(g)} > 0)} - I_{(k_{u,\tau}^{(g)} < 0, k_{s,\tau}^{(g)} < 0)}) \cdot \left( \sum_{n \in \tau} d_{n,g} \right) \cdot l_\tau, \quad (1)$$

where $\mathcal{T}_g$ is the collection of rising time sections (in which both $\tilde{\boldsymbol{u}}_g$ and $\tilde{\boldsymbol{s}}_g$ are increasing) or falling time sections (in which both $\tilde{\boldsymbol{u}}_g$ and $\tilde{\boldsymbol{s}}_g$ are decreasing), $k_{u,\tau}^{(g)}$ and $k_{s,\tau}^{(g)}$ are the slopes of the fitted linear tree for $\tilde{\boldsymbol{u}}_g$ and $\tilde{\boldsymbol{s}}_g$ in section $\tau$, $d_{n,g} = \tilde{u}_{n,g} - \tilde{s}_{n,g}$, and $l_\tau$ is the length of the section $\tau$. The rationale behind this score is as follows. During transcription induction, both $u$ and $s$ are increasing, and the condition $I_{(k_{u,\tau}^{(g)} > 0, k_{s,\tau}^{(g)} > 0)} = 1$ should hold. If $d_{n,g} > 0$, it indicates that $u$ is expressed earlier than $s$. Conversely, during transcription repression, both $u$ and $s$ are decreasing, and $I_{(k_{u,\tau}^{(g)} < 0, k_{s,\tau}^{(g)} < 0)} = 1$ should hold. If $d_{n,g} < 0$, it indicates that $u$ decreases earlier than $s$. In both cases the score will be positive, which supports that the given direction (cell pseudotime) is correct. Otherwise the given direction is wrong and should be reversed. The factor $l_\tau$ ensures that longer rising/falling time sections, which possess a stronger signal, carry a larger weight when calculating the score.

Typically, in scRNA-seq datasets the magnitude of $s$ is much larger than that of $u$[9]. In this case, when we calculate the orientation score, the difference $d_{n,g} = \tilde{u}_{n,g} - \tilde{s}_{n,g}$ will always be negative. To address this issue, $\tilde{\boldsymbol{u}}_g$ and $\tilde{\boldsymbol{s}}_g$ should be normalized before constructing the linear tree. In this normalization, $\tilde{\boldsymbol{u}}_g$ and $\tilde{\boldsymbol{s}}_g$ are rescaled to sum to 1 for each gene $g$, i.e., $\tilde{u}_{n,g} \leftarrow \tilde{u}_{n,g}/\sum_n \tilde{u}_{n,g}$ and $\tilde{s}_{n,g} \leftarrow \tilde{s}_{n,g}/\sum_n \tilde{s}_{n,g}$. This is to facilitate the comparison between the relative expression levels of $u$ and $s$. We further evaluated an alternative normalization approach, in which $u$ and $s$ are rescaled by the maximum values of $u$ and $s$ for each gene. The details are discussed in Supplementary Note 5.

To validate the robustness of TIVelo in practical applications, we carried out several experiments, including the evaluation of TIVelo's performance on the perturbed main path, and on the main path with reversed direction. Furthermore, we can enhance TIVelo's robustness with respect to the origin node through the refinement of the origin node selection strategy. The details are discussed in Supplementary Notes 6 and 7.

**Infer new origin node (root node).** To accurately infer the new origin node (root node), we only aggregate orientation scores from velocity genes[9,10] with high $u$-$s$ expression data quality. Briefly speaking, velocity gene refers to one with a positive coefficient ($\gamma > 0.01$) between $u$ and $s$, and a positive coefficient of determination ($R^2 > 0.01$). In addition, the ratio of standard deviations of $u$ and $s$ should be moderate ($0.03 < \sigma_{\text{ratio}} < 3$). For cells along the main path, the orientation score can be calculated for each velocity gene $g$, i.e., $S_1, \cdots, S_G$, where $G$ is the number of velocity genes in the data.

Usually there exist some branches in the cluster graph, which refer to all paths except the main path (Fig. 3b, green paths). We infer the new origin node by two cases, determined by the relationship between the main path and the branches.

In the first case, both endpoints of the main path are not connected to the branches (Supplementary Fig. 14, left). In this case, once $\frac{1}{G}\sum_g S_g < 0$, we consider the direction on the main path to be incorrect, and a new origin node should be set as the other end of the main path. Otherwise when $\frac{1}{G}\sum_g S_g \geq 0$, the origin node remains unchanged.

In the second case, one endpoint $i$ of one branch $b$ is also an endpoint of the main path (Supplementary Fig. 14, right). If this happens, we check the direction on branch $b$ similar to the way for the main path. Specifically, we assign a pseudotime to cells along branch $b$ by diffusion pseudotime. Time series $\tilde{\boldsymbol{u}}_g^b$ and $\tilde{\boldsymbol{s}}_g^b$ of gene $g$ are extracted for calculating orientation score $S_g^b$. The average score $\frac{1}{G}\sum_g S_g^b$ is used to see if the direction of $b$ should be corrected. If the corrected direction on the main path is aligned with the corrected direction on branch $b$, the new origin node is set as the other endpoint $j$ of branch $b$; otherwise the new origin node is set as node $i$ (Supplementary Fig. 14, right).

### RNA velocity estimation

After the new origin node (root node) of the cluster graph is determined, we infer the RNA velocity for each cell. The velocity vector of each cell should follow the direction of the cluster graph. We assign a level to each node in the cluster graph and identify the child nodes of each node based on their levels. In addition, a new pseudotime $t'$ is assigned to each cell, beginning from the new origin node. RNA velocity of each cell in node $i$ is inferred according to its pseudotime and child nodes.

### Level and child node assignment

Firstly, we assign a level for each node in the original cluster graph without graph pruning. We rerun PAGA on the whole dataset and obtain connectivities weight $w_{i,j}$ for each pair of clusters $(i, j)$. The levels for the nodes are assigned as follows:

- The level of new origin node $o$ is set to be 0;
- Assume that node $i$ has level $k$. For any node $i'$ if (1) $w_{i,i'} > z_1$; (2) there does not exist node $j$ such that $w_{i,i'} < z_2 w_{i,j} w_{j,i'}$, where $z_1$ and $z_2$ are prespecified thresholds (the default values for $z_1$ and $z_2$ are 0.1 and 1, respectively), then the level of $i'$ is assigned as $k+1$.
- Continue the above steps until every nodes have a level.

Secondly, based on the level assigned for each node, we determine the child nodes for each node in the graph as follows:

- For every node $i$, node $i'$ is the child node of $i$ if (1) $w_{i,i'} > z_1$; (2) level($i'$)-level($i$) = 1;
- If node $j$ is not the child node of any node, it will be the child node of $j' = \text{argmax}_{j'} w_{j,j'}$, i.e., each node should have at least one parent node, except for the origin node;
- If node $j$ has two or more parent nodes, e.g., $i_1, i_2 \cdots i_k$ then the parent node of $j$ will be $i_k = \text{argmax}_k w_{j,i_k}$, i.e., each node is not allowed to have more than one parent node.

After the child nodes are determined, we can draw a directed edge from any node $i$ to its child node $i'$. This gives us the result of DTI for the cell clusters in the dataset.

**RNA velocity inference.** To infer RNA velocity of each cell, we create a dNN graph for each cell $n$, which signifies the future state of cell $n$ in its $k$-nearest neighborhood ($k$-NN) (as illustrated in Fig. 1h). We construct a $k$-NN graph for each cell $n$ denoted as $\mathcal{N}_n$, through `scvelo.pp.neighbors` from scVelo package, setting $k=30$ by default. We also reapply diffusion pseudotime for the entire dataset (setting one root cell in the new origin node as time 0 by the default way as before: `adata.uns['iroot'] = np.flatnonzero(adata.obs['cell_types']=='Origin')[0]`) to assign a pseudotime $t'_n$ for each cell. Then cell $n'$ is considered to be in the dNN of cell $n$ if (1) $n' \in \mathcal{N}_n$; (2) $t'_n < t'_{n'}$; (3) $i' \in \text{child}(i)$ or $i = i'$, where clusters $i$ and $i'$ are the clusters that cells $n$ and $n'$ belong to, respectively.

The velocity vector for cell $n$ should point toward the mean expression level of cells within dNN of cell $n$. That is equivalent to minimizing the following objective $L_g$ for each gene $g$:

$$
\begin{aligned}
L_{u,g} &= \sum_{n=1}^{N} \| u_{n,g} + \tilde{v}_{n,g}^{(u)} - \frac{1}{N_n} \sum_{n' \in dnn(n)} u_{n',g} \|^2, \\
L_{s,g} &= \sum_{n=1}^{N} \| s_{n,g} + \tilde{v}_{n,g}^{(s)} - \frac{1}{N_n} \sum_{n' \in dnn(n)} s_{n',g} \|^2, \\
L_g &= L_{u,g} + \lambda L_{s,g},
\end{aligned}
\tag{2}
$$

where $dnn(n)$ is the dNN of cell $n$, $N_n$ is the number of cells in $dnn(n)$, and $\lambda$ is a hyperparameter controlling the weight of loss from $u$ and $s$, which is set as 1 by default. $\tilde{v}_{n,g}^{(u)}$ and $\tilde{v}_{n,g}^{(s)}$ are inferred velocities for $u$ and $s$ of cell $n$ and gene $g$.

One straightforward way to infer $\tilde{v}_{n,g}^{(u)}$ and $\tilde{v}_{n,g}^{(s)}$ is to let $L_g$ in Equation (2) be 0 directly, to obtain:

$$
\tilde{v}_{n,g}^{(u)} = \frac{1}{N_n} \sum_{n' \in dnn(n)} u_{n',g} - u_{n,g},
\tag{3}
$$

and

$$
\tilde{v}_{n,g}^{(s)} = \frac{1}{N_n} \sum_{n' \in dnn(n)} s_{n',g} - s_{n,g}.
\tag{4}
$$

This is the `simple_fit` function of TIVelo.

Instead of directly calculating $\tilde{v}_{n,g}^{(u)}$ and $\tilde{v}_{n,g}^{(s)}$ by Equation (3) and (4), we can add some regularization to the objective in order to enhance the consistency of the velocity vector of similar cells. Specifically, the regularization term is designed as

$$
L_c = \sum_{n=1}^{N} \sum_{n' \in \mathcal{N}_n \cap \mathcal{C}(n)} \cos(\bar{v}_n, \tilde{v}_{n'}),
\tag{5}
$$

where $\bar{v}_n = (\tilde{v}_{n,g})_g$ and $\bar{v}_{n'} = (\tilde{v}_{n',g})_g$ are RNA velocity vectors of cell $n$ and $n'$, and $\mathcal{C}(n)$ is the cluster cell $n$ belongs to. We update matrices $(\tilde{v}_{n,g})_{n,g}$ to minimize $\sum_g L_g + \lambda' L_c$, where $\lambda'$ is another hyperparameter controlling the weight of the regularization term. This is the `fit` function of TIVelo. This function is used by default with $\lambda = 1$ and $\lambda' = 0.1$.

In addition to the default strategy for selecting the root cell when applying the diffusion pseudotime, we refined this strategy for two datasets: mouse gastrulation (erythroid) and RPE1-FUCCI, for improved velocity inference in their root cluster. The details are discussed in Supplementary Note 8.

**Kinetic rate mode.** Kinetic rate mode of TIVelo enables simultaneous inference of cell-specific kinetic rates ($\alpha$, $\beta$ and $\gamma$) when inferring RNA velocity for each cell. In this mode, instead of updating $\tilde{v}_{n,g}^{(u)}$ and $\tilde{v}_{n,g}^{(s)}$ in Equation (2) directly, we calculate the inferred velocity $\tilde{v}_{n,g}^{(u)}$ and $\tilde{v}_{n,g}^{(s)}$ by cell-specific kinetic rates $\alpha_{n,g}, \beta_{n,g}$, and $\gamma_{n,g}$ as follows:

$$
\begin{aligned}
\tilde{v}_{n,g}^{(u)} &= \alpha_{n,g} - \beta_{n,g} u_{n,g}, \\
\tilde{v}_{n,g}^{(s)} &= \beta_{n,g} u_{n,g} - \gamma_{n,g} s_{n,g}.
\end{aligned}
\tag{6}
$$

Furthermore, we infer the kinetic rates $\alpha_{n,g}, \beta_{n,g}$, and $\gamma_{ng}$ from unspliced and spliced expressions as follows:

$$
(\boldsymbol{\alpha}_n^\top, \boldsymbol{\beta}_n^\top, \boldsymbol{\gamma}_n^\top)^\top = f_\theta([\boldsymbol{u}_n^\top, \boldsymbol{s}_n^\top]^\top),
\tag{7}
$$

where $f_\theta$ is fully connected layers with parameters $\theta$, $\boldsymbol{\alpha}_n, \boldsymbol{\beta}_n$, and $\boldsymbol{\gamma}_n$ are $G$-dimensional kinetic rates for cell $n$ across all $G$ genes, and $\boldsymbol{u}_n$ and $\boldsymbol{s}_n$ are $G$-dimensional unspliced and spliced expressions of cell $n$. This approach provides natural regularization, ensuring cells with similar expression profiles have comparable kinetic rates.

By default, we use a $f_\theta$ with three layers in Equation (7). The input dimensions, (two) hidden dimensions and output dimensions of $f$ are $2G$, (256, 64) and $3G$, respectively. We further calculate the inferred velocities $\tilde{v}_{n,g}^{(u)}$ and $\tilde{v}_{n,g}^{(s)}$ by Equation (6), and minimize the objective $L_g$ in Equation (2) with respect to $\theta$. The model is trained for 300 epochs with a learning rate 0.001.

**Velocity graph and embedding velocity stream plot.** The velocity graph $\boldsymbol{\Pi} = (\pi_{n,n'})$ and transition matrix $\tilde{\boldsymbol{\Pi}} = (\tilde{\pi}_{n,n'})$ for each pair of cells $(n, n')$ can be constructed as follows[9]:

$$
\pi_{n,n'} = \cos(\boldsymbol{s}_{n'} - \boldsymbol{s}_n, \boldsymbol{v}_n),
\tag{8}
$$

and

$$
\tilde{\pi}_{n,n'} = \frac{1}{Z_n} \exp\left(\frac{\pi_{n,n'}}{\sigma_n^2}\right),
\tag{9}
$$

where $\boldsymbol{s}_n$ is the spliced RNA expression of cell $n$, $\boldsymbol{v}_n$ is the inferred velocity vector of cell $n$, $Z_n = \sum_{n'} \exp\left(\frac{\pi_{n,n'}}{\sigma_n^2}\right)$, and $\sigma_n$ is the kernel width parameter.

By using the embedding position $\boldsymbol{e}_n$ of each cell $n$, the embedding velocity vector of cell $n$ can be derived by

$$
\boldsymbol{v}_{e,n} = \sum_{n' \neq n} \left(\tilde{\pi}_{n,n'} - \frac{1}{N}\right) \tilde{\boldsymbol{\delta}}_{n,n'},
\tag{10}
$$

where $\tilde{\boldsymbol{\delta}}_{n,n'} = \frac{\boldsymbol{e}'_n - \boldsymbol{e}_n}{\|\boldsymbol{e}'_n - \boldsymbol{e}_n\|}$, and $N$ is the total number of cells.

We construct the velocity graph and visualize embedding velocity stream plots using `scvelo.tl.velocity_graph` and `scvelo.pl.velocity_embedding_stream` from scVelo.

## Model evaluation

To compare the performance of different RNA velocity estimation methods, we compute several metrics to assess if the estimated velocity vectors are consistent and have the correct direction following the expected developmental trajectory.

The first one is CBDir[11]. This measures if the velocity vector of each cell has a correct direction to the downstream cells along the developmental trajectory. Specifically, assume that the expected developmental trajectory is from cluster $A$ to cluster $B$, then $(A, B)$ is called a pair of cluster edge and

$$\text{CBDir}_{A \to B} = \sum_{n \in A} \sum_{n' \in \mathcal{N}_n(i) \cap B} \cos(\boldsymbol{v}_{e,n}, \boldsymbol{x}_{e,n'} - \boldsymbol{x}_{e,n}),\qquad(11)$$

where $\mathcal{N}_n(n)$ is the $k$-NN for cell $n$, $\boldsymbol{x}_{e,n}$ and $\boldsymbol{x}_{e,n'}$ are the embedding expression vectors of cell $n$ and $n'$ in 2-dimensional space, and $\boldsymbol{v}_{e,n}$ is the embedding velocity vector for cell $n$ using the same dimensional-reduction algorithm as for expression, which is calculated as in Equation (10). To better check the situation in high-dimensional gene space, we also propose a new metric CBDir (Gene space) using the original expression vector $\boldsymbol{x}_n$ and velocity vector $\boldsymbol{v}_n$ instead of the embedding version:

$$\text{CBDir}_{A \to B} = \sum_{n \in A} \sum_{n' \in \mathcal{N}_n(n) \cap B} \cos(\boldsymbol{v}_n, \boldsymbol{x}_{n'} - \boldsymbol{x}_n).\qquad(12)$$

To distinguish CBDir (Gene space) from the original metrics, we refer to the initially proposed version as CBDir (UMAP space).

The second metric is transition cosine similarities (TransCosine), which is calculated by

$$\text{TransCosine}_{A \to B} = \sum_{n \in A} \sum_{n' \in \mathcal{N}_n(n) \cap B} \pi_{n,n'},\qquad(13)$$

for each pair of cluster edge $(A, B)$, where $\pi_{n,n'}$ is the $(n, n')$ element of velocity graph[9]. $\pi_{n,n'}$ is the cosine similarities between velocities and potential cell state transitions, calculated as in Equation (8). TransCosine measures if the velocity graph constructed is consistent with the cluster edge $(A, B)$.

The third metric is velocity coherence (VeloCoh)[14], which is calculated by

$$\text{VeloCoh}_n = \cos([\tilde{\boldsymbol{\Pi}}\mathbf{S}]_{n,:} - \boldsymbol{s}_n, \boldsymbol{v}_n),\qquad(14)$$

where $\tilde{\boldsymbol{\Pi}} = (\tilde{\pi}_{n,n'})$ is the cell-cell transitions probability matrix as in Eq. (9), $\mathbf{S}$ is the spliced RNA expression matrix and $[\tilde{\boldsymbol{\Pi}}\mathbf{S}]_{n,:}$ is the $n$-th row of $\tilde{\boldsymbol{\Pi}}\mathbf{S}$, which is the predicted future state of cell $n$. It measures if the predicted displacement of a cell $[\tilde{\boldsymbol{\Pi}}\mathbf{S}]_{n,:} - \boldsymbol{s}_n$ is coherent with the inferred velocity vector. To simplify the calculation, we use a version of $\tilde{\boldsymbol{\Pi}}$ as $\boldsymbol{\Pi}$ divided by its row sums.

For each cell $n$ (VeloCoh) and each pair of cluster edge $(A, B)$ (CBDir (Gene space), CBDir (UMAP space), TransCosine), the aforementioned scores are calculated and the mean values are used as the comparative metrics.

For cell-cycle datasets of FUCCI, we compare the performance of different methods using velocity sign accuracy proposed by veloVI. Specifically, assume that each cell in cell-cycle datasets of FUCCI has a cell cycle position $p_i$ with $p_i < p_{i+1}$ (each cell cycle position may correspond to several cells). The empirical velocity $\hat{\boldsymbol{v}}^{(p_i)}$ at position $p_i$ is calculated by $\hat{\boldsymbol{v}}^{(p_i)} \propto \bar{\boldsymbol{s}}^{(p_{i+1})} - \bar{\boldsymbol{s}}^{(p_i)}$, where $\bar{\boldsymbol{s}}^{(p_i)}$ is the mean spliced expression level of cells at position $p_i$. The estimated velocity at position $p_i$ is calculated by $\tilde{\boldsymbol{v}}^{(p_i)} \propto \bar{\boldsymbol{v}}^{(p_{i+1})} - \bar{\boldsymbol{v}}^{(p_i)}$, where $\bar{\boldsymbol{v}}^{(p_i)}$ is the mean of inferred velocity of cells at position $p_i$. Sign accuracy of position $p_i$ is calculated as the fraction of components that the signs of $\hat{\boldsymbol{v}}^{(p_i)}$ and $\tilde{\boldsymbol{v}}^{(p_i)}$

agree, accounting for positive velocity, negative velocity and zero velocity.

## Benchmarking methods

**scVelo.** scVelo implements preprocessing by the following functions:

```
scvelo.pp.filter_and_normalize(adata, min_shared_counts
=20, n_top_genes=2000)
scvelo.pp.moments(adata, n_neighbors=30, n_pcs=30)
```

For both stochastic mode and dynamical mode, we use the default setting of scVelo following the tutorial: https://scvelo.readthedocs.io/en/stable/.

Notably, the inherent randomness in scVelo's dynamical mode and veloVI may lead to differences between the reproduced results and the results shown in the original paper. The details are discussed in Supplementary Note 9.

**UniTVelo.** UniTVelo follows the preprocessing procedure of scVelo. The default configuration settings of UniTVelo are used as follows (https://unitvelo.readthedocs.io/en/latest/index.html):

```
velo_config=unitvelo.config.Configuration()
velo_config.R2_ADJUST=True
velo_config.IROOT=None
velo_config.FIT_OPTION='1'
velo_config.AGENES_R2=1
```

To ensure fairness, we set `velo_config.IROOT` to `None` for all datasets by default.

In UniTVelo, the preferred mode for datasets with cell cycle phase included (pancreas, retina, RPE1-FUCCI and U2OS-FUCCI) or with sparse cell types included (dentate gyrus) should be the independent mode (mode 2). For such datasets, we reproduced the results of UniTVelo by mode 2, using `velo_config.FIT_OPTION='2'`.

The details of configuration settings in UniTVelo are discussed in Supplementary Notes 10 and 11.

**cellDancer.** We adopted the same preprocessing procedure and hyperparameters in the model for all datasets when we implemented cellDancer:

- Using `scvelo.pp.filter_and_normalize` and `scvelo.pp.moments` for preprocessing with default arguments. Using "velocity genes" defined in UniTVelo for RNA velocity analysis.
- Inferring RNA velocity for each cell following the instructions in the tutorial page for mouse gastrulation (erythroid)(https://guangyuwanglab2021.github.io/cellDancer_website/notebooks/case_study_gastrulation.html):

  Using `celldancer.velocity` with `permutation_ratio`=0.125. Using `celldancer.compute_cell_velocity` with `projection_neighbor_size`=10.

For consistency across methods, we used the `scvelo.pl.velocity_embedding_stream` function in the scVelo package to visualize the results from cellDancer. To provide additional context, we also include the built-in visualization approaches from cellDancer.

The implementation details of cellDancer, including the preprocessing procedure, hyperparameter settings and visualization methods, are documented in Supplementary Note 12.

**veloVI.** veloVI follows the preprocessing procedure of scVelo. The model training and the velocity inference are performed using default model settings according to its tutorial: https://velovi.readthedocs.io/en/latest/index.html.

**DeepVelo.** DeepVelo follows the preprocessing procedure of scVelo. The model training and the velocity inference are performed

using default model settings according to its tutorial: https://github.com/bowang-lab/DeepVelo.

## Reporting summary
Further information on research design is available in the Nature Portfolio Reporting Summary linked to this article.

## Data availability
All datasets used in this research are openly accessible to the public.

**Pancreatic endocrinogenesis**: The pancreatic endocrinogenesis[38] data used in this study is available in the Gene Expression Omnibus (GEO) under accession code GSE132188. The dataset can be downloaded by running `scvelo.datasets.pancreas` in Python.

**Dentate gyrus development**: The dentate gyrus development[39] data (dentate gyrus 2) used in this study is available in GEO under accession code GSE95753. A sub-dataset (dentate gyrus) comprising two time points (P12 and P35) can be accessed by running `scvelo.datasets.dentategyrus` in Python.

**Mouse gastrulation (erythroid)**: The mouse gastrulation (erythroid)[40] data used in this study is available in ArrayExpress under accession code E-MTAB-6967. The dataset can be downloaded by running `scvelo.datasets.gastrulation_erythroid` in Python.

**Mouse hindbrain (Oligo)**: The mouse hindbrain (Oligo)[41] data can be downloaded from the website of Kharchenko Lab at https://pklab.med.harvard.edu/ruslan/velocity/oligos/.

**Mouse hindbrain (GABA, Glial)**: The mouse hindbrain (GABA, Glial)[42] data is available in GEO under accession code GSE118068. The processed loom files are available from Figshare[43] of DeepVelo.

**Intestinal organoid**: The intestinal organoid[32] data is available in GEO under accession code GSE128365. The dataset can be downloaded from the Dynamo package by running `dynamo.sample_data.scEU_seq_organoid` in Python.

**Mouse retina development**: The mouse retina development[44] data is available in GEO under accession code GSM3466902. The dataset can be downloaded from the website of Kharchenko Lab at http://pklab.med.harvard.edu/peterk/review2020/examples/retina/.

**scNT-seq neuron KCl stimulation**: The scNT-seq neuron KCl stimulation[45] data is available in GEO under accession code GSE141851. The dataset can be downloaded from GitHub page at https://github.com/wulabupenn/scNT-seq.

**Mouse embryonic fibroblast reprogramming**: The mouse embryonic fibroblast reprogramming[46] data is available in GEO under accession code GSE99915. The dataset can be downloaded from the CellRank package by running `cellrank.datasets.reprogramming_morris` in Python.

**FUCCI**: The RPE1-FUCCI[34] and U2OS-FUCCI[32] data can be downloaded from Figshare[47,48] of veloVI.

**Embryonic mouse brain**: The embryonic mouse brain data from 10x Genomics is available at 10x website.

**SHARE-seq mouse skin**: The SHARE-seq mouse skin[49] data is available in GEO under accession code GSE140203. The RNA and ATAC datasets can be downloaded from Figshare[50,51] of MultiVelo.

**Human HSPCs**: The human HSPCs[52] data is available in GEO under accession code GSE70677. The RNA and ATAC datasets can be downloaded from Figshare[53,54] of MultiVelo.

**Developing human brain**: The developing human brain[55] data is available in GEO under accession code GSE162170. The RNA and ATAC datasets can be downloaded from Figshare[56,57] of MultiVelo.

The processed datasets used in this study have been deposited to Figshare and can be accessed at: https://doi.org/10.6084/m9.figshare.27643494.

## Code availability
The TIVelo tool is developed as a Python package and is openly available for use at https://github.com/cuhklinlab/TIVelo, as well as deposited in Zenodo[58] (https://doi.org/10.5281/zenodo.15637938). Comprehensive documentation, including detailed installation guidelines and steps to reproduce the results presented in this study, is available at https://tivelo.readthedocs.io/en/latest/.

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

## Acknowledgements

We thank Prof. Hongyu Zhao from Yale University for his valuable and constructive suggestions on the manuscript. This work has been supported by the Chinese University of Hong Kong startup grant (4930181 to Z.L.), the Chinese University of Hong Kong Science Faculty's Direct Grant for Research 2023/2024 (to Z.L.), the Chinese University of Hong Kong Science Faculty's Collaborative Research Impact Matching Scheme (CRIMS 4620033 to Z.L.), the Hong Kong Research Grant Council (GRF 14301120 to Z.L., GRF 14300923 to Z.L.).

## Author contributions

Z.L. supervised this study. M.G. and Z.L. proposed and developed TIVelo's computational method. M.G. conducted data analysis. J.Q., J.M., and X.Z. provided advice on data analysis. M.G. and J.M. developed and released TIVelo's package. M.G. and Z.L. drafted the manuscript. J.Q. and X.Z. revised the manuscript.

## Competing interests
The authors declare no competing interests.

## Additional information

**Peer review information** : *Nature Communications* thanks the anonymous reviewers for their contribution to the peer review of this work. A peer review file is available.

