## [Transparent Peer Review file · Nature Communications]

TIVelo: RNA velocity estimation leveraging cluster-level trajectory inference

Corresponding Author: Dr Zhixiang Lin

Version 0:

Reviewer comments:

Reviewer #1

(Remarks to the Author)

In this manuscript, the author proposed a new computational method TIVelo to perform a directed trajectory inference and RNA velocity estimation. Uniquely, this method primarily focuses on detecting the directionality of the main path in a cell-type graph, where the unspliced and spliced RNAs will be leveraged to determine if a proposed directionality is correct or not. Once the directed main path is confirmed, the RNA velocity and cell transition probabilities will be computed by averaging of the descendent neighboring cells.

Despite there being already quite a few RNA velocity methods, this paper has an important angle to prioritizing the directionality inference, which all current RNA velocity methods may struggle occasionally. The manuscript is well written with a nice literature review. It also contains comprehensive benchmarking datasets, with notebooks for all 16 datasets included.

Here are a few comments for the authors to consider for revision:

1. A key part of the method is to use unspliced and spliced RNAs to correct the direction of the main path. Since this is the foundation of the whole method, it is worth more evaluation of how robust the proposed methods are in correcting the directionality and the choice of the main path. While there are cases where the authors showed the reverse directionality can be corrected, it will be highly beneficial to add such analysis for all datasets, e.g., via a table if there might be too many figures. As this task is challenging, it will be acceptable if sometimes it cannot work correctly but it will be a good reference for the users.
2. Related to point 1, another more critical challenge is how to correct if the main path is not right. The space of such an issue can be extremely large, but sometimes critically important for systems with unknown prior. A few example analyses would be a nice reference for the users too, for example, what if choosing GMP as the origin node in the HSC dataset (Fig. 6a)?
3. For line 104, the re-scale of u and s are critical for the method, I would suggest the authors add a sentence about the normalization (rescale to sum to 1, for each gene) in the main text and possibly also in Fig. 1e. Also, how about other alternative normalization methods if the current option cannot achieve all correct results regarding point 1?
4. For metric CBDir2 used in Fig 6 and 7, you may consider renaming it with a more informative term, e.g., CBDir (Gene space), to distinguish it from the default CBDir (UMAP space).
5. For a few cases, the preferred mode of UniTVelo may be "2" (the independent mode), especially when there is a cell cycle, e.g., for Fig. 5, S2, and S7.
6. Some minor text issues:
 - line 268: "as is shown" -> "as shown"
 - line 503: "induction" -> "repression"
 - line 509: "more larger" -> "larger"

(Remarks on code availability)

I didn't run the package, but the repository looks very well organized and provides analysis notebooks for all 16 datasets.

Reviewer #2

(Remarks to the Author)

(Remarks on code availability)

The code is well documented, and it contains the guidance to reproduce the figures.

Reviewer #3

(Remarks to the Author)

The authors introduce TIVelo, a novel RNA velocity estimation method that avoids explicit reliance on ordinary differential equation (ODE) assumptions, allowing it to capture complex transcriptional dynamics. The method involves clustering and trajectory detection, followed by determining the velocity direction using the relationship between unspliced and spliced RNA. The time derivative of spliced and unspliced RNA is calculated based on the differences between a cell and its directed neighbors. I commend the authors for validating their approach on numerous datasets, as well as their new ideas about inferring the orientation of trajectory using the relations between unspliced and spliced RNA. However, I have the following concerns about this study:

1. The motivation behind this study could be better explained. For instance, in lines 71-72, the authors mention: Some variants of the ODE model attempt to address this by introducing variable rate parameters; however, the fundamental issue persists as long as the ODE framework is employed.

Could the authors clarify which variant ODE models have been proposed to address complex transcriptional dynamics? Additionally, why do these models fail to overcome the limitations? More detailed examples or references would strengthen this argument, which could make the contribution of this study be clearer.

2. TIVelo adopts a distinct approach to RNA velocity estimation by first inferring trajectories and pseudotime. It calculates velocities between neighboring cells, rather than directly modeling gene-specific dynamics. In contrast, other methods typically model the transcriptional dynamics of each gene individually. As a result, TIVelo appears more as a trajectory inference method with automatic initial state detection, with RNA velocity calculation being a downstream application. Intuitively, this strategy may lead to smoother velocity streams generated by TIVelo. Therefore, I am concerned that the metrics shown in Fig. 7, which tend to favor smoother streams, may introduce a bias toward TIVelo.

3. Currently, the authors focus on demonstrating that the trajectory and initial state inferred by TIVelo accurately fit the ground truth process across all datasets. However, it would strengthen the contribution of this study and make the paper more compelling if the authors could present some scientific findings using TIVelo. Specifically, given that velocity estimation is the final step of the framework, it would be valuable to see how the inferred velocities can benefit biological studies.

4. In line 437, that terminal cells are identified using `scvelo.tl.terminal_states`, which assumes that terminal cells exhibit extreme values of unspliced and spliced RNA. This assumption may not hold in complex or multi-branch datasets. Could the authors provide more details on how terminal states were modeled and optimized for such cases? For instance, I recommend provide similar details shown in Figure 1 (e.g., initialization of terminal states, pseudotime, orientation scores, main paths, etc.) for more datasets, such as HSPCs and dentate gyrus. This could provide a clearer understanding of how TIVelo works across diverse scenarios.

5. Fig 2k. The arrows in "Blood progenitors 1" do not appear to point toward the next cluster, "Blood progenitors 2." Could the authors clarify if this result is accurate?

6. Fig 5b. The arrows in cluster 1 lack coherence in their orientation. How do these results reflect the advantages of TIVelo compared to other methods? Additional explanation or alternative visualizations would be helpful.

7. The absence of installation documentation and tutorials in the GitHub repository makes it challenging for users to apply the code. I recommend including a clear installation guide and example usage tutorials to enhance reproducibility and accessibility.

(Remarks on code availability)

The absence of installation documentation and tutorials in the GitHub repository makes it challenging for users to apply the code. I recommend including a clear installation guide and example usage tutorials to enhance reproducibility and accessibility.

Reviewer #4

(Remarks to the Author)

The manuscript introduces TIVelo, a cluster-level tree-based method for estimating RNA velocity under the premise that unspliced RNA is invariably expressed and repressed earlier than spliced RNA. Although this approach applies an interesting perspective by linking unspliced and spliced dynamics with a cluster-level trajectory tree, its novelty appears limited. The current study does not compellingly demonstrate the potential of TIVelo to yield groundbreaking insights or robust practical applications. Moreover, without ODE assumptions, claiming TIVelo is solving RNA velocity may not be accurate. By avoiding the estimation of kinetic parameters (e.g., transcription, splicing, and degradation rates), the method cannot provide quantitative insights that are typically essential for understanding RNA velocity and cellular dynamics.

Major Concerns

1. Inconsistencies in Comparative Evaluations

o The comparative analyses provided for UniTVelo, veloVI, scVelo, and cellDancer frequently conflict with results reported in their respective original studies. For instance, UniTVelo's directionality in the manuscript appears reversed compared to the original publication. Similar discrepancies arise in figures for scVelo (Fig. S1), cellDancer (Figs. S2, S3, S5), and again UniTVelo (Fig. 3, S7).

o The authors must verify that all external methods were correctly implemented and that their parameters align with the original protocols. Such inconsistencies undermine the reliability of the study's comparisons and conclusions.

2. Limited Novelty and Justification

o The notion that unspliced RNA consistently precedes spliced RNA is not unique to TIVelo; other methods (e.g., DeepVelo, cellDancer) already account for variable kinetics rather than constant parameters. Thus, the manuscript's argument that TIVelo offers a more flexible or "model-free" strategy seems incomplete without acknowledging these existing advancements.

o The authors should offer a clearer explanation of how TIVelo substantially advances the field beyond prior methods, especially if it does not infer cell-specific rates.

3. Insufficient Clarity and Context

o The statement that scVelo's velocity vectors are reversed (lines 223–225) lacks supporting visual evidence in Fig. 3a, which does not include scVelo results. The manuscript needs additional figures or references to substantiate this claim.

o In the supplementary file (lines 438–439), the reasoning behind using `scvelo.tl.terminal_states` requires further contextualization. The claim that root/end cells "typically exhibit extreme values in unspliced and spliced RNA" lacks proper references and does not universally apply to all biological systems.

4. Lack of Broader Comparisons

o The study would benefit from direct comparisons with additional tools such as DeepVelo and CellRank 2, which also address limitations in constant-parameter modeling. Incorporating these methods would give a more comprehensive benchmark and help situate TIVelo within the broader landscape of RNA velocity approaches.

(Remarks on code availability)

Version 1:

Reviewer comments:

Reviewer #1

(Remarks to the Author)

I thank the authors for the detailed revision; all my concerns have been well addressed.

One more note is on Fig. 4a, where UniTVelo's results are reversed from its original report. Possibly, the author may briefly mention the reason, for example, something like "Of note, the result of UniTVelo is different from its original report, probably due to the use of (default) random initialization here, while the original study used a warm initialization."

(Remarks on code availability)

On briefly looked at some notebooks. Seems all fine; probably it needs to use a publicly available data directory (or path).

Reviewer #2

(Remarks to the Author)

(Remarks on code availability)

Reviewer #3

(Remarks to the Author)

The authors have addressed all my concerns, and I am happy to recommend the manuscript for publication in Nature Communications.

(Remarks on code availability)

I have assessed the provided code and found it to be well-documented and user-friendly. The repository includes a comprehensive README file with clear instructions on installation, dependencies, and execution of the main analysis pipeline. I was able to install the required packages and run the core components of the code without major issues.

Reviewer #4

(Remarks to the Author)

The authors addressed my concerns. I have no more questions.

(Remarks on code availability)

Response to Reviewers

We sincerely thank the reviewers for their constructive comments and suggestions, which have significantly improved our manuscript. We have carefully addressed each of the comments and incorporated the necessary revisions. Below is a summary of the key changes made:

- **The advantages of TIVelo over methods with variable kinetic rate parameters in the ODE equation:** We discussed the limitations of two methods which utilize variable kinetic rate parameters in the ODE equation rather than constant parameters: DeepVelo and cellDancer. We further demonstrated the advantages of TIVelo over these two methods (Supplementary Notes S1).
- **Scientific findings based on TIVelo’s inferred velocities:** We carried out several downstream analyses based on TIVelo’s inferred velocities, including fate probability visualization, lineage-specific driver gene identification, and identification of macrostates with functional insights, highlighting the key scientific insights enabled by TIVelo (Supplementary Notes S2).
- **Kinetic rate mode of TIVelo:** We developed a new mode (kinetic rate mode) for fitting RNA velocity in TIVelo, which enables simultaneous inference of cell-specific kinetic rates (α , β and γ) while maintaining the framework’s core functionality. This reveals the cellular dynamics of the transcriptional phase for individual genes (Methods; Supplementary Notes S2).
- **Clarification for the inferred velocity in RPE1-FUCCI:** We provided more explanation for the different orientations of the inferred velocity streams from TIVelo in cluster 1 of dataset *RPE1-FUCCI*. We also compared the *RPE1-FUCCI* results from TIVelo, DeepVelo, UniTVelo and veloVI (Supplementary Notes S3).
- **Origin node selection of TIVelo:** We provided detailed explanation for the origin node selection strategy in TIVelo, offering clearer insight into how the method works across diverse scenarios (Supplementary Notes S4).
- **Normalization strategies comparison:** We proposed an alternative strategy of normalization in orientation score calculation. We compared this alternative strategy with the default normalization strategy (Supplementary Notes S5).
- **Robustness of TIVelo:** We carried out several experiments, including the evaluation of TIVelo’s performance on the perturbed main path, and on the main path with reversed direction. Furthermore, we enhanced TIVelo’s robustness with respect to the origin node through the refinement of the origin node selection strategy (Supplementary Notes S6, S7).
- **Root cell selection:** We refined the strategy for root cell selection in RNA velocity inference, which improved TIVelo’s inference accuracy within the root cluster for *Gastrulation (Erythroid)* and *RPE1-FUCCI* (Supplementary Notes S8).
- **Clarification for the reproduction of benchmarking methods:** We provided detailed explanation for the differences between the results in our reproduction and in the paper of benchmarking methods, including scVelo, veloVI, UniTVelo and cellDancer. We updated the reproduction of UniTVelo in five datasets and the quantitative comparison results correspondingly (Fig.7; Supplementary Notes S9, S10, S11, S12).

We believe that we have addressed each comment and suggestion from the reviewers thoroughly. Detailed responses to their comments are provided below.

Replies to Reviewer 1

In this manuscript, the author proposed a new computational method TIVelo to perform a directed trajectory inference and RNA velocity estimation. Uniquely, this method primarily focuses on detecting the directionality of the main path in a cell-type graph, where the unspliced and spliced RNAs will be leveraged to determine if a proposed directionality is correct or not. Once the directed main path is confirmed, the RNA velocity and cell transition probabilities will be computed by averaging of the descendent neighboring cells.

Despite there being already quite a few RNA velocity methods, this paper has an important angle to prioritizing the directionality inference, which all current RNA velocity methods may struggle occasionally. The manuscript is well written with a nice literature review. It also contains comprehensive benchmarking datasets, with notebooks for all 16 datasets included.

Here are a few comments for the authors to consider for revision.

Response: Thank you for your thoughtful comments. We have thoroughly revised the manuscript to address your suggestions and enhance its overall quality.

1. A key part of the method is to use unspliced and spliced RNAs to correct the direction of the main path. Since this is the foundation of the whole method, it is worth more evaluation of how robust the proposed methods are in correcting the directionality and the choice of the main path. While there are cases where the authors showed the reverse directionality can be corrected, it will be highly beneficial to add such analysis for all datasets, e.g., via a table if there might be too many figures. As this task is challenging, it will be acceptable if sometimes it cannot work correctly but it will be a good reference for the users.

Response: We sincerely appreciate the reviewer’s constructive comments. Firstly, to evaluate how robust TIVelo is in correcting the directionality and the choice of the main path, we added perturbations to the original main path M selected by TIVelo. For datasets containing multiple branches, the perturbed main path incorporates one new branch that was not included in the original main path, while for single-branch datasets, the perturbed main path is shorter compared to the original main path. The details for obtaining the perturbed main path M' are as follows:

Denote the main path as $M = \{O, C_1, \dots, C_m\}$, where O is the origin node and C_1, \dots, C_m are subsequent nodes.

1. For the case when the expected root cluster is selected as the origin node:
 - If there is one branch B connecting to C_j , $j = 1, \dots, m$, namely $B = \{C_j, C'_1, \dots, C'_n\}$, then we select a perturbed main path $M' = \{O, C_1, \dots, C_j, C'_1, \dots, C'_n\}$.
 - If there are several such branches B_1, \dots, B_b , we prioritize the branch that maximizes the number of cells included in the perturbed main path.
 - If there is no such branch, we drop the last cluster C_m in the main path, and the perturbed main path is selected as $M' = \{O, C_1, \dots, C_{m-1}\}$.
2. For the case when the expected end cluster is selected as the origin node:
 - If there exist several expected end clusters, we select another expected end cluster as the origin node O' . The perturbed main path is selected based on the new origin node O' (Methods).
 - If there is only one expected end cluster, we drop O in the main path, and the perturbed main path is selected as $M' = \{C_1, \dots, C_{m-1}\}$.

For the perturbed main path, we tested whether TIVelo can correctly infer the expected direction on the main path. The results are summarized in Table R1.

For all 16 datasets used in TIVelo, the inferred velocity direction of the perturbed main path M' from TIVelo remains correct in 13 of them. This indicates that the velocity direction of the main path inferred by TIVelo is robust to moderate perturbations to the main path.

Datasets	Score on M	Direction	Score on M'	Direction
Pancreas	16.072	✓	16.371	✓
Dentate Gyrus	1.138	✓	3.905	✓
Gastrulation	4.742	✓	5.106	✓
Hindbrain (Oligo)	48.319	✓	48.570	✓
Intestinal Organoid	-0.649	✓	-1.614	✓
Retina	-6.399	✓	-0.006	✓
scNT	-17.159	✓	-13.269	✓
Dentate Gyrus 2	9.082	✓	4.640	✓
Reprogramming	1.401	✓	10.188	✗
Hindbrain (GABA, Glial)	-0.158	✓	-1.292	✓
FUCCI RPE1	7.965	✓	9.906	✓
FUCCI U2OS	-1.610	✓	-1.544	✓
Mouse Brain	0.933	✓	5.215	✓
Mouse Skin	1.025	✓	-1.464	✗
HSPCs	-7.773	✓	0.093	✗
Human Brain	6.923	✓	4.163	✓

Table R1 | The comparison of velocity direction inferred by TIVelo for the original main path M and the perturbed main path M' across the 16 datasets. Score on $M(M')$: mean orientation score on the main path $M(M')$. Direction: if the inferred velocity direction on the main path $M(M')$ is correct.

Secondly, for datasets where the mean orientation score on the main path M is positive (there are ten datasets in total), we reversed the direction of the main path, i.e., main path $M = \{O, C_1, \dots, C_m\}$ will become $M' = \{C_m, \dots, C_1, O\}$. We then assessed if TIVelo can correctly infer the velocity direction of the perturbed main path M' . The results of this evaluation are summarized in Table R2.

Datasets	Score on M	Direction	Score on M'	Direction
Pancreas	16.072	✓	-15.501	✓
Dentate Gyrus	1.138	✓	0.184	✗
Gastrulation	4.742	✓	-13.490	✓
Hindbrain (Oligo)	48.319	✓	-50.613	✓
Dentate Gyrus 2	9.082	✓	0.189	✓
Reprogramming	1.401	✓	2.119	✗
FUCCI RPE1	7.965	✓	-19.005	✓
Mouse Brain	0.933	✓	-3.637	✓
Mouse Skin	1.025	✓	-1.968	✓
Human Brain	6.923	✓	-3.166	✓

Table R2 | The comparison of velocity direction inferred by TIVelo for the original main path M and the perturbed main path M' across ten datasets where the mean orientation score on the main path M is positive. Score on $M(M')$: mean orientation score on the main path $M(M')$. Direction: if the inferred velocity direction on the main path $M(M')$ is correct.

From the analysis, TIVelo can correctly infer the velocity direction of the new main path M' in eight

out of ten datasets we tested. This finding further underscores the robustness of TIVelo in handling variations in main path selection.

The discussion on this topic is now presented in Supplementary Notes S6 of the revised manuscript.

2. Related to point 1, another more critical challenge is how to correct if the main path is not right. The space of such an issue can be extremely large, but sometimes critically important for systems with unknown prior. A few example analyses would be a nice reference for the users too, for example, what if choosing GMP as the origin node in the HSC dataset (Fig. 6a)?

Response: We sincerely appreciate your insightful comment regarding the robustness of TIVelo in handling potential inaccuracies in origin node selection. The selection of the origin node impacts the main path selection and more specifically the inference of root node in TIVelo. To evaluate this impact of the different selections of origin node on TIVelo’s performance, in each dataset we selected the terminal state O' with the **second strongest signal** as the origin node. The details for selecting this new origin node and its corresponding main path are as follows:

1. If the origin node O is selected according to the root score (i.e., with largest R_O and $R_O > 0.1$);
 - A new origin node O' is selected when the end score $E_{O'}$ is the largest and $E_{O'} > 0.1$;
 - If there is no such O' , O' is selected as the cell cluster with the second largest root score.
2. If the origin node O is selected according to the end score (i.e., with largest E_O and $E_O > 0.1$);
 - A new origin node O' is selected when the root score $R_{O'}$ is the largest and $R_{O'} > 0.1$;
 - If there is no such O' , O' is selected as the cell cluster with the second largest end score.
3. The procedure for selecting main path remains the same as before using the new origin node (Methods).

Datasets	O	Root	O'	Root
Pancreas	Ductal	✓	Beta	✓
Dentate Gyrus	nIPC	✓	Astrocytes	✗
Gastrulation	Blood progenitors 1	✓	Erythoid 2	✓
Hindbrain (Oligo)	OPCs	✓	MFOLs	✓
Intestinal Organoid	Paneth cells	✓	Entrocytes	✓
Retina	AC/HC	✓	PR	✓
scNT	Time 120	✓	Time 60	✓
Dentate Gyrus 2	nIPC	✓	Granule	✗
Reprogramming	Cluster 9	✓	Cluster 3	✗
Hindbrain (GABA, Glial)	GABA interneurons	✓	Differentiating GABA	✓
FUCCI RPE1	Leiden 3	✓	Leiden 1	✗
FUCCI U2OS	Leiden 1	✓	Leiden 2	✓
Mouse Brain	RG, Astro, OPC	✓	Deeper Layer	✗
Mouse Skin	TAC-1	✓	IRS	✗
HSPCs	Platelet	✓	LMPP	✗
Human Brain	Cyc.	✓	ExDp	✗

Table R3 | The comparison of velocity direction inferred by TIVelo for the original main path M based on origin node O , and the new main path M' based on new origin node O' . Root: If TIVelo can correctly select the expected root node R .

For the newly selected origin node O' and its corresponding main path M' , we tested whether TIVelo can correctly infer the expected root cluster R : Specifically, $M' = \{O', C_1, \dots, C_m\}$ and we tested if $O' = R$ when the mean orientation score on M' is positive, and if $C_m = R$ when the mean orientation score on M' is negative. If the root cluster is inferred correctly in TIVelo, it is likely that the velocity inference will also be accurate. The results are summarized in Table R3.

From Table R3, TIVelo can correctly infer the expected root cluster in 8 out of 16 datasets when the new origin node O' is used. This highlights the importance of origin node selection in TIVelo's performance, demonstrating that origin node selection can impact the accuracy of the final results.

To enhance the robustness of TIVelo, we integrated **CytoTRACE2**¹, a method for inferring cell potency from scRNA-seq data, into TIVelo to assist in origin node selection in the revised manuscript. In TIVelo, after selecting the origin node O based on `scvelo.tl.terminal_states`, we can use CytoTRACE2 to calculate the median potency score P_i (from 0 to 1) for each cell cluster i . The results from CytoTRACE2 serve as an additional reference for refining origin node selection. The details are as follows.

1. If cell cluster O has the highest or lowest median potency scores across all clusters, O is selected as the origin node.
2. If cell cluster O does not have the highest or lowest median potency scores across all cell clusters, we select the cell cluster O'' with highest median potency score $P_{O''}$ as the origin node.

Fig.R1 displays the box plots of potency scores of each cell cluster across the 16 datasets. We observed that in most datasets, the origin node O selected by `scvelo.tl.terminal_states` exhibits either the highest or lowest median potency score.

(c) *Gastrulation (Erythroid)*. O: Blood progenitors 1.

(d) *Hindbrain (Oligo)*. O: OPCs.

(e) *Intestinal Organoid*. O: Paneth cells.

(f) *Retina*. O: AC/HC.

(g) *scNT*. O: Time 120.

(h) *Dentate Gyrus 2*. O: nIPC.

(i) *Reprogramming. O*: Cluster 9.

(j) *Hindbrain (GABA Glial). O*: GABA interneurons.

(k) *FUCCI RPE1. O*: Leiden 3.

(l) *FUCCI U2OS. O*: Leiden 1.

(m) *Mouse Brain. O*: RG, Astro, OPC.

(n) *Mouse skin. O*: TAC-1.

(o) *HSPCs*. O : Platelet.

(p) *Human Brain*. O : Cyc.

Fig. R1 | Box plots of median potency scores of each cell cluster across the 16 datasets, inferred by CytoTRACE2.

Furthermore, we denote the cluster with the second strongest signal in `scvelo.tl.terminal_states` as O' , and the origin node after CytoTRACE2 refinement (for O') as O'' . We then evaluated whether TIVelo could correctly infer the expected root node when O' and O'' were used as the origin node, respectively. From Table R4, when O'' is selected as the origin node, TIVelo successfully infers the expected root node for all datasets except *Dentate Gyrus 2* and *Mouse Brain*. In contrast, when O' is used as the origin node, only 8 out of the 16 datasets yield the correct root node inference.

Datasets	O'	Root	O''	Root
Pancreas	Beta	✓	Beta	✓
Dentate Gyrus	Astrocytes	✗	nIPC	✓
Gastrulation	Erythoid 2	✓	Blood Progenitors 1	✓
Hindbrain (Oligo)	MFOLs	✓	MFOLs	✓
Intestinal Organoid	Entrocytes	✓	Entrocytes	✓
Retina	PR	✓	Progenitor	✓
scNT	Time 60	✓	Time 60	✓
Dentate Gyrus 2	Granule	✗	Granule	✗
Reprogramming	Cluster 3	✗	Cluster 9	✓
Hindbrain (GABA, Glial)	Differentiating GABA interneurons	✓	Neural stem cells	✓
FUCCI RPE1	Leiden 1	✗	Leiden 0	✓
FUCCI U2OS	Leiden 2	✓	Leiden 1	✓
Mouse Brain	Deeper Layer	✗	Deeper Layer	✗
Mouse Skin	IRS	✗	TAC-1	✓
HSPCs	LMPP	✗	Platelet	✓
Human Brain	ExDp	✗	Cyc.	✓

Table R4 | The comparison of TIVelo’s results when O' or O'' is selected as the origin node. Root: if TIVelo can correctly infer the expected root node R when O' (O'') is selected as the origin node.

Finally, let’s address the specific example raised by the reviewer. In the dataset *HSPCs*, if the cell cluster GMP is selected as the origin node, it does not exhibit the highest or lowest median potency

score inferred by CytoTRACE2 (Fig.R1(o)). In this case, GMP is replaced by Platelet, with the highest median potency score, as the origin node. This adjustment enables TIVelo to correctly infer the expected root node HSC.

The discussion on this topic is now presented in Supplementary Notes S7 of the revised manuscript.

3. For line 104, the re-scale of u and s are critical for the method, I would suggest the authors add a sentence about the normalization (rescale to sum to 1, for each gene) in the main text and possibly also in Fig. 1e. Also, how about other alternative normalization methods if the current option cannot achieve all correct results regarding point 1?

Response: We sincerely thank the reviewer for this valuable suggestion and insightful question. As recommended, we have added a sentence in the main text and Fig.1e to clarify the normalization process, specifically stating that u and s are rescaled to sum to 1 for each gene.

From our perspective, an alternative normalization approach is to normalize by the maximum values of u and s for each gene, i.e., $\tilde{u}_g = u_g/u_{g,\max}$ and $\tilde{s}_g = s_g/s_{g,\max}$. In our experiments, this alternative way of normalization cannot correctly infer the velocity direction in four datasets (*Intestinal Organoid*, *Hindbrain (GABA, Glial)*, *FUCCI RPE1* and *FUCCI U2OS*), as shown in Table R5. Furthermore, for the datasets where the current normalization method does not fully address the issues raised in point 1 (e.g., *Reprogramming* in Table R2), normalization by the maximum value also fails to resolve these issues. Therefore, we have retained the strategy of normalization by sum as the default approach.

Datasets	Sum	Direction	Maximum	Direction
Pancreas	16.072	✓	7.915	✓
Dentate Gyrus	1.138	✓	2.311	✓
Gastrulation	4.742	✓	0.507	✓
Hindbrain (Oligo)	48.319	✓	14.965	✓
Intestinal Organoid	-0.649	✓	2.919	✗
Retina	-6.399	✓	-2.063	✓
scNT	-17.159	✓	-9.801	✓
Dentate Gyrus 2	9.082	✓	3.033	✓
Reprogramming	1.401	✓	1.210	✓
Hindbrain (GABA, Glial)	-0.158	✓	1.515	✗
FUCCI RPE1	7.965	✓	-2.443	✗
FUCCI U2OS	-1.610	✓	6.085	✗
Mouse Brain	0.933	✓	1.129	✓
Mouse Skin	1.025	✓	2.099	✓
HSPCs	-7.773	✓	-1.703	✓
Human Brain	6.923	✓	7.066	✓

Table R5 | The comparison of inferred velocity direction on the main path, by two normalization ways across the 16 datasets. Sum: mean orientation score (across all genes) on the main path, by normalization in which u and s are rescaled to sum to 1. Maximum: mean orientation score (across all cells and genes) on the main path, by normalization in which u and s are rescaled by $u_{g,\max}$ and $s_{g,\max}$. Direction: if the inferred velocity direction on the main path is correct.

The discussion on this topic is now presented in Supplementary Notes S5 of the revised manuscript.

4. For metric CDir2 used in Fig 6 and 7, you may consider renaming it with a more informative

term, e.g., CDBir (Gene space), to distinguish it from the default CDBir (UMAP space).

Response: We sincerely appreciate the reviewer’s thoughtful suggestion. In response, we have renamed CDBir2 to CDBir (Gene space) throughout the manuscript to clearly distinguish it from the default CDBir (UMAP space). This change ensures better clarity and alignment with the methodological context.

5. For a few cases, the preferred mode of UniTVelo may be “2” (the independent mode), especially when there is a cell cycle, e.g., for Fig. 5, S2, and S7.

Response: We sincerely thank the reviewer for raising this important point. In response, we have updated the results for the mentioned datasets (Fig.5, S2, and S7) using UniTVelo² in the independent mode (mode 2). The updated results are presented in Fig.R2-R5. We also updated the quantitative comparison results in Fig.5 and Fig.7 correspondingly in our manuscript, as shown in Fig.R6-R10.

Fig. R2 | Comparison of velocity stream plots inferred by UniTVelo mode 1 and mode 2 for *Pancreas*.

Fig. R3 | Comparison of velocity stream plots inferred by UniTVelo mode 1 and mode 2 for *Retina*.

Fig. R4 | Comparison of velocity stream plots inferred by UniTVelo mode 1 and mode 2 for *RPE1-FUCCI*.

Fig. R5 | Comparison of velocity stream plots inferred by UniTVelo mode 1 and mode 2 for *U2OS-FUCCI*.

Fig. R6 | Comparison of TIVelo and six benchmarking methods by cross-boundary direction correctness (Gene space) score across ten datasets.

Fig. R7 | Comparison of TIVelo and six benchmarking methods by transition cosine similarities (TransCosine) across ten datasets.

Fig. R8 | Comparison of TIVelo and six benchmarking methods by velocity coherence (VeloCoh) across ten datasets.

Fig. R9 | Violin plots showing the comparison of sign accuracy from TIVelo and other six benchmarking methods on *RPE1-FUCCI*. The mean sign accuracy of each method is annotated below.

Fig. R10 | Violin plots showing the comparison of sign accuracy from TIVelo and other six benchmarking methods on *U2OS-FUCCI*. The mean sign accuracy of each method is annotated below.

The discussion on this topic is now presented in Supplementary Notes S10 of the revised manuscript.

6. Some minor text issues: - line 268: “as is shown” -> “as shown”; - line 503: “induction” -> “repression”; - line 509: “more larger” -> “larger”.

Response: We sincerely appreciate the reviewer’s careful reading and valuable suggestions. We have corrected the typos mentioned in our manuscript.

Remarks on code availability: I didn’t run the package, but the repository looks very well organized and provides analysis notebooks for all 16 datasets.

Response: We thank the reviewer for their positive feedback on our code organization and comprehensive analysis notebooks.

Replies to Reviewer 2

Response: We sincerely appreciate the reviewer's time and constructive comments for evaluating our manuscript.

Remarks on code availability: The code is well documented, and it contains the guidance to reproduce the figures.

We thank the reviewer for their positive feedback on our code organization.

Replies to Reviewer 3

The authors introduce TIVelo, a novel RNA velocity estimation method that avoids explicit reliance on ordinary differential equation (ODE) assumptions, allowing it to capture complex transcriptional dynamics. The method involves clustering and trajectory detection, followed by determining the velocity direction using the relationship between unspliced and spliced RNA. The time derivative of spliced and unspliced RNA is calculated based on the differences between a cell and its directed neighbors. I commend the authors for validating their approach on numerous datasets, as well as their new ideas about inferring the orientation of trajectory using the relations between unspliced and spliced RNA. However, I have the following concerns about this study.

Response: We sincerely thank the reviewer for their insightful and constructive comments on our work. We appreciate their recognition of TIVelo’s novel approach and thorough benchmarking. We have carefully addressed each of their concerns in the revised manuscript to improve clarity and rigor.

1. The motivation behind this study could be better explained. For instance, in lines 71-72, the authors mention: Some variants of the ODE model attempt to address this by introducing variable rate parameters; however, the fundamental issue persists as long as the ODE framework is employed. Could the authors clarify which variant ODE models have been proposed to address complex transcriptional dynamics? Additionally, why do these models fail to overcome the limitations? More detailed examples or references would strengthen this argument, which could make the contribution of this study be clearer.

Response: We sincerely thank the reviewer for this insightful comment. To address the reviewer’s question, we would like to highlight two representative methods that infer variable rate parameters across cells: DeepVelo³ and cellDancer⁴, and illustrate the advantages of TIVelo over these two methods. The following discussion is presented in the Supplementary Notes S1 in the revised manuscript to provide a more comprehensive and detailed argument.

DeepVelo

DeepVelo uses graph convolutional networks (GCN) to estimate cell-specific kinetic rates:

$$(\alpha, \beta, \gamma) = \text{GCN}(U, S|A) \quad (1)$$

where $U, S \in \mathbb{R}^{N \times G}$ are cell by gene unspliced and spliced RNA input, $\alpha, \beta, \gamma \in \mathbb{R}^{N \times G}$ are estimated kinetic rates, and A is the adjacency matrix depicting the cell nearest neighbor graph.

The loss function designed by DeepVelo is as follows

$$\begin{aligned} \mathcal{L}_+ &= \frac{1}{|\Omega|} \sum_{i \in \Omega} \left[s_i + \tilde{v}_i - \sum_{j \in \mathcal{N}_i} s_j P_{c+}(i \rightarrow j) \right]^2, \\ \mathcal{L}_- &= \frac{1}{|\Omega|} \sum_{i \in \Omega} \left[s_i - \tilde{v}_i - \sum_{j \in \mathcal{N}_i} s_j P_{c-}(i \leftarrow j) \right]^2, \\ \mathcal{L}_{\text{Pearson}} &= -(\lambda_u \text{corr}(\tilde{v}_i, u_i) + \lambda_s \text{corr}(\tilde{v}_i, -s_i)), \end{aligned} \quad (2)$$

where Ω is the set of all cells, \mathcal{N}_i is the estimated nearest neighborhood of cell i , and \tilde{v}_i is the estimated velocity vector for the spliced RNA in cell i . $P_{c+}(i \rightarrow j)$ is the normalized binary indicator of cell i ’s neighboring cells j with $S_{\cos}(s_j - s_i, \tilde{v}_i) > 0$, indicating that cells j are the downstream cells of cell i according to the estimated velocity \tilde{v}_i . Similarly, $P_{c-}(i \leftarrow j)$ is the normalized binary indicator of cell i ’s neighboring cells j with $S_{\cos}(s_j - s_i, \tilde{v}_i) < 0$, indicating that cells j are the upstream cells of cell i according to \tilde{v}_i . These binary indicators are normalized across the neighborhood to form probability distributions.

It has been mentioned in the original DeepVelo paper³ that the first two terms in Eq.(2) are symmetric to the sign of \tilde{v}_i , i.e., $\mathcal{L}_+(\tilde{v}_i) + \mathcal{L}_-(\tilde{v}_i) = \mathcal{L}_+(-\tilde{v}_i) + \mathcal{L}_-(-\tilde{v}_i)$. The sign of \tilde{v}_i is determined by the third term, $\mathcal{L}_{\text{Pearson}}$. However, the design of $\mathcal{L}_{\text{Pearson}}$ may introduce bias to genes where cells are all in the induction/repression phase. Here we take dataset *Gastrulation (Erythroid)* as an example.

First, we compared the velocity stream plots inferred by DeepVelo and TIVelo. DeepVelo exhibits a backward velocity flow from Erythroid3 to Erythroid2, as shown in Fig.R11(a) (blue box). This result was generated using DeepVelo’s default parameter settings (<https://github.com/bowang-lab/DeepVelo/blob/main/README.md>). However, as shown in Fig.R12, the result presented in Fig.S2 of the DeepVelo paper³ is different and it required a specific training adjustment: the removal of $\mathcal{L}_{\text{Pearson}}$ from the loss function after 10 training epochs (https://github.com/bowang-lab/DeepVelo/blob/main/examples/mouse_gastrulation.py). This adjustment was not applied in the analysis of the other five datasets discussed in DeepVelo’s paper: $\mathcal{L}_{\text{Pearson}}$ is retained throughout the training process.

Fig. R11 | The comparison of velocity stream plots inferred by DeepVelo and TIVelo in *Gastrulation (Erythroid)*. Blue box: backward velocity stream estimated by DeepVelo.

Fig. R12 | The comparison of velocity stream plots inferred by DeepVelo, with default parameter setting and with $\mathcal{L}_{\text{Pearson}}$ removed from the loss function after 10 training epochs. (b) is from Fig.S2 of DeepVelo’s paper³.

To further investigate the impact of including $\mathcal{L}_{\text{Pearson}}$ and why the default setting of DeepVelo fails in this dataset, we inspected three genes where all cells are in the **repression phase**, where the

velocities for spliced RNA are expected to be negative for all cells. For each gene g , we visualized the fitted velocities $\tilde{v}_i^{(g)}$ for spliced RNA given by DeepVelo for a subset of sampled cells i . From the results shown in Fig.R13, cells i with high u_i and low s_i tend to have positive $\tilde{v}_i^{(g)}$ (green box), while cells i with low u_i and high s_i tend to have negative $\tilde{v}_i^{(g)}$ (yellow box).

This behavior arises from the design of the $\mathcal{L}_{\text{Pearson}}$ term in the loss function (2). As a result of this term, cells with high u_i and low s_i tend to have positive velocities, and cells with low u_i and high s_i tend to have negative velocities, regardless of the transcriptional phase of cells. This is particularly problematic for genes with all cells entirely in the repression or induction phase, where velocities should be negative or positive across all cells, respectively. This inherent limitation in the design of $\mathcal{L}_{\text{Pearson}}$ in DeepVelo leads to the backward velocity stream from Erythroid3 and Erythroid2 observed in Fig.R11(a) (blue box).

Fig. R13 | The velocity inferred by DeepVelo for three genes in *Gastrulation* (*Erythroid*). Here only the velocities for spliced RNA (\tilde{v}_i) are shown. Green box: cells with positive \tilde{v}_i due to high u_i and low s_i . Yellow box: cells with negative \tilde{v}_i due to low u_i and high s_i .

In contrast, TIVelo does not have the inherent constraints imposed by $\mathcal{L}_{\text{Pearson}}$ in DeepVelo, allowing genes with all cells in the repression (or induction) phase to exhibit negative (or positive) velocities across all cells, as shown in Fig.R14.

Fig. R14 | The velocity inferred by TIVelo for three genes in *Gastrulation* (*Erythroid*). Here only the velocities for spliced RNA (\tilde{v}_i) are shown.

cellDancer

cellDancer trains a deep neural network (DNN) Φ_{θ^g} for each gene g independently to estimate cell-specific kinetic rates $(\alpha_i^g, \beta_i^g, \gamma_i^g)$ for cell i :

$$(\alpha_i^g, \beta_i^g, \gamma_i^g)^T = \Phi_{\theta^g}(u_i^g, s_i^g) \quad (3)$$

For each gene g , the loss function is designed as follows:

$$\mathcal{L} = \sum_{i=1}^n \mathcal{L}_i \quad (4)$$

$$\mathcal{L}_i = 1 - \max_{\{i'\}} \frac{v_i \cdot v_{i'}}{|v_i| \cdot |v_{i'}|}$$

where $v_i = (v_i^u, v_i^s)$, $v_i^u = \alpha_i^g - \beta_i^g u_i^g$, $v_i^s = \beta_i^g u_i^g - \gamma_i^g s_i^g$. Cell i' is in the nearest neighborhood of cell i and $v_{i'} = (u_{i'}^g - u_i^g, s_{i'}^g - s_i^g)$.

For each gene, cellDancer minimizes \mathcal{L}_i by selecting a cell i' in the nearest neighborhood of cell i . However, this neighboring cell i' can vary across different genes for the same cell i , potentially leading to errors in velocity direction estimation. This issue is illustrated using the dataset *Intestinal Organoid*.

First, we compared the velocity stream plots inferred by cellDancer and TIVelo. In *Intestinal Organoid*, the Stem cells differentiate into Goblet cells and Paneth cells in the Secretory lineage, and differentiate into Enterocytes in the Enterocytes lineage (Fig.R15(a)). cellDancer's velocity estimation fails to accurately capture the differentiation trajectory from Stem cells into two distinct lineages, as shown in Fig.R15(b) (red boxes).

Fig. R15 | The comparison of velocity stream plots inferred by TIVelo and cellDancer in *Intestinal Organoid*.

To further investigate this issue of cellDancer, we inspected six genes in this dataset, and visualized the velocities (v_i^u, v_i^s) estimated by cellDancer. For the first three genes, the estimated velocities indicate a reversed direction of differentiation from Enterocytes to Stem cells, as shown in Fig.R16. We also inspected a randomly selected subset of five cells and, for each cell i , identified its corresponding neighboring cell i' for each gene as defined in Eq.(4). In Eq.(4), the selected cell i' should be the downstream cells to cell i in differentiation. The results presented in Fig.R17 reveal that for the first three genes, cellDancer tends to select the upstream cells of cell i as i' , while for the last three genes, cellDancer tends to select cell i' from the downstream cells of cell i . The error in velocity estimation for such genes arises from the error in neighboring cell selection in cellDancer.

Fig. R16 | The velocity inferred by cellDancer for six genes in *Intestinal Organoid*. Here both the velocities for unspliced RNA and for spliced RNA are shown.

Fig. R17 | Velocity estimation from cellDancer and the cells i' for five randomly selected cells i . Red points: selected cells i . Green points: the cells i' for corresponding cell i .

This error in the selection of neighboring cell i' for the top three genes arises from the flexible design of cellDancer's loss function (4), which does not guarantee that cell i' is selected from biologically plausible downstream cells of cell i . Consequently, the neighboring cell i' may be selected from the upstream cells of cell i , as demonstrated in Fig. R17 (top row).

Fig. R18 | Velocity estimation $v_i/-v_i$ and cell i' selected based on $v_i/-v_i$ for one randomly selected cell i . Red point: selected cell i . Green points: cell i' for cell i . Light green points: neighboring cells of cell i .

The design of L_i in Eq.(4) in cellDancer is too flexible with respect to the direction of v_i . Even if v_i is inverted or incorrect, we may still find a neighboring cell i' that makes L_i small. To further illustrate this limitation introduced by L_i in Eq.(4), we inverted the direction of inferred velocities from cellDancer for one randomly selected cell i and a gene *Gsto1*, i.e., setting v_i to $-v_i$. We then selected cell i' for both v_i and $-v_i$, and compared the cosine similarity calculated in Eq.(4) based on v_i and $-v_i$. Notably, the cosine similarity for $-v_i$ is higher than that for v_i , leading to an even lower loss L_i compared to that with v_i , as shown in Fig.R18. This demonstrates that the design of loss L_i in cellDancer is too flexible, and regardless of the direction of v_i , there may exist a neighboring cell i' that makes L_i small.

Fig. R19 | Velocity estimation from TIVelo and the directed nearest neighborhood (dNN) for five randomly selected cell. Red point: selected cell i . Green points: dNN for cell i .

In the contrary, in TIVelo, we selected the same directed nearest neighborhood $dnn(i)$ for each cell i across different genes (Fig.R19), based on the overall direction on the main path. This strategy provides constraints on the inferred velocity of individual genes, ensuring that the velocity will point to the downstream cells (downstream cells can be in the same cell cluster as cell i or in the downstream cluster), bypassing the limitation of L_i of Eq.(4) in cellDancer. In Fig.R19, each blue arrow represents the velocity for each selected cell and each gene, which is obtained based on the ensemble of neighboring cells in the dNN of the selected cell. In some cases, the velocity does not point towards the mean expression value of cells in the dNN, which is due to the regularization term in our loss objective to enhance the consistency of velocity vector of similar cells (Methods).

The discussion on this topic is now presented in Supplementary Notes S1 of the revised manuscript.

2. TIVelo adopts a distinct approach to RNA velocity estimation by first inferring trajectories and pseudotime. It calculates velocities between neighboring cells, rather than directly modeling gene-specific dynamics. In contrast, other methods typically model the transcriptional dynamics of each gene individually. As a result, TIVelo appears more as a trajectory inference method with automatic initial state detection, with RNA velocity calculation being a downstream application. Intuitively, this strategy may lead to smoother velocity streams generated by TIVelo. Therefore, I am concerned that the metrics shown in Fig. 7, which tend to favor smoother streams, may introduce a bias toward TIVelo.

Response: We sincerely thank the reviewer for their constructive comments.

Firstly, the two metrics we used in Fig.7, namely cross-boundary direction correctness (CBDir) and transition cosine similarity (TransCosine), evaluate whether velocity streams from cell n flow to downstream clusters of cell n . CBDir and TransCosine do not favor the velocity streams produced by TIVelo: the velocity of cell n in TIVelo may not only point towards cells in the downstream clusters, but also cells in the same cluster as cell n . The velocity of cell n in TIVelo is obtained based on the ensemble of neighboring cells in the directed nearest neighborhood (dNN), which signifies cells in the future state of cell n within its k -nearest neighborhood (k -NN). The dNN of cell n may contain cells in the same cluster as cell n and also cells in the downstream clusters of cell n . Specifically, cell n' is in the dNN of cell n if n' satisfies the following three conditions at the same time:

- $n' \in \mathcal{N}_n$, where \mathcal{N}_n is the nearest neighborhood (based on gene expression similarity) of cell n ;
- $t_n < t_{n'}$, where t_n is the diffusion pseudotime for cell n , which indicates that cell n' is potentially a future state of cell n ;
- $i' \in \text{child}(i)$ or $i = i'$, where clusters i and i' are the clusters that cells n and n' belong to, respectively. $\text{child}(i)$ is the downstream clusters of i .

By incorporating gene expression similarity and diffusion pseudotime into the selection of dNN, TIVelo avoids generating velocity streams that merely point toward cells in downstream clusters of cell n . Thus the metrics CBDir and TransCosine do not favor the velocity stream generated from TIVelo.

Secondly, in Fig.7, except for CBDir and TransCosine, we also used velocity coherence⁵ (VeloCoh) to measure the performance of different methods, which does not depend on the downstream cell clusters of each cell. The velocity v_n of a given cell n is coherent if it points in the same direction as the empirical displacement $[\tilde{\mathbf{I}}\mathbf{S}]_{n,:} - \mathbf{s}_n$, where $\tilde{\mathbf{I}}$ is the cell-cell transitions probability matrix based on the inferred velocity. Furthermore, in Fig.5, we utilized sign accuracy to evaluate whether the inferred velocities agree with the reference cell cycle scores in two FUCCI datasets, which also does not depend on the downstream cell clusters of each cell. These diverse metrics collectively validate TIVelo’s superiority over other methods from multiple perspectives, ensuring a robust and unbiased evaluation.

3. Currently, the authors focus on demonstrating that the trajectory and initial state inferred by

TIVelo accurately fit the ground truth process across all datasets. However, it would strengthen the contribution of this study and make the paper more compelling if the authors could present some scientific findings using TIVelo. Specifically, given that velocity estimation is the final step of the framework, it would be valuable to see how the inferred velocities can benefit biological studies.

Response: We sincerely thank the reviewer for their constructive feedback. To demonstrate how TIVelo's inferred velocities can contribute to biological studies, we present an analysis of the *Intestinal Organoid* dataset, highlighting key scientific insights enabled by TIVelo.

Fate Probability Visualization

Fate probability analysis enables the quantification of each cell's likelihood to differentiate into specific terminal states. We calculated the fate probabilities of each cell in *Intestinal Organoid* based on TIVelo's inferred velocities⁶. These probabilities were visualized using circular projections⁷ (Fig.R20), where each terminal state was arranged at a vertex of an equilateral triangle, and cells are positioned inside the triangle according to their fate probabilities. The spatial proximity of a cell to a particular vertex directly reflects its tendency to differentiate towards the corresponding terminal state. This visualization provides a comprehensive overview of cell fate decisions during differentiation.

Fig. R20 | Fate probabilities visualization towards terminal states via circular projections⁷, where each terminal state was arranged at a vertex of an equilateral triangle, and cells are positioned inside the triangle according to their fate probabilities.

Lineage-Specific Driver Gene Identification

The *Intestinal Organoid* dataset features two distinct lineages: the Secretory lineage and the Enterocytes lineage. Based on the velocities inferred by TIVelo, we identified driver genes associated with each lineage⁶. In Fig.R21 and Fig.R22, we show some candidate driver genes for both lineages, along with their expression patterns across all cells.

To further validate these findings, we analyzed the correlation between gene expression and fate probabilities of each lineage (across cells) for genes in the data⁶. As shown in Fig.R23, genes such as *Agr2*, *Galnt7*, *Creb3l1* and *Muc13* exhibit strong correlations with the Secretory lineage, while genes like *Aldob*, *Dmbt1*, *Myo1a* and *Lgals4* are highly correlated with the Enterocytes lineage. The functions of those identified driver genes for the corresponding lineage are supported by literature references, summarized in Table R6.

Fig. R21 | Selected driver genes for Secretory lineage (red boxes) and their expression, identified based on TIVelo's inferred velocity.

Fig. R22 | Selected driver genes for Enterocytes lineage (red boxes) and their expression, identified based on TIVelo's inferred velocity.

Fig. R23 | The correlation of the gene expression with fate probabilities (of two lineages) for genes in *Intestinal Organoid*.

Gene Name	Reference	Lineage
Agr2	Park et al., 2009 ⁸ ; Zhao et al., 2010 ⁹	Secretory
Galnt7	Bennett et al., 2012 ¹⁰	Secretory
Creb3l1	Fox et al., 2010 ¹¹	Secretory
Muc13	Sheng et al., 2011 ¹²	Secretory
Aldob	Gao et al., 2023 ¹³	Enterocytes
Dmbt1	Kaemmereret al., 2012 ¹⁴	Enterocytes
Myo1a	Mcconnell et al., 2007 ¹⁵	Enterocytes
Lgals4	Cao et al., 2016 ¹⁶	Enterocytes

Table R6 | Some driver genes identified based on TIVelo’s inferred velocity for both lineages, with references validating their functions for the corresponding lineage.

Identification of Macrostates and Functional Insights

In the *Intestinal Organoid* dataset, we identified macrostates proposed in CellRank¹⁷ to delineate metastable states within the Markov chain of cellular transitions. These macrostates represent cell populations that maintain relative stability during the differentiation process. Based on the velocities inferred by TIVelo⁶, we identified a distinct macrostate within the "Stem cell" cluster, and subsequent pathway analysis revealed significant enrichment of the Ribosome pathway in this macrostate population.

Fig. R24 | Left: differential gene expression analysis for macrostate we identified in "Stem cell" cluster against the rest stem cells. Right: KEGG analysis for differential expressed genes of macrostate "Stem cells 2" against the rest stem cells.

Differential gene expression analysis of this macrostate compared to other stem cells revealed significant upregulation of ribosome-related genes (Fig.R24(a)). KEGG pathway enrichment analysis further confirmed the enrichment of the Ribosome pathway (Fig.R24(b)), suggesting that ribosome activity plays a critical role in intestinal stem cell (ISC) differentiation. This aligns with recent findings that ribosomes act as nutrient sensors, enabling ISCs to adapt to their local nutrient environment¹⁸.

Kinetic Rates Inference via TIVelo’s New Mode

To provide more biological insights, we developed a new mode (**kinetic rate mode**) for fitting RNA velocity in TIVelo, which enables simultaneous inference of cell-specific kinetic rates (α , β and γ) for

each gene, while maintaining the framework’s core functionality. This reveals the cellular dynamics of the transcriptional phase for individual genes.

In our original manuscript, once obtaining the directed nearest neighborhood (dNN) of each cell, the velocity of cell n is obtained based on the ensemble of neighboring cells in the directed nearest neighborhood (dNN). That is equivalent to minimizing the following objective L_g for each gene g :

$$\begin{aligned}
 L_{u,g} &= \sum_{n=1}^N \|u_{n,g} + \tilde{v}_{n,g}^{(u)} - \frac{1}{N_n} \sum_{n' \in dnn(n)} u_{n',g}\|^2, \\
 L_{s,g} &= \sum_{n=1}^N \|s_{n,g} + \tilde{v}_{n,g}^{(s)} - \frac{1}{N_n} \sum_{n' \in dnn(n)} s_{n',g}\|^2, \\
 L_g &= L_{u,g} + \lambda L_{s,g},
 \end{aligned} \tag{5}$$

where $dnn(n)$ is the directed nearest neighborhood of cell n , N_n is the number of cells in $dnn(n)$, and λ is a hyperparameter controlling the weight of loss from u and s , which is set as 1 by default. $\tilde{v}_{n,g}^{(u)}$ and $\tilde{v}_{n,g}^{(s)}$ are inferred velocity for u and s of cell n and gene g .

In kinetic rate mode, instead of updating $\tilde{v}_{n,g}^{(u)}$ and $\tilde{v}_{n,g}^{(s)}$ directly, we calculate the inferred velocity $\tilde{v}_{n,g}^{(u)}$ and $\tilde{v}_{n,g}^{(s)}$ by cell-specific kinetic rates $\alpha_{n,g}$, $\beta_{n,g}$ and $\gamma_{n,g}$ as follows:

$$\begin{aligned}
 \tilde{v}_{n,g}^{(u)} &= \alpha_{n,g} - \beta_{n,g} u_{n,g}, \\
 \tilde{v}_{n,g}^{(s)} &= \beta_{n,g} u_{n,g} - \gamma_{n,g} s_{n,g}.
 \end{aligned} \tag{6}$$

Furthermore, we infer the kinetic rates $\alpha_{n,g}$, $\beta_{n,g}$ and $\gamma_{n,g}$ from unspliced and spliced expressions as follows:

$$(\alpha_n^\top, \beta_n^\top, \gamma_n^\top)^\top = f_\theta([\mathbf{u}_n^\top, \mathbf{s}_n^\top]^\top), \tag{7}$$

where f_θ is fully connected layers with parameters θ , α_n , β_n and γ_n are G -dimensional kinetic rates for cell n across all G genes, and \mathbf{u}_n and \mathbf{s}_n are G -dimensional unspliced and spliced expressions of cell n . This approach provides natural regularization, ensuring cells with similar expression profiles have comparable kinetic rates.

Fig. R25 | The comparison of velocity stream plots inferred by TIVelo and TIVelo (kinetic rate mode) in *Intestinal Organoid*.

By default, we use a f_θ with three layers in Eq.(7). The input dimensions, (two) hidden dimensions and output dimensions of f are $2G$, (256, 64) and $3G$, respectively. We further calculate the inferred

velocities $\tilde{v}_{n,g}^{(u)}$ and $\tilde{v}_{n,g}^{(s)}$ by Eq.(6), and minimize the objective L_g in Eq.(5) with respect to θ . The model is trained for 300 epochs with a learning rate 0.001.

Fig.R25 shows the comparison of velocity stream plots inferred by TIVelo and TIVelo (kinetic rate mode). The result inferred by TIVelo (kinetic rate mode) closely matches those from TIVelo in our original manuscript. This indicates that TIVelo (kinetic rate mode) does not compromise velocity accuracy while providing additional biological information about the kinetic rates in cell dynamics.

The inferred rate parameters α , β and γ reveal the transcriptional phase of individual genes, and we take two driver genes for Enterocytes lineage in this dataset as examples. As shown in Fig.R26, for genes *Aldob* and *Dmbt1*, there is a rapid increase of α and β rates for cells in the Enterocytes lineage. This indicates that such cells are in the active transcription phases of their corresponding driver genes, providing additional biological insights for cell dynamics.

Fig.R26 | Cell-specific kinetic rates α , β and γ inferred by TIVelo, for two driver genes of Enterocytes lineage. Red boxes: the rapid growth of α and β for cells in the Enterocytes lineage.

The discussion on this topic is now presented in Supplementary Notes S2 of the revised manuscript.

4. In line 437, that terminal cells are identified using `scvelo.tl.terminal_states`, which assumes that terminal cells exhibit extreme values of unspliced and spliced RNA. This assumption may not hold in complex or multi-branch datasets. Could the authors provide more details on how terminal states were modeled and optimized for such cases? For instance, I recommend provide similar details shown in Figure 1 (e.g., initialization of terminal states, pseudotime, orientation scores, main paths, etc.) for more datasets, such as HSPCs and dentate gyrus. This could provide a clearer understanding of how TIVelo works across diverse scenarios.

Response: We sincerely thank the reviewer for their thoughtful questions regarding the identification of terminal states in TIVelo. In our framework, terminal states (root cluster or end clusters) are identified using `scvelo.tl.terminal_states`¹⁹. The origin node is selected as the cell cluster exhibiting the strongest signal as the root cluster or the end cluster among the inferred terminal states.

The sentence stated in our original manuscript was: terminal (root/end) cells typically exhibit extreme values in u and s , making them easily identified by ODE-based methods such as scVelo. We would like to clarify that the actual procedure for our origin node selection is more nuanced than merely based on extreme values in u and s and involves the following steps:

Firstly, it calculates the transition matrix $\pi_{c,c'} = \cos \angle(\delta_{c,c'}, v_c)$, where v_c is the inferred velocity of cell c , $\delta_{c,c'} = s_{c'} - s_c$ and s_c is the spliced expression of cell c . $\pi_{c,c'}$ is normalized by

$$\tilde{\pi}_{c,c'} = \frac{1}{z_c} \exp\left(\frac{\pi_{c,c'}}{\sigma_c^2}\right) \quad (8)$$

with row normalization factors $z_c = \sum_{c'} \exp(\frac{\pi_{c,c'}}{\sigma_c^2})$ and kernel width parameters σ_c optionally adjusted for each cell locally.

Secondly, the end and root clusters are obtained as stationary states of the velocity-inferred transition matrix $\tilde{\pi}$ and its transpose $\tilde{\pi}^T$, respectively. A root score vector μ^{root} and an end score vector μ^{end} will be calculated for all cells, which is given by left eigenvectors corresponding to an eigenvalue of 1, that is

$$\mu^{\text{end}} = \mu^{\text{end}} \tilde{\pi}, \quad \mu^{\text{root}} = \mu^{\text{root}} \tilde{\pi}^T. \quad (9)$$

An example of inferred μ^{root} and μ^{end} by scVelo for dataset *HSPCs* is shown in Fig.R27.

Fig. R27 | Root score (μ^{root}) and end score (μ^{end}) for *HSPCs* inferred by scVelo.

Finally, the cell cluster with the largest mean root score will be selected as the origin node. If there is no cell cluster with a root score larger than a threshold (0.1), we select the one with the largest mean end score as the origin node (Methods).

For multi-branch datasets with multiple expected end clusters, this strategy for selecting the origin node remains effective, as long as one of the end clusters or the root cluster is selected as the origin node. We illustrate this using the *Intestinal Organoid* dataset with two lineages.

As shown in Fig.R28(a), the origin node is selected based on the root score. In this dataset, two cell clusters (Enterocytes and Paneth cells) exhibit relatively high root scores (red boxes). Regardless of whether the Enterocytes or Paneth cell cluster is selected as the origin node, TIVelo produces the same (and correct) velocity inference (Fig.R28(b)). Depending on whether the Enterocytes or Paneth cell is selected as the origin node, different main paths are inferred, as demonstrated in Fig.R29(a) and (b). The mean orientation scores inferred on both main paths are negative, and TIVelo correctly infers the velocity direction along both main paths.

Fig. R28 | (a) Root score (μ^{root}) of *Intestinal Organoid* inferred by scVelo. (b) Velocity stream plot inferred by TIVelo, based on either Enterocytes or Paneth cell selected as the origin node.

Fig. R29 | The main path selection in *Intestinal Organoid*. Main path selection 1 is based on the case when Enterocytes is selected as the origin node, while main path selection 2 is based on the case when Paneth cells is selected as the origin node.

While `scvelo.tl.terminal_states` does not directly select cell clusters based on extreme u and s values, we observed that the origin node selected by `scvelo.tl.terminal_states` often exhibits extreme values of unspliced and spliced RNA. We performed the following analysis for several multi-branch datasets, to demonstrate that the origin node selected exhibits a relatively high or low expression level of spliced RNA in the data:

1. For each gene g , we calculated the mean expression of s for each cell cluster i , denoted as $s_{i,g}$;
2. For each gene g and each cell cluster i , we calculated the relative expression as $\tilde{s}_{i,g} = s_{i,g}/s_{g,\max}$, where $s_{g,\max}$ are the maximum expression values of s in gene g .
3. We visualized the distribution of $\tilde{s}_{i,g}$ using violin plots for each cell cluster i .

The results shown in Fig.R30 confirm that origin nodes typically exhibit extreme (high or low) relative s expression levels compared to other cell clusters.

Fig. R30 | The violin plot of relative expression of s for each cell cluster in several multi-branch datasets. The violin for the origin node is marked by darker color. The median values of each violin are annotated.

Consistent with the theory of transcriptional kinetics, terminal states (root or end clusters) typically occupy the initiation or termination phases of transcriptional processes for most genes. Therefore, these states tend to exhibit extreme relative expression levels of s , which can be easily identified using `scvelo.tl.terminal_states`.

In our revised manuscript, Fig.1(b) shows an example to select the origin node based on the root/end score. More details for selecting the origin node are presented in the Supplementary Notes S4.

5. Fig 2k. The arrows in "Blood progenitors 1" do not appear to point toward the next cluster, "Blood progenitors 2." Could the authors clarify if this result is accurate?

Response: We sincerely thank the reviewer for pointing out this observation regarding Fig.2k. We acknowledge that the velocity arrows in "Blood progenitors 1" do not appear to point toward the next cluster, "Blood progenitors 2," as expected. This issue arises from the challenge of inferring velocity direction within the root cluster. While TIVelo effectively infers velocity directions between clusters, the directionality within the root cluster can sometimes be ambiguous.

To address this, we have implemented an improved strategy for root cell selection in the revised manuscript. Specifically, we now identify the cell within the inferred root cluster that has the longest distance (in the UMAP space) to the other clusters as the root cell. This adjustment ensures a more biologically plausible velocity stream, as demonstrated in Fig.R31, where the arrows in "Blood progenitors 1" now correctly point toward "Blood progenitors 2."

Fig. R31 | The comparison of velocity stream plots inferred by TIVelo with/without root cell selection in *Gastrulation (Erythroid)*.

Fig. R32 | The comparison of velocity stream plots inferred by TIVelo with/without root cell selection in *RPE1-FUCCI*. The numbers and colors represent different cell clusters.

After we implement this revised strategy for root cell selection, the result of the *RPE1-FUCCI* dataset

is also affected, and the revised velocity stream plots are included in Fig.5 of the manuscript (Fig.R32). This refinement enhances the accuracy and interpretability of TIVelo’s velocity inference, particularly within root clusters.

The discussion on this topic is now presented in Supplementary Notes S8 of the revised manuscript.

6. Fig 5b. The arrows in cluster 1 lack coherence in their orientation. How do these results reflect the advantages of TIVelo compared to other methods? Additional explanation or alternative visualizations would be helpful.

Response: We sincerely thank the reviewer for raising this point. We acknowledge that the arrows in the upper and lower regions of cluster 1 exhibit different orientations, which may initially appear inconsistent. However, this directional pattern is not indicative of an inconsistency in TIVelo’s velocity estimation but may rather reflect the underlying biological dynamics.

Firstly, cell cycle scores support the velocity stream of TIVelo in cluster 1. As demonstrated in Fig.R33(b), which focuses specifically on cluster 1 cells with relatively large UMAP2 components, the cell cycle scores clearly indicate that cells in the upper region exhibit higher scores compared to those in the lower region. This supports the biological plausibility of the upward velocity direction observed in the upper region of cluster 1 (see Fig.R33(a)).

Fig. R33 | (a) Velocity stream plots for *RPE1-FUCCI*, inferred by TIVelo. The numbers and colors represent different cell clusters. (b) Cell cycle scores of *RPE1-FUCCI* cells, highlighting cluster 1 cells with relatively large UMAP2 components.

Fig. R34 | Velocity stream plots for *RPE1-FUCCI*, inferred by UniTVelo and DeepVelo.

Secondly, similar directional patterns in cluster 1 are also observed in the velocity stream plots inferred by UniTVelo and DeepVelo, as demonstrated in Fig.R34.

Lastly, while the directional pattern in cluster 1 is less pronounced in the velocity stream plot of veloVI⁵ (Fig.R35(a)), plotting the velocity vectors longer reveals a similar trend (Fig.R35(b)).

Fig. R35 | (a) Velocity stream plots for *RPE1-FUCCI*, inferred by veloVI. (b) Inferred RNA velocity vectors from veloVI projected onto 2D UMAP. Only the second velocity component for cells with relatively large UMAP2 components (highlighted in red) is shown. The length of the velocity vectors has been scaled up for clarity.

The discussion on this topic is now presented in Supplementary Notes S3 of the revised manuscript.

7. The absence of installation documentation and tutorials in the GitHub repository makes it challenging for users to apply the code. I recommend including a clear installation guide and example usage tutorials to enhance reproducibility and accessibility.

Remarks on code availability: The absence of installation documentation and tutorials in the GitHub repository makes it challenging for users to apply the code. I recommend including a clear installation guide and example usage tutorials to enhance reproducibility and accessibility.

Response: We sincerely thank the reviewer for highlighting this important issue. To improve the accessibility and reproducibility of TIVelo, we have updated our GitHub repository to include comprehensive installation documentation and example usage tutorials. These resources are now available at: <https://github.com/cuhklinlab/TIVelo>.

Additionally, detailed documentation and tutorials can also be found on our Read the Docs page: <https://tiveloc.readthedocs.io/en/latest/>. We hope these updates will make it easier for users to install and apply TIVelo to their own datasets.

Replies to Reviewer 4

The manuscript introduces TIVelo, a cluster-level tree-based method for estimating RNA velocity under the premise that unspliced RNA is invariably expressed and repressed earlier than spliced RNA. Although this approach applies an interesting perspective by linking unspliced and spliced dynamics with a cluster-level trajectory tree, its novelty appears limited. The current study does not compellingly demonstrate the potential of TIVelo to yield groundbreaking insights or robust practical applications. Moreover, without ODE assumptions, claiming TIVelo is solving RNA velocity may not be accurate. By avoiding the estimation of kinetic parameters (e.g., transcription, splicing, and degradation rates), the method cannot provide quantitative insights that are typically essential for understanding RNA velocity and cellular dynamics.

Response: We sincerely appreciate the reviewer’s thoughtful and constructive evaluation of our manuscript. Their valuable comments have helped us strengthen both the methods and biological applications of our study. We have addressed each key concern through additional analyses and improvements:

Novelty of TIVelo

We further demonstrated TIVelo’s novelty by comparing TIVelo with two other methods mentioned by the reviewer: DeepVelo and cellDancer, which infer variable rate parameters across cells. Please see our detailed response to **reviewer 4’s second concern** for these results.

Insights Provided by TIVelo

To demonstrate how TIVelo can provide insights for biological studies, we carried out several downstream analyses based on TIVelo’s inferred velocities, including fate probability visualization, lineage-specific driver gene identification, and identification of macrostates with functional insights. Please see our detailed response to **reviewer 3’s third concern** for these results.

Robust Practical Applications

To validate the robustness of TIVelo in practical applications, we carried out several experiments, including the evaluation of TIVelo’s performance on the perturbed main path, and on the main path with reversed direction. Furthermore, we enhanced TIVelo’s robustness with respect to the origin node through the refinement of the origin node selection strategy. Please see our detailed response to **reviewer 1’s first & second concerns** for these results.

Kinetic Parameters Estimation

We developed a new mode (**kinetic rate mode**) for fitting RNA velocity in TIVelo, which enables simultaneous inference of cell-specific kinetic rates (α , β and γ) for each gene, while maintaining the framework’s core functionality. This reveals the cellular dynamics of the transcriptional phase for individual genes.

In our original manuscript, once obtaining the directed nearest neighborhood (dNN) of each cell, the velocity of cell n is obtained based on the ensemble of neighboring cells in the directed nearest neighborhood (dNN). That is equivalent to minimizing the following objective L_g for each gene g :

$$\begin{aligned} L_{u,g} &= \sum_{n=1}^N \|u_{n,g} + \tilde{v}_{n,g}^{(u)} - \frac{1}{N_n} \sum_{n' \in dnn(n)} u_{n',g}\|^2, \\ L_{s,g} &= \sum_{n=1}^N \|s_{n,g} + \tilde{v}_{n,g}^{(s)} - \frac{1}{N_n} \sum_{n' \in dnn(n)} s_{n',g}\|^2, \\ L_g &= L_{u,g} + \lambda L_{s,g}, \end{aligned} \tag{10}$$

where $dnn(n)$ is the directed nearest neighborhood of cell n , N_n is the number of cells in $dnn(n)$, and λ is a hyperparameter controlling the weight of loss from u and s , which is set as 1 by default. $\tilde{v}_{n,g}^{(u)}$

and $\tilde{v}_{n,g}^{(s)}$ are inferred velocity for u and s of cell n and gene g .

In kinetic rate mode, instead of updating $\tilde{v}_{n,g}^{(u)}$ and $\tilde{v}_{n,g}^{(s)}$ directly, we calculate the inferred velocity $\tilde{v}_{n,g}^{(u)}$ and $\tilde{v}_{n,g}^{(s)}$ by cell-specific kinetic rates $\alpha_{n,g}$, $\beta_{n,g}$ and $\gamma_{n,g}$ as follows:

$$\begin{aligned}\tilde{v}_{n,g}^{(u)} &= \alpha_{n,g} - \beta_{n,g}u_{n,g}, \\ \tilde{v}_{n,g}^{(s)} &= \beta_{n,g}u_{n,g} - \gamma_{n,g}s_{n,g}.\end{aligned}\tag{11}$$

Furthermore, we infer the kinetic rates $\alpha_{n,g}$, $\beta_{n,g}$ and $\gamma_{n,g}$ from unspliced and spliced expressions as follows:

$$(\boldsymbol{\alpha}_n^\top, \boldsymbol{\beta}_n^\top, \boldsymbol{\gamma}_n^\top)^\top = f_\theta([\mathbf{u}_n^\top, \mathbf{s}_n^\top]^\top),\tag{12}$$

where f_θ is fully connected layers with parameters θ , $\boldsymbol{\alpha}_n$, $\boldsymbol{\beta}_n$ and $\boldsymbol{\gamma}_n$ are G -dimensional kinetic rates for cell n across all G genes, and \mathbf{u}_n and \mathbf{s}_n are G -dimensional unspliced and spliced expressions of cell n . This approach provides natural regularization, ensuring cells with similar expression profiles have comparable kinetic rates.

By default, we use a f_θ with three layers in Eq.(12). The input dimensions, (two) hidden dimensions and output dimensions of f are $2G$, (256, 64) and $3G$, respectively. We further calculate the inferred velocities $\tilde{v}_{n,g}^{(u)}$ and $\tilde{v}_{n,g}^{(s)}$ by Eq.(11), and minimize the objective L_g in Eq.(10) with respect to θ . The model is trained for 300 epochs with a learning rate 0.001.

Fig.R36 shows the comparison of velocity stream plots inferred by TIVelo and TIVelo (kinetic rate mode). The result inferred by TIVelo (kinetic rate mode) closely matches those from TIVelo in our original manuscript. This indicates that TIVelo (kinetic rate mode) does not compromise velocity accuracy while providing additional biological information about the kinetic rates in cell dynamics.

Fig. R36 | The comparison of velocity stream plots inferred by TIVelo and TIVelo (kinetic rate mode) in *Intestinal Organoid*.

The inferred rate parameters α , β and γ reveal the transcriptional phase of individual genes, and we take two driver genes for Enterocytes lineage in this dataset as examples. As shown in Fig.R37, for genes *Aldob* and *Dmbt1*, there is a rapid increase of α and β rates for cells in the Enterocytes lineage. This indicates that such cells are in the active transcription phases of their corresponding driver genes, providing additional biological insights for cell dynamics.

Fig. R37 | Cell-specific kinetic rates α , β and γ inferred by TIVelo, for two driver genes of Enterocytes lineage. Red boxes: the rapid growth of α and β for cells in the Enterocytes lineage.

1. Inconsistencies in Comparative Evaluations.

The comparative analyses provided for UniTVelo, veloVI, scVelo, and cellDancer frequently conflict with results reported in their respective original studies. For instance, UniTVelo’s directionality in the manuscript appears reversed compared to the original publication. Similar discrepancies arise in figures for scVelo (Fig. S1), cellDancer(Figs. S2, S3, S5), and again UniTVelo (Fig. 3, S7).

The authors must verify that all external methods were correctly implemented and that their parameters align with the original protocols. Such inconsistencies undermine the reliability of the study’s comparisons and conclusions.

Response: We sincerely thank the reviewer for raising this important concern regarding inconsistencies in the comparative evaluations. Below, we address the discrepancies and provide clarifications for each method.

scVelo

In Fig.S1 of our original manuscript, we produced the result for *Dentate Gyrus* using the same sets of hyperparameters following the instructions in the scVelo tutorial (<https://scvelo.readthedocs.io/en/stable/DynamicalModeling.html>). The result of scVelo (dynamical mode) reproduced by our experiment is slightly different from the one given in the tutorial page, as shown in Fig.R38 (red box). The observed differences may arise from inherent randomness in scVelo’s dynamical mode. Notably, a similar velocity backward pattern is observed in the reproduction of scVelo (dynamical mode) for *Dentate Gyrus* from DeepVelo, as shown in Fig.2(a) of the original DeepVelo paper³ (see Fig.R39).

(a) scVelo (dynamical mode) from our experiment (b) scVelo (dynamical mode) from the tutorial page

Fig. R38 | Comparison of velocity stream plots inferred by scVelo (dynamical mode) for *Dentate Gyrus* from our experiment and scVelo tutorial page (https://scvelo.readthedocs.io/en/stable/vignettes/Fig2_dentategyrus.html).

Fig. R39 | Fig.2(a) in DeepVelo paper³, showing the reproduction of scVelo (dynamical mode) for *Dentate Gyrus* from DeepVelo (right hand side). Red box: the velocity backward pattern similar to the results reproduced in our experiment (Fig.R38(a)).

veloVI

Similar to scVelo, our experimental reproduction of veloVI yields results that are slightly different from those presented in Fig.4 of the original veloVI paper (Fig.R40). These discrepancies may also stem from the inherent randomness of the veloVI algorithm.

Fig. R40 | Comparison of velocity stream plots inferred by veloVI for *Pancreas* from our experiment and Fig.4 of original veloVI paper.

cellDancer

The discrepancies between the results shown in our reproduction (manuscript Fig.S2, S3, and S5) and in cellDancer⁴ (Fig.4, Fig.2, Fig.3) stem from two key factors:

Firstly, we did not use the built-in visualization method provided by cellDancer, which was used in the original cellDancer paper⁴. Instead, for consistency across methods, we used the `scvelo.pl.velocity_embedding_stream` function in the scVelo package to visualize the results from all methods, including cellDancer. We note that the built-in visualization in cellDancer (`celldancer.cdplt.scatter_gene`) uses Bezier curves to generate smoother and more aesthetically pleasing arrows, which may contribute to the observed differences.

To provide additional context, we now include both visualization approaches from scVelo and cellDancer for all datasets in our manuscript. Comparisons of the two visualization methods for Figs.S2, S3, and S5 are shown in Fig.R41, Fig.R42, and Fig.R43, respectively.

(a) Velocity stream inferred by cellDancer, using visualization from scVelo (b) Velocity stream inferred by cellDancer, using cellDancer built-in visualization

Fig. R41 | Comparison of velocity stream plots inferred by cellDancer for *Pancreas*, through scVelo visualization and cellDancer built-in visualization, respectively.

(a) Velocity stream inferred by cellDancer, using visualization from scVelo (b) Velocity stream inferred by cellDancer, using cellDancer built-in visualization

Fig. R42 | Comparison of velocity stream plots inferred by cellDancer for *Gastrulation (Erythroid)*, through scVelo visualization and cellDancer built-in visualization, respectively.

(a) Velocity stream inferred by cellDancer, using visualization from scVelo (b) Velocity stream inferred by cellDancer, using cellDancer built-in visualization

Fig. R43 | Comparison of velocity stream plots inferred by cellDancer for *Dentate Gyrus 2*, through scVelo visualization and cellDancer built-in visualization, respectively.

Secondly, cellDancer did not use the same preprocessing procedure and hyperparameters in the model for different datasets in the tutorial (https://guangyuwanglab2021.github.io/cellDancer_website/index.html). Specifically,

1. Pancreas

- Using `permutation_ratio=0.5` in `celldancer.velocity` function, which is the sampling ratio of cells in each epoch when training each gene.
- Using 200 neighboring cells for calculating the transition probability matrix for each cell (which means `projection_neighbor_size=200` in `celldancer.compute_cell_velocity` function).

2. Gastrulation (Erythroid)

- Selecting cells from cell type haemato-endothelial progenitors, blood progenitors 1/2, and erythroid 1/2/3 in a large mouse gastrulation dataset²⁰, which is different from the more

commonly used smaller dataset (used by UniTVelo, DeepVelo, veloAE²¹, etc.), with cells selected from blood progenitors 1/2 and erythroid 1/2/3.

- Using 100 neighboring cells for first-moment calculation in preprocessing, (which means `n_neighbors=100` in `scvelo.pp.moments` function, instead of 30 by default).
- Using `permutation_ratio=0.125` in `celldancer.velocity` function.
- Using 10 neighboring cells for calculating the transition probability matrix for each cell.

3. Dentate Gyrus 2

- Without using `scvelo.pp.filter_and_normalize` to filter genes.
- Using `permutation_ratio=0.1` in `celldancer.velocity` function.
- Using 100 neighboring cells for calculating the transition probability matrix for each cell.

For a fair comparison, in our manuscript, we adopted the same preprocessing procedure and hyperparameters in the model for all datasets when we implemented cellDancer:

1. Using `scvelo.pp.filter_and_normalize` and `scvelo.pp.moments` for preprocessing with default arguments. Using "velocity genes" defined in UniTVelo for RNA velocity analysis.
2. Inferring RNA velocity for each cell following the instruction in the tutorial page for *Gastrulation (Erythroid)* (https://guangyuwanglab2021.github.io/cellDancer_website/notebooks/case_study_gastrulation.html):
 - Using `celldancer.velocity` with `permutation_ratio=0.125`.
 - Using `celldancer.compute_cell_velocity` with `projection_neighbor_size=10`.

For the three datasets *Pancreas*, *Gastrulation (Erythroid)* and *Dentate Gyrus 2*, we also reproduced the results by cellDancer, following the preprocessing procedure and the hyperparameter settings in cellDancer's tutorial. We can reproduce the same result as in the cellDancer paper, as shown in Fig.R44.

(a) Velocity stream inferred by cellDancer for *Pancreas*, following the tutorial of cellDancer

(b) Velocity stream inferred by cellDancer for *Gastrulation (Erythroid)*, following the tutorial of cellDancer

(c) Velocity stream inferred by cellDancer for *Dentate Gyrus 2*, following the tutorial of cellDancer

Fig. R44 | Velocity stream plots inferred by cellDancer for *Pancreas*, *Gastrulation (Erythroid)* and *Dentate Gyrus 2*, following the preprocessing procedure and the hyperparameter settings in the tutorial of cellDancer.

We compared the performance of cellDancer (in our experiments with uniform settings), cellDancer (following the different settings for different datasets in cellDancer’s tutorial) and TIVelo for these three datasets using three quantitative metrics we used in the manuscript, as shown in Table R7. While some specific dataset and metric combinations show improvement when following the settings in the tutorial of cellDancer, others actually perform worse. In contrast, TIVelo consistently outperforms cellDancer across all datasets and metrics.

Metrics	Methods	Pancreas	Gastrulation	Dentate Gyrus 2
CBDir (Gene space)	cellDancer (uniform settings)	0.0583	-0.2989	0.0404
	cellDancer (tutorial settings)	0.0720	0.0271	0.0035
	TIVelo	0.4850	0.6164	0.4265
TransCosine	cellDancer (uniform settings)	0.0479	0.0001	0.0001
	cellDancer (tutorial settings)	0.2169	0.1258	0.0547
	TIVelo	0.4745	0.4436	0.3742
VeloCoh	cellDancer (uniform settings)	-0.2845	-0.1609	-0.1316
	cellDancer (tutorial settings)	-0.4270	-0.2383	0.0005
	TIVelo	0.1696	0.2504	0.1552

Table R7 | The comparison of cellDancer (in our experiments with uniform settings), cellDancer (following the settings in cellDancer’s tutorial) and TIVelo through three quantitative metrics for *Pancreas*, *Gastrulation (Erythroid)* and *Dentate Gyrus 2*.

UniTVelo

For certain datasets (e.g., Fig.3, S7), the "independent mode" (mode 2) of UniTVelo is more appropriate. This point was also noted by Reviewer 1. We have corrected the corresponding figures in the manuscript (see Fig.R45 and Fig.R46) and updated the quantitative comparison (see Fig.R47-Fig.R51).

Fig. R45 | Comparison of velocity stream plots inferred by UniTVelo mode 1 and mode 2 for *Dentate Gyrus*.

Fig. R46 | Comparison of velocity stream plots inferred by UniTVelo mode 1 and mode 2 for *Retina*.

Fig. R47 | Comparison of TIVelo and six benchmarking methods by cross-boundary direction correctness (Gene space) score across ten datasets.

Fig. R48 | Comparison of TIVelo and six benchmarking methods by transition cosine similarities (TransCosine) across ten datasets.

Fig. R49 | Comparison of TIVelo and six benchmarking methods by velocity coherence (VeloCoh) across ten datasets.

Fig. R50 | Violin plots showing the comparison of sign accuracy from TIVelo and other six benchmarking methods on *RPE1-FUCCI*. The mean sign accuracy of each method is annotated below.

Fig. R51 | Violin plots showing the comparison of sign accuracy from TIVelo and other six benchmarking methods on *U2OS-FUCCI*. The mean sign accuracy of each method is annotated below.

For the *Intestinal Organoid* dataset (Fig.4 of the original UniTVelo paper²), UniTVelo originally set `velo_config.IROOT` to the expected root cell cluster (Stem cells) (https://github.com/StatBiomed/UniTVelo/blob/main/notebooks/Figure4_IntestinalOrganoid.ipynb), which leads to unfairness when comparing with other methods that do not use this information (Fig.R52(a)). To ensure fairness, we set `velo_config.IROOT` to None by default (Fig.R52(b)).

Fig. R52 | Comparison of velocity stream plots inferred by UniTVelo with and without root cell cluster given for *Intestinal Organoid*.

Despite these differences between our reproduction and the results shown in the original paper, the analysis in this part validated that TIVelo outperforms or matches the benchmarking methods in the quantitative comparison, underlining its superior performance in RNA velocity inference.

The discussion on this topic is now presented in Supplementary Notes S9, S10, S11 and S12 of the revised manuscript.

2. Limited Novelty and Justification.

The notion that unspliced RNA consistently precedes spliced RNA is not unique to TIVelo; other methods (e.g., DeepVelo, cellDancer) already account for variable kinetics rather than constant parameters. Thus, the manuscript’s argument that TIVelo offers a more flexible or “model-free” strategy seems incomplete without acknowledging these existing advancements.

The authors should offer a clearer explanation of how TIVelo substantially advances the field beyond prior methods, especially if it does not infer cell-specific rates.

Response: We sincerely thank the reviewer for this valuable comment. To better address this point, we have expanded our discussion comparing TIVelo with DeepVelo and cellDancer, highlighting key methodological differences and TIVelo’s advantages. The following discussion is presented in the Supplementary Notes S1 in the revised manuscript to provide a more comprehensive and detailed argument.

DeepVelo

DeepVelo uses graph convolutional networks (GCN) to estimate cell-specific kinetic rates:

$$(\alpha, \beta, \gamma) = \text{GCN}(U, S|A) \quad (13)$$

where $U, S \in \mathbb{R}^{N \times G}$ are cell by gene unspliced and spliced RNA input, $\alpha, \beta, \gamma \in \mathbb{R}^{N \times G}$ are estimated kinetic rates, and A is the adjacency matrix depicting the cell nearest neighbor graph.

The loss function designed by DeepVelo is as follows

$$\begin{aligned} \mathcal{L}_+ &= \frac{1}{|\Omega|} \sum_{i \in \Omega} \left[s_i + \tilde{v}_i - \sum_{j \in \tilde{\mathcal{N}}_i} s_j P_{c+}(i \rightarrow j) \right]^2, \\ \mathcal{L}_- &= \frac{1}{|\Omega|} \sum_{i \in \Omega} \left[s_i - \tilde{v}_i - \sum_{j \in \tilde{\mathcal{N}}_i} s_j P_{c-}(i \leftarrow j) \right]^2, \\ \mathcal{L}_{\text{Pearson}} &= -(\lambda_u \text{corr}(\tilde{v}_i, u_i) + \lambda_s \text{corr}(\tilde{v}_i, -s_i)), \end{aligned} \quad (14)$$

where Ω is the set of all cells, $\tilde{\mathcal{N}}_i$ is the estimated nearest neighborhood of cell i , and \tilde{v}_i is the estimated velocity vector for the spliced RNA in cell i . $P_{c+}(i \rightarrow j)$ is the normalized binary indicator of cell i ’s neighboring cells j with $S_{\text{cos}}(s_j - s_i, \tilde{v}_i) > 0$, indicating that cells j are the downstream cells of cell i according to the estimated velocity \tilde{v}_i . Similarly, $P_{c-}(i \rightarrow j)$ is the normalized binary indicator of cell i ’s neighboring cells j with $S_{\text{cos}}(s_j - s_i, \tilde{v}_i) < 0$, indicating that cells j are the upstream cells of cell i according to \tilde{v}_i . These binary indicators are normalized across the neighborhood to form probability distributions.

It has been mentioned in the original DeepVelo paper³ that the first two terms in Eq.(14) are symmetric to the sign of \tilde{v}_i , i.e., $\mathcal{L}_+(\tilde{v}_i) + \mathcal{L}_-(\tilde{v}_i) = \mathcal{L}_+(-\tilde{v}_i) + \mathcal{L}_-(-\tilde{v}_i)$. The sign of \tilde{v}_i is determined by the third term, $\mathcal{L}_{\text{Pearson}}$. However, the design of $\mathcal{L}_{\text{Pearson}}$ may introduce bias to genes where cells are all in the induction/repression phase. Here we take dataset *Gastrulation (Erythroid)* as an example.

First, we compared the velocity stream plots inferred by DeepVelo and TIVelo. DeepVelo exhibits a backward velocity flow from Erythroid3 to Erythroid2, as shown in Fig.R53(a) (blue box). This result was generated using DeepVelo’s default parameter settings (<https://github.com/bowang-lab/DeepVelo/blob/main/README.md>). However, as shown in Fig.R54, the result presented in Fig.S2 of the DeepVelo paper³ is different and it required a specific training adjustment: the removal of $\mathcal{L}_{\text{Pearson}}$ from the loss function after 10 training epochs (https://github.com/bowang-lab/DeepVelo/blob/main/examples/mouse_gastrulation.py). This adjustment was not applied in the analysis of the other five datasets discussed in DeepVelo’s paper: $\mathcal{L}_{\text{Pearson}}$ is retained throughout the training process.

Fig. R53 | The comparison of velocity stream plots inferred by DeepVelo and TIVelo in *Gastrulation* (*Erythroid*). Blue box: backward velocity stream estimated by DeepVelo.

Fig. R54 | The comparison of velocity stream plots inferred by DeepVelo, with default parameter setting and with $\mathcal{L}_{\text{Pearson}}$ removed from the loss function after 10 training epochs. (b) is from Fig.S2 of DeepVelo’s paper³.

To further investigate the impact of including $\mathcal{L}_{\text{Pearson}}$ and why the default setting of DeepVelo fails in this dataset, we inspected three genes where all cells are in the **repression phase**, where the velocities for spliced RNA are expected to be negative for all cells. For each gene g , we visualized the fitted velocities $\tilde{v}_i^{(g)}$ for spliced RNA given by DeepVelo for a subset of sampled cells i . From the results shown in Fig.R55, cells i with high u_i and low s_i tend to have positive $\tilde{v}_i^{(g)}$ (green box), while cells i with low u_i and high s_i tend to have negative $\tilde{v}_i^{(g)}$ (yellow box).

This behavior arises from the design of the $\mathcal{L}_{\text{Pearson}}$ term in the loss function (14). As a result of this term, cells with high u_i and low s_i tend to have positive velocities, and cells with low u_i and high s_i tend to have negative velocities, regardless of the transcriptional phase of cells. This is particularly problematic for genes with all cells entirely in the repression or induction phase, where velocities should be negative or positive across all cells, respectively. This inherent limitation in the design of $\mathcal{L}_{\text{Pearson}}$ in DeepVelo leads to the backward velocity stream from Erythroid3 and Erythroid2 observed in Fig.R53(a) (blue box).

Fig. R55 | The velocity inferred by DeepVelo for three genes in *Gastrulation (Erythroid)*. Here only the velocities for spliced RNA (\tilde{v}_i) are shown. Green box: cells with positive \tilde{v}_i due to high u_i and low s_i . Yellow box: cells with negative \tilde{v}_i due to low u_i and high s_i .

In contrast, TIVelo does not have the inherent constraints imposed by $\mathcal{L}_{\text{Pearson}}$ in DeepVelo, allowing genes with all cells in the repression (or induction) phase to exhibit negative (or positive) velocities across all cells, as shown in Fig.R56.

Fig. R56 | The velocity inferred by TIVelo for three genes in *Gastrulation (Erythroid)*. Here only the velocities for spliced RNA (\tilde{v}_i) are shown.

cellDancer

cellDancer trains a deep neural network (DNN) Φ_{θ^g} for each gene g independently to estimate cell-specific kinetic rates ($\alpha_i^g, \beta_i^g, \gamma_i^g$) for cell i :

$$(\alpha_i^g, \beta_i^g, \gamma_i^g)^T = \Phi_{\theta^g}(u_i^g, s_i^g) \quad (15)$$

For each gene g , the loss function is designed as follows:

$$\mathcal{L} = \sum_{i=1}^n \mathcal{L}_i \quad (16)$$

$$\mathcal{L}_i = 1 - \max_{\{i'\}} \frac{v_i \cdot v_{i'}}{|v_i| \cdot |v_{i'}|}$$

where $v_i = (v_i^u, v_i^s)$, $v_i^u = \alpha_i^g - \beta_i^g u_i^g$, $v_i^s = \beta_i^g u_i^g - \gamma_i^g s_i^g$. Cell i' is in the nearest neighborhood of cell i and $v_{i'} = (u_{i'}^g - u_i^g, s_{i'}^g - s_i^g)$.

For each gene, cellDancer minimizes \mathcal{L}_i by selecting a cell i' in the nearest neighborhood of cell i . However, this neighboring cell i' can vary across different genes for the same cell i , potentially

leading to errors in velocity direction estimation. This issue is illustrated using the dataset *Intestinal Organoid*.

First, we compared the velocity stream plots inferred by cellDancer and TIVelo. In *Intestinal Organoid*, the Stem cells differentiate into Goblet cells and Paneth cells in the Secretory lineage, and differentiate into Enterocytes in the Enterocytes lineage (Fig.R57(a)). cellDancer’s velocity estimation fails to accurately capture the differentiation trajectory from Stem cells into two distinct lineages, as shown in Fig.R57(b) (red boxes).

Fig. R57 | The comparison of velocity stream plots inferred by TIVelo and cellDancer in *Intestinal Organoid*.

To further investigate this issue of cellDancer, we inspected six genes in this dataset, and visualized the velocities (v_i^u, v_i^s) estimated by cellDancer. For the first three genes, the estimated velocities indicate a reversed direction of differentiation from Enterocytes to Stem cells, as shown in Fig.R58.

Fig. R58 | The velocity inferred by cellDancer for six genes in *Intestinal Organoid*. Here both the velocities for unspliced RNA and for spliced RNA are shown.

We also inspected a randomly selected subset of five cells and, for each cell i , identified its corresponding neighboring cell i' for each gene as defined in Eq.(16). In Eq.(16), the selected cell i' should be the downstream cells to cell i in differentiation. The results presented in Fig.R59 reveal that for the first three genes, cellDancer tends to select the upstream cells of cell i as i' , while for the last three genes, cellDancer tends to select cell i' from the downstream cells of cell i . The error in velocity estimation for such genes arises from the error in neighboring cell selection in cellDancer.

This error in the selection of neighboring cell i' for the top three genes arises from the flexible design of cellDancer's loss function (16), which does not guarantee that cell i' is selected from biologically plausible downstream cells of cell i . Consequently, the neighboring cell i' may be selected from the upstream cells of cell i , as demonstrated in Fig.R59 (top row).

Fig. R59 | Velocity estimation from cellDancer and the cells i' for five randomly selected cells i . Red points: selected cells i . Green points: the cells i' for corresponding cell i .

Fig. R60 | Velocity estimation $v_i / -v_i$ and cell i' selected based on $v_i / -v_i$ for one randomly selected cell i . Red point: selected cell i . Green points: cell i' for cell i . Light green points: neighboring cells of cell i .

The design of L_i in Eq.(16) in cellDancer is too flexible with respect to the direction of v_i . Even if v_i is inverted or incorrect, we may still find a neighboring cell i' that makes L_i small. To further illustrate this limitation introduced by L_i in Eq.(16), we inverted the direction of inferred velocities from cellDancer for one randomly selected cell i and a gene *Gsto1*, i.e., setting v_i to $-v_i$. We then selected cell i' for both v_i and $-v_i$, and compared the cosine similarity calculated in Eq.(16) based on v_i and $-v_i$. Notably, the cosine similarity for $-v_i$ is higher than that for v_i , leading to an even lower loss L_i compared to that with v_i , as shown in Fig.R60. This demonstrates that the design of the loss L_i in cellDancer is too flexible, and regardless of the direction of v_i , there may exist a neighboring cell i' that makes L_i small.

In the contrary, in TIVelo, we selected the same directed nearest neighborhood $dnn(i)$ for each cell i across different genes (Fig.R61), based on the overall direction on the main path. This strategy provides constraints on the inferred velocity of individual genes, ensuring that the velocity will point to the downstream cells, bypassing the limitation of L_i of Eq.(16) in cellDancer. In Fig.R61, each blue arrow represents the RNA velocity for each selected cell and each gene, which is obtained based on the ensemble of neighboring cells in the dNN of the selected cell. In some cases, the velocity does not point towards the mean expression value of cells in the dNN, which is due to the regularization term in our loss objective to enhance the consistency of velocity vector of similar cells (Methods).

Fig. R61 | Velocity estimation from TIVelo and the directed nearest neighborhood (dNN) for one randomly selected cell. Red point: selected cell i . Green points: dNN for cell i .

In addition, we developed a new mode (**kinetic rate mode**) for fitting RNA velocity in TIVelo, which enables simultaneous inference of cell-specific kinetic rates (α , β and γ) for each gene, while maintaining the framework's core functionality. Please see our detailed response to **reviewer 4's general comments**.

The discussion on this topic is now presented in Supplementary Notes S1 of the revised manuscript.

3. Insufficient Clarity and Context.

(1) The statement that scVelo's velocity vectors are reversed (lines 223–225) lacks supporting visual evidence in Fig.3a, which does not include scVelo results. The manuscript needs additional figures or references to substantiate this claim.

Response: We sincerely thank the reviewer for their insightful comment. We acknowledge that there was a typo in our original manuscript: the correct figure reference should be "Fig.3d (top row)" instead of "Fig.3a". This reversed velocity pattern is also illustrated in Fig.S1c of our original manuscript (see Fig.R62 and Fig.R63). We have now updated the relevant section to properly reference Fig.3d (top row) and Fig.S1c.

Also, this point was mentioned by DeepVelo, which is in Fig.2(a) of the DeepVelo paper³, as shown in Fig.R64.

Fig. R62 | Fig.3(d) in our original manuscript. The top row shows the reversed velocity stream from Granule mature to Granule immature in the result of scVelo.

Fig. R63 | Fig.S1(c) in our original manuscript, showing the reversed velocity stream from Granule mature to Granule immature in the result of scVelo.

Fig.R64 | Fig.2(a) in DeepVelo, showing the reversed velocity stream from Granule mature to Granule immature in the result of scVelo.

(2) In the supplementary file (lines 438–439), the reasoning behind using `scvelo.tl.terminal_states` requires further contextualization. The claim that root/end cells “typically exhibit extreme values in unspliced and spliced RNA” lacks proper references and does not universally apply to all biological systems.

Response: We sincerely thank the reviewer for the thoughtful questions regarding the identification of terminal states in TIVelo. In our framework, terminal states (root cluster or end clusters) are identified using `scvelo.tl.terminal_states`¹⁹. The origin node is selected as the cell cluster exhibiting the strongest signal as the root cluster or the end cluster among the inferred terminal states.

The sentence stated in our original manuscript was: terminal (root/end) cells typically exhibit extreme values in u and s , making them easily identified by ODE-based methods such as scVelo. We would like to clarify that the actual procedure for our origin node selection is more nuanced than merely based on extreme values in u and s and involves the following steps:

Firstly, it calculates the transition matrix $\pi_{c,c'} = \cos \angle(\delta_{c,c'}, v_c)$, where v_c is the inferred velocity of cell c , $\delta_{c,c'} = s_{c'} - s_c$ and s_c is the spliced expression of cell c . $\pi_{c,c'}$ is normalized by

$$\tilde{\pi}_{c,c'} = \frac{1}{z_c} \exp\left(\frac{\pi_{c,c'}}{\sigma_c^2}\right) \quad (17)$$

with row normalization factors $z_c = \sum_{c'} \exp\left(\frac{\pi_{c,c'}}{\sigma_c^2}\right)$ and kernel width parameters σ_c optionally adjusted for each cell locally.

Secondly, the end and root clusters are obtained as stationary states of the velocity-inferred transition matrix $\tilde{\pi}$ and its transpose $\tilde{\pi}^T$, respectively. A root score vector μ^{root} and an end score vector μ^{end} will be calculated for all cells, which is given by left eigenvectors corresponding to an eigenvalue of 1, that is

$$\mu^{\text{end}} = \mu^{\text{end}} \tilde{\pi}, \quad \mu^{\text{root}} = \mu^{\text{root}} \tilde{\pi}^T. \quad (18)$$

An example of inferred μ^{root} and μ^{end} by scVelo for dataset *HSPCs* is shown in Fig.R65.

Fig. R65 | Root score (μ^{root}) and end score (μ^{end}) for *HSPCs* inferred by scVelo.

Finally, the cell cluster with the largest mean root score will be selected as the origin node. If there is no cell cluster with a root score larger than a threshold (0.1), we select the one with the largest mean end score as the origin node (Methods).

For multi-branch datasets with multiple expected end clusters, this strategy for selecting the origin node remains effective, as long as one of the end clusters or the root cluster is selected as the origin node. We illustrate this using the *Intestinal Organoid* dataset with two lineages.

As shown in Fig.R66(a), the origin node is selected based on the root score. In this dataset, two cell clusters (Enterocytes and Paneth cells) exhibit relatively high root scores (red boxes). Regardless of whether the Enterocytes or Paneth cell cluster is selected as the origin node, TIVelo produces the same (and correct) velocity inference (Fig.R66(b)). Depending on whether the Enterocytes or Paneth cell is selected as the origin node, different main paths are inferred, as demonstrated in Fig.R67(a) and (b). The mean orientation scores inferred on both main paths are negative, and TIVelo correctly infers the velocity direction along both main paths.

Fig. R66 | (a) Root score (μ^{root}) of *Intestinal Organoid* inferred by scVelo. (b) Velocity stream plot inferred by TIVelo, based on either Enterocytes or Paneth cell selected as the origin node.

Fig. R67 | The main path selection in *Intestinal Organoid*. Main path selection 1 is based on the case when Enterocytes is selected as the origin node, while main path selection 2 is based on the case when Paneth cells is selected as the origin node.

While `scvelo.tl.terminal_states` does not directly select cell clusters based on extreme u and s values, we observed that the origin node selected by `scvelo.tl.terminal_states` often exhibits extreme values of unspliced and spliced RNA. We performed the following analysis for several multi-branch datasets, to demonstrate that the origin node selected exhibits a relatively high or low expression level of spliced RNA in the data:

1. For each gene g , we calculated the mean expression of s for each cell cluster i , denoted as $s_{i,g}$;
2. For each gene g and each cell cluster i , we calculated the relative expression as $\tilde{s}_{i,g} = s_{i,g}/s_{g,\max}$, where $s_{g,\max}$ are the maximum expression values of s in gene g .
3. We visualized the distribution of $\tilde{s}_{i,g}$ using violin plots for each cell cluster i .

The results shown in Fig. R68 confirm that origin nodes typically exhibit extreme (high or low) relative s expression levels compared to other cell clusters.

Fig. R68 | The violin plot of relative expression of s for each cell cluster in several multi-branch datasets. The violin for the origin node is marked by darker color. The median values of each violin are annotated.

Consistent with the theory of transcriptional kinetics, terminal states (root or end clusters) typically occupy the initiation or termination phases of transcriptional processes for most genes. Therefore, these states tend to exhibit extreme relative expression levels of s , which can be easily identified using `scvelo.tl.terminal_states`.

The discussion on this topic is now presented in Supplementary Notes S4 of the revised manuscript.

4. Lack of Broader Comparisons

The study would benefit from direct comparisons with additional tools such as DeepVelo and CellRank 2, which also address limitations in constant-parameter modeling. Incorporating these methods would give a more comprehensive benchmark and help situate TIVelo within the broader landscape of RNA velocity approaches.

Response: We thank the reviewer for their comment. We have included DeepVelo as a benchmarking method in our quantitative comparison part (Fig.5 and Fig.7 of the revised manuscript), as shown in Fig.R69-R73. TIVelo outperforms or matches the benchmarking methods, underlining its superior performance in RNA velocity inference.

Fig. R69 | Comparison of TIVelo and six benchmarking methods by cross-boundary direction correctness (Gene space) score across ten datasets.

Fig. R70 | Comparison of TIVelo and six benchmarking methods by transition cosine similarities (TransCosine) across ten datasets.

Fig. R71 | Comparison of TIVelo and six benchmarking methods by velocity coherence (VeloCoh) across ten datasets.

Fig. R72 | Violin plots showing the comparison of sign accuracy from TIVelo and other six benchmarking methods on *RPE1-FUCCI*. The mean sign accuracy of each method is annotated below.

Fig. R73 | Violin plots showing the comparison of sign accuracy from TIVelo and other six benchmarking methods on *U2OS-FUCCI*. The mean sign accuracy of each method is annotated below.

Regarding CellRank2, we agree that it is a powerful tool for downstream analysis of RNA velocity, such as inferring initial/terminal cell states, fate probabilities, and expression dynamics. However, CellRank2 is not primarily designed for RNA velocity estimation itself. Instead, CellRank2 primarily integrates pre-computed RNA velocity estimates as one of its multiple input modalities for downstream analyses. As such, we believe it is more appropriate to use CellRank2 as a downstream analytical tool rather than a direct benchmarking method for RNA velocity inference.

References

- [1] Minji Kang, Jose Juan Almagro Armenteros, Gunsagar S Gulati, Rachel Gleyzer, Susanna Avagyan, Erin L Brown, Wubing Zhang, Abul Usmani, Noah Earland, Zhenqin Wu, et al. Mapping single-cell developmental potential in health and disease with interpretable deep learning. *BioRxiv*, pages 2024–03, 2024.
- [2] Mingze Gao, Chen Qiao, and Yuanhua Huang. Unitvelo: temporally unified rna velocity reinforces single-cell trajectory inference. *Nature Communications*, 13(1):6586, 2022.
- [3] Haotian Cui, Hassaan Maan, Maria C Vladoiu, Jiao Zhang, Michael D Taylor, and Bo Wang. Deepvelo: deep learning extends rna velocity to multi-lineage systems with cell-specific kinetics. *Genome Biology*, 25(1):27, 2024.
- [4] Shengyu Li, Pengzhi Zhang, Weiqing Chen, Lingqun Ye, Kristopher W Brannan, Nhat-Tu Le, Jun-ichi Abe, John P Cooke, and Guangyu Wang. A relay velocity model infers cell-dependent rna velocity. *Nature biotechnology*, 42(1):99–108, 2024.
- [5] Adam Gayoso, Philipp Weiler, Mohammad Lotfollahi, Dominik Klein, Justin Hong, Aaron Streets, Fabian J Theis, and Nir Yosef. Deep generative modeling of transcriptional dynamics for rna velocity analysis in single cells. *Nature methods*, 21(1):50–59, 2024.
- [6] Philipp Weiler, Marius Lange, Michal Klein, Dana Pe’er, and Fabian Theis. Cellrank 2: unified fate mapping in multiview single-cell data. *Nature Methods*, 21(7):1196–1205, 2024.
- [7] Lars Velten, Simon F Haas, Simon Raffel, Sandra Blaszkiewicz, Saiful Islam, Bianca P Hennig, Christoph Hirche, Christoph Lutz, Eike C Buss, Daniel Nowak, et al. Human haematopoietic stem cell lineage commitment is a continuous process. *Nature cell biology*, 19(4):271–281, 2017.
- [8] Sung-Woo Park, Guohua Zhen, Catherine Verhaeghe, Yasuhiro Nakagami, Louis T Nguyenvu, Andrea J Barczak, Nigel Killeen, and David J Erle. The protein disulfide isomerase agr2 is essential for production of intestinal mucus. *Proceedings of the National Academy of Sciences*, 106(17):6950–6955, 2009.
- [9] Fang Zhao, Robert Edwards, Diana Dizon, Kambiz Afrasiabi, Jennifer R Mastroianni, Mikhail Geyfman, André J Ouellette, Bogi Andersen, and Steven M Lipkin. Disruption of paneth and goblet cell homeostasis and increased endoplasmic reticulum stress in agr2^{-/-} mice. *Developmental biology*, 338(2):270–279, 2010.
- [10] Eric P Bennett, Ulla Mandel, Henrik Clausen, Thomas A Gerken, Timothy A Fritz, and Lawrence A Tabak. Control of mucin-type o-glycosylation: a classification of the polypeptide galnac-transferase gene family. *Glycobiology*, 22(6):736–756, 2012.
- [11] Rebecca M Fox, Caitlin D Hanlon, and Deborah J Andrew. The creba/creb3-like transcription factors are major and direct regulators of secretory capacity. *Journal of Cell Biology*, 191(3):479–492, 2010.
- [12] Yong H Sheng, Rohan Lourie, Sara K Lindén, Penny L Jeffery, Deborah Roche, Thu V Tran, Chin W Png, Nigel Waterhouse, Philip Sutton, Timothy HJ Florin, et al. The muc13 cell-surface mucin protects against intestinal inflammation by inhibiting epithelial cell apoptosis. *Gut*, 60(12):1661–1670, 2011.
- [13] Fei Gao, Hengwei Wu, Xin Jin, Jimin Shi, Yi Luo, Yanmin Zhao, and He Huang. Hspa5 deficiency blocks intestinal enterocyte differentiation in graft-versus-host disease. *Blood*, 142:2049, 2023.

- [14] Elke Kaemmerer, Ursula Schneider, Christina Klaus, Patrick Plum, Andrea Reinartz, Maximilian Adolf, Marcus Renner, Tim GAM Wolfs, Boris W Kramer, Norbert Wagner, et al. Increased levels of deleted in malignant brain tumours 1 (dmbt1) in active bacteria-related appendicitis. *Histopathology*, 60(4):561–569, 2012.
- [15] Russell E McConnell and Matthew J Tyska. Myosin-1a powers the sliding of apical membrane along microvillar actin bundles. *The Journal of cell biology*, 177(4):671–681, 2007.
- [16] Zhan-Qi Cao and Xiu-Li Guo. The role of galectin-4 in physiology and diseases. *Protein & cell*, 7(5):314–324, 2016.
- [17] Marius Lange, Volker Bergen, Michal Klein, Manu Setty, Bernhard Reuter, Mostafa Bakhti, Heiko Lickert, Meshal Ansari, Janine Schniering, Herbert B Schiller, et al. Cellrank for directed single-cell fate mapping. *Nature methods*, 19(2):159–170, 2022.
- [18] Joana Silva, Ferhat Alkan, Sofia Ramalho, Goda Snieckute, Stefan Prekovic, Ana Krotenberg Garcia, Santiago Hernández-Pérez, Rob van der Kammen, Danielle Barnum, Liesbeth Hoekman, et al. Ribosome impairment regulates intestinal stem cell identity via $\text{zak}\alpha$ activation. *Nature Communications*, 13(1):4492, 2022.
- [19] Volker Bergen, Marius Lange, Stefan Peidli, F Alexander Wolf, and Fabian J Theis. Generalizing rna velocity to transient cell states through dynamical modeling. *Nature biotechnology*, 38(12):1408–1414, 2020.
- [20] Blanca Pijuan-Sala, Jonathan A Griffiths, Carolina Guibentif, Tom W Hiscock, Wajid Jawaid, Fernando J Calero-Nieto, Carla Mulas, Ximena Ibarra-Soria, Richard CV Tyser, Debbie Lee Lian Ho, et al. A single-cell molecular map of mouse gastrulation and early organogenesis. *Nature*, 566(7745):490–495, 2019.
- [21] Chen Qiao and Yuanhua Huang. Representation learning of rna velocity reveals robust cell transitions. *Proceedings of the National Academy of Sciences*, 118(49):e2105859118, 2021.

Response to Reviewers

We would like to express our sincere gratitude to all reviewers for their time and effort in evaluating our manuscript. Their constructive comments and suggestions have significantly improved the quality of our work.

Responses to Reviewer 1

(Remarks to the Author):

I thank the authors for the detailed revision; all my concerns have been well addressed.

One more note is on Fig. 4a, where UniTVelo's results are reversed from its original report. Possibly, the author may briefly mention the reason, for example, something like "Of note, the result of UniTVelo is different from its original report, probably due to the use of (default) random initialization here, while the original study used a warm initialization."

Response: We sincerely appreciate your careful review and helpful suggestion. We have added the proposed clarification to the manuscript to prevent potential misunderstandings regarding Fig.4a.

(Remarks on code availability):

On briefly looked at some notebooks. Seems all fine; probably it needs to use a publicly available data directory (or path).

Response: Thank you for this valuable suggestion. We have updated the README file in our GitHub repository to include the Figshare data path, making it easier for users to access the required datasets.

Responses to Reviewer 2

Response: We sincerely appreciate the time and effort invested by both reviewers in evaluating our work. The constructive feedback has been invaluable in improving our manuscript.

Responses to Reviewer 3

(Remarks to the Author):

The authors have addressed all my concerns, and I am happy to recommend the manuscript for publication in Nature Communications.

Response: We are grateful for your positive assessment and recommendation. Thank you for your time and thoughtful comments throughout the review process.

(Remarks on code availability):

I have assessed the provided code and found it to be well-documented and user-friendly. The repository includes a comprehensive README file with clear instructions on installation, dependencies, and execution of the main analysis pipeline. I was able to install the required packages and run the core components of the code without major issues.

Response: We appreciate you taking the time to test our code package. We are committed to maintaining and improving this resource for the research community.

Responses to Reviewer 4

(Remarks to the Author):

The authors addressed my concerns. I have no more questions.

Response: We sincerely appreciate your time and valuable feedback, which has helped strengthen our manuscript.